# A Survey of Self-Evolving Agents
## What, When, How, and Where to Evolve on the Path to Artificial Super Intelligence

**Huan-ang Gao**$^{\gamma\dagger}$, **Jiayi Geng**$^{\alpha\dagger}$, **Wenyue Hua**$^{\epsilon\dagger}$, **Mengkang Hu**$^{\omega\dagger}$, **Xinzhe Juan**$^{\sigma\mu\dagger}$, **Hongzhang Liu**$^{\xi\dagger}$, **Shilong Liu**$^{\alpha\dagger}$, **Jiahao Qiu**$^{\alpha\delta\dagger}$, **Xuan Qi**$^{\gamma\dagger}$, **Qihan Ren**$^{\sigma\dagger}$, **Yiran Wu**$^{\rho\dagger}$, **Hongru Wang**$^{k\dagger\boxtimes}$, **Han Xiao**$^{\tau\dagger}$, **Yuhang Zhou**$^{\lambda\dagger}$, **Shaokun Zhang**$^{\rho\dagger}$, **Jiayi Zhang**$^{\pi}$, **Jinyu Xiang**, **Yixiong Fang**$^{\theta}$, **Qiwen Zhao**$^{\zeta}$, **Dongrui Liu**$^{\sigma}$, **Cheng Qian**$^{\beta}$, **Zhenhailong Wang**$^{\beta}$, **Minda Hu**$^{\tau}$, **Huazheng Wang**$^{\eta}$, **Qingyun Wu**$^{\rho}$, **Heng Ji**$^{\beta}$, **Mengdi Wang**$^{\alpha\delta\boxtimes}$

$^{\alpha}$*Princeton University*, $^{\delta}$*Princeton AI Lab*, $^{\gamma}$*Tsinghua University*, $^{\theta}$*Carnegie Mellon University*, $^{\xi}$*University of Sydney*, $^{\sigma}$*Shanghai Jiao Tong University*, $^{\rho}$*Pennsylvania State University*, $^{\mu}$*University of Michigan*, $^{\eta}$*Oregon State University*, $^{\tau}$*The Chinese University of Hong Kong*, $^{\lambda}$*Fudan University*, $^{\pi}$*The Hong Kong University of Science and Technology (Guangzhou)*, $^{\omega}$*The University of Hong Kong*, $^{\epsilon}$*University of California, Santa Barbara*, $^{\zeta}$*University of California San Diego*, $^{k}$*University of Edinburgh*, $^{\beta}$*University of Illinois Urbana-Champaign*

**Github Repo:** `https://github.com/CharlesQ9/Self-Evolving-Agents`
$^{\dagger}$**Equal contribution and the order is determined alphabetically,** $^{\boxtimes}$**Corresponding Author**

**Reviewed on OpenReview:** `https://openreview.net/forum?id=CTr3bovS5F`

## Abstract

Large Language Models (LLMs) have demonstrated remarkable capabilities across diverse tasks but remain fundamentally static, unable to adapt their internal parameters to novel tasks, evolving knowledge domains, or dynamic interaction contexts. As LLMs are increasingly deployed in open-ended, interactive environments, this static nature has become a critical bottleneck, necessitating agents that can adaptively reason, act, and evolve in real time. This paradigm shift —from scaling static models to developing self-evolving agents — has sparked growing interest in architectures and methods enabling continual learning and adaptation from data, interactions, and experiences. This survey provides the first systematic and comprehensive review of self-evolving agents, organizing the field around three foundational dimensions — *what to evolve, when to evolve, and how to evolve.* We examine evolutionary mechanisms across agent components (e.g., models, memory, tools, architecture), categorize adaptation methods by stages (e.g., intra-test-time, inter-test-time), and analyze the algorithmic and architectural designs that guide evolutionary adaptation (e.g., scalar rewards, textual feedback, single-agent and multi-agent systems). Additionally, we analyze evaluation metrics and benchmarks tailored for self-evolving agents, highlight applications in domains such as coding, education, and healthcare, and identify critical challenges and research directions in safety, scalability, and co-evolutionary dynamics. By providing a structured framework for understanding and designing self-evolving agents, this survey establishes a roadmap for advancing more adaptive, capable, robust, and versatile agentic systems in both research and real-world deployments, and ultimately sheds light on the realization of Artificial Super Intelligence (ASI) where agents evolve autonomously and perform beyond human-level intelligence across a wide array of tasks.

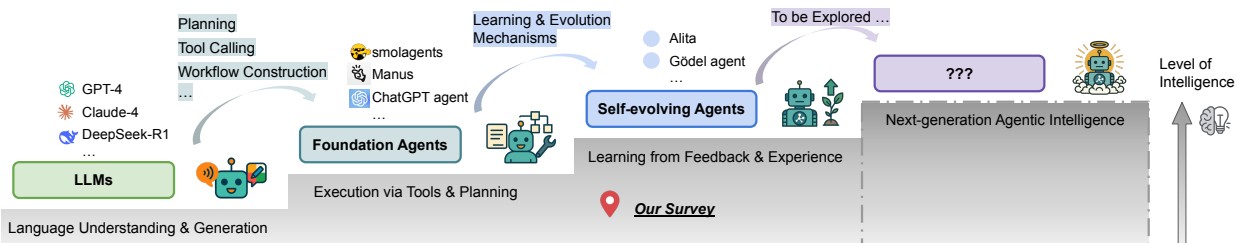

Figure 1: A conceptual trajectory illustrating the progression from large language models (LLMs) to foundation agents, and then to self-evolving agents—our focus, and ultimately toward the hypothetical Artificial Super Intelligence (ASI). Along this path, intelligence and adaptivity increase, marking a shift toward more autonomous and agentic AI systems. The future directions beyond self-evolving agents remain open and subject to ongoing exploration.

# 1 Introduction

> *"It is not the most intellectual of the species that survives; it is not the strongest that survives; but the species that survives is the one that is able best to adapt and adjust to the changing environment in which it finds itself."*
> — Charles Darwin[1]

Large Language Models (LLMs) have demonstrated remarkable capabilities across a wide range of tasks. Yet, they remain fundamentally static (Luo et al., 2025a), unable to adapt their internal parameters when encountering novel tasks, evolving knowledge domains, or dynamic interaction contexts. As LLMs are increasingly deployed in open-ended, interactive environments, this limitation becomes a critical bottleneck. In such settings, conventional knowledge retrieval mechanisms prove inadequate, giving rise to agents capable of dynamically adapting their perception, reasoning, and actions in real time. This emerging need for dynamic, continual adaptation signals a conceptual shift in artificial intelligence: *from scaling up static models to developing self-evolving agents.* While established techniques like Supervised Fine-Tuning (SFT) and Reinforcement Learning (RL) provide the mechanisms for improvement, we define self-evolution not merely by the algorithms used, but by the locus of autonomy. Unlike traditional pipelines where human engineers curate data and schedule updates, a self-evolving agent is capable of continuously learning from new data, interactions, and experiences in real-time, leading to systems that are more robust, versatile, and capable of tackling complex, dynamic real-world problems (Wang et al., 2024a). This shift is currently driving us toward a promising and transformative path to Artificial Super Intelligence (ASI), where the agents not only can learn and evolve from experience with an unpredictable speed but also perform at or above human-level intelligence across a wide array of tasks (Wang et al., 2025g).

Unlike static LLMs, which remain constrained by their inability to adapt to novel and evolving contexts, self-evolving agents are designed to overcome these limitations by continuously learning from real-world feedback. This progression reshapes our understanding of agents. Self-evolving agents, as a core concept, represent a significant step forward in the evolution of intelligent systems, acting as intermediaries that pave the way for more adaptive and autonomous AI, as shown in Figure 1. Recent research initiatives have increasingly focused on developing adaptive agent architectures capable of continually learning and adapting from experience, such as recent advancements in agent frameworks (Yin et al., 2025), prompting strategies (Fernando et al., 2023), and different optimization ways to evolve. Notwithstanding these advances, existing surveys predominantly address agent evolution as a subsidiary component within comprehensive agent taxonomies. Previous surveys primarily provide systematic overviews of general agent development, while offering limited coverage of self-evolving mechanisms across constrained scenarios in self-evolving agents (Luo et al., 2025a; Liu et al., 2025a).

---

[1]This quote is widely attributed to Charles Darwin, but it does not appear verbatim in his writings. The phrasing is believed to originate from Professor Leon C. Megginson, who paraphrased Darwin's ideas. Despite its frequent misattribution, the quote effectively captures the essence of Darwinian evolution and has since been popularized in both scientific and managerial literature.

For example, Luo et al. (2025a) discuss several ways to evolve, such as self-learning and multi-agent co-evolution, while Liu et al. (2025a) explicitly introduce the evolution in terms of different components of agents, such as tools and prompts. Moreover, some studies focus specifically on the evolution of language models themselves (Tao et al., 2024), rather than on the broader concept of agents. However, these works address isolated components rather than the holistic agent system. Therefore, there is no systematic survey devoted to a dedicated, comprehensive investigation of self-evolving agents as a first-class research paradigm. This gap has left fundamental questions underexplored: ***What aspects of an agent should evolve? When should adaptation occur? And how should that evolution be implemented in practice?***

To the best of our knowledge, this is the first systematic and comprehensive survey focusing on self-evolving agents, offering a clear roadmap for both theoretical inquiry and practical deployment. However, given that this represents a rapidly forming research area where conceptual boundaries are still being actively negotiated within the community, we frame this survey as a guiding synthesis rather than a review of a fully established paradigm. Instead of enforcing rigid boundaries, we aim to structure the heterogeneous mechanisms emerging in the community into a coherent framework. We organize our analysis around three foundational questions — *what, when, and how to evolve* — and provide a structured framework for understanding each. Specifically, we systematically examine individual agent components, including the model, memory, tools and corresponding workflow, investigating their distinct evolutionary mechanisms (what to evolve of agent in Section 3); then we divide existing evolving methods according to different temporal stages with different learning paradigms such as supervised fine-tuning, reinforcement learning and inference-time evolving (when to evolve in Section 4). We finally summarize different signals to guide the evolution of agents, such as textual feedback or scalar rewards, and also different architectures of agents to evolve, such as single-agent and multi-agent evolution (how to evolve in Section 5). Furthermore, we review certain evaluation metrics and benchmarks to track existing advancements of self-evolving agents, emphasizing the importance of co-evolution between evaluation and agents (Section 6). We also examine emerging applications in domains such as coding, education, and healthcare, where continual adaptation and evolution are essential (Section 7). Finally, we identify persistent challenges and outline promising research directions to guide the development of self-evolving agents (Section 8). Through this systematic decomposition of self-evolutionary processes across orthogonal dimensions, we provide a structured and practical framework enabling researchers to systematically analyze, compare, and design more robust and adaptive agentic systems. To sum up, our key contributions are as follows:

- We establish a unified theoretical framework for characterizing self-evolutionary processes in agent systems, anchored around three fundamental dimensions: what evolves, how it evolves, and when it evolves, providing clear design guidance for future self-evolving agentic systems.

- We further investigate the evaluation benchmark or environment tailored for self-evolving agents, highlighting emerging metrics and challenges related to adaptability, robustness, and real-world complexity.

- We showcase several key real-world applications across various domains, including autonomous software engineering, personalized education, healthcare, and intelligent virtual assistance, illustrating the practical potential of self-evolving agents.

- We identify critical open challenges and promising future research directions, emphasizing aspects like safety, personalization, multi-agent co-evolution, and scalability.

In doing so, our survey provides researchers and practitioners with a more structured taxonomy for understanding, comparing, and advancing research of self-evolving agents from different perspectives. As LLM-based agents are increasingly integrated into mission-critical applications, understanding their evolutionary dynamics becomes essential, extending beyond academic research to encompass industrial applications, regulatory considerations, and broader societal implications.

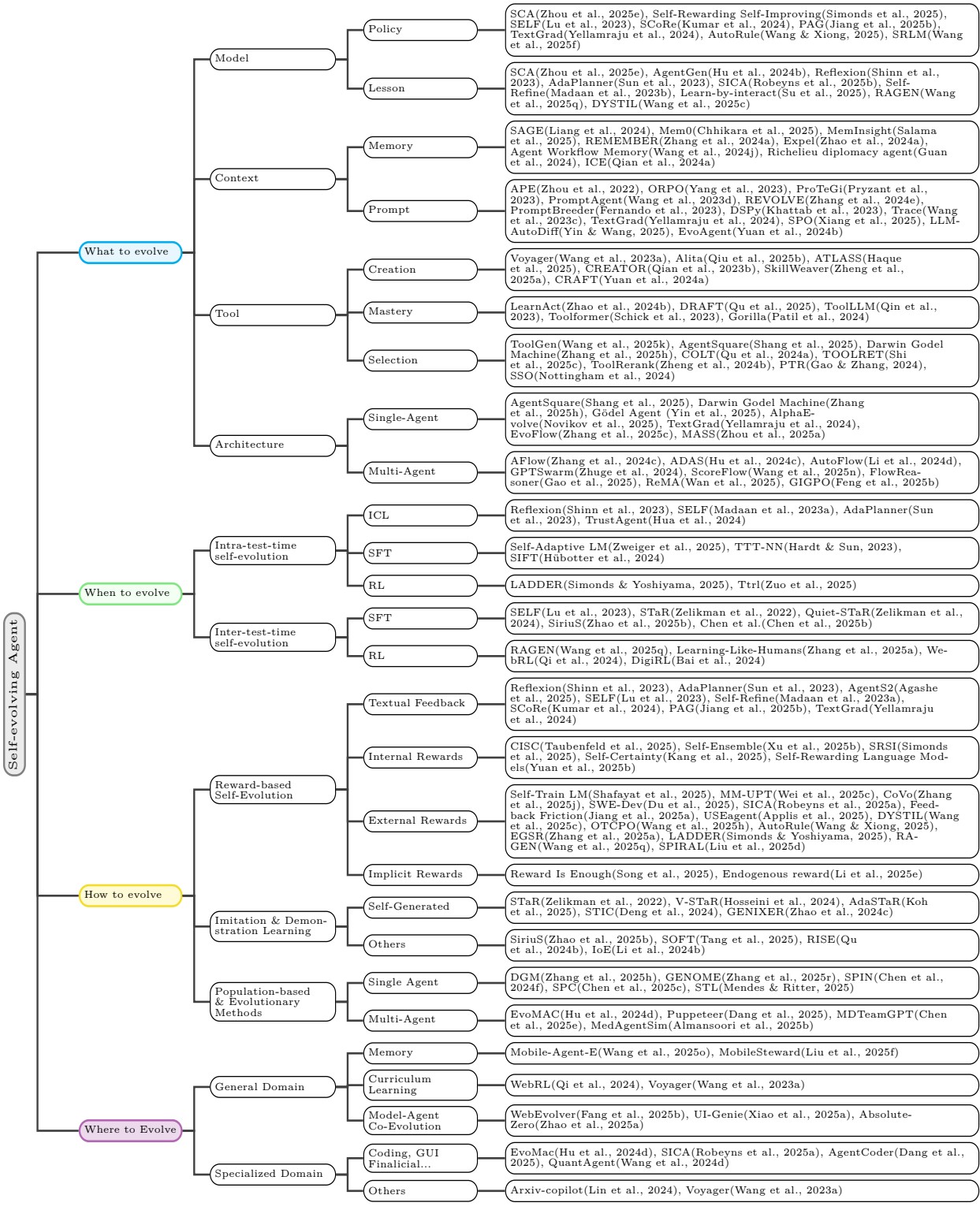

Figure 2: Taxonomy of self-evolving agents, in which agents are analyzed along the *what, when, how,* and *where* dimensions, with selected representative methods and systems annotated at each leaf node.

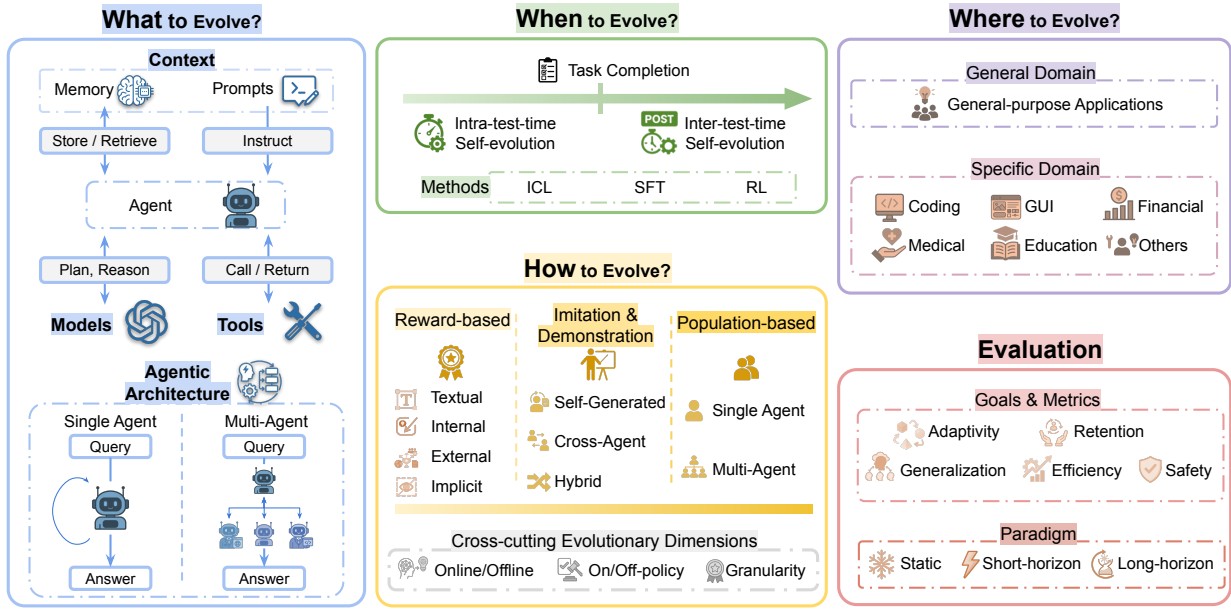

Figure 3: **A comprehensive overview of self-evolving agents across key dimensions.** From left to right and top to bottom, the figure mirrors the organization of Sections 3–7. **What to evolve** (Sec. 3) decomposes agent components: model, context, tools, and architecture, showing where evolution operates. **When to evolve** (Sec. 4) distinguishes intra-test-time versus inter-test-time self-evolution, corresponding to ICL, SFT, and RL paradigms. **How to evolve** (Sec. 5) summarizes methodological families—reward-based, imitation & demonstration, and population-based—together with cross-cutting dimensions such as online/offline, on/off-policy, and reward granularity. **Where to evolve** (Sec. 6) contrasts general-purpose and domain-specific deployments (e.g., coding, GUI, finance, medical, education). **Evaluation** (Sec. 7) outlines goals and metrics—adaptivity, generalization, efficiency, safety—and corresponding evaluation paradigms (static, short-horizon, long-horizon). Overall, the taxonomy maps the survey's reasoning flow: defining *what, when, and how* to evolve establishes the foundation for evaluating and advancing self-evolving agents.

## 2 Definitions and Foundations

Before delving into a comprehensive survey, we first present a formal definition of self-evolving agents and introduce a taxonomy of the key aspects in self-evolving agents. We also discuss the relationships between self-evolving agents and other renowned learning paradigms, such as curriculum learning, lifelong learning, model editing, and unlearning, highlighting the adaptive, dynamic, and autonomous nature of self-evolving agents.

### 2.1 Definitions

**Environment** We first define the environment (including the user and the execution environment, e.g., Linux shell) of an agent system as a partially observable Markov Decision Process (POMDP), represented as a tuple $E = (\mathcal{G}, \mathcal{S}, \mathcal{A}, T, R, \Omega, O, \gamma)$, where:

- $\mathcal{G}$ is a set of potential goals. Each $g \in \mathcal{G}$ is a task objective that the agent needs to achieve, e.g., a user query.

- $\mathcal{S}$ is a set of states. Each $s \in \mathcal{S}$ represents the internal state of the environment.

- $\mathcal{A}$ is a set of actions. Each action $a \in \mathcal{A}$ can be a combination of textual reasoning, retrieval of external knowledge, and tool calls.

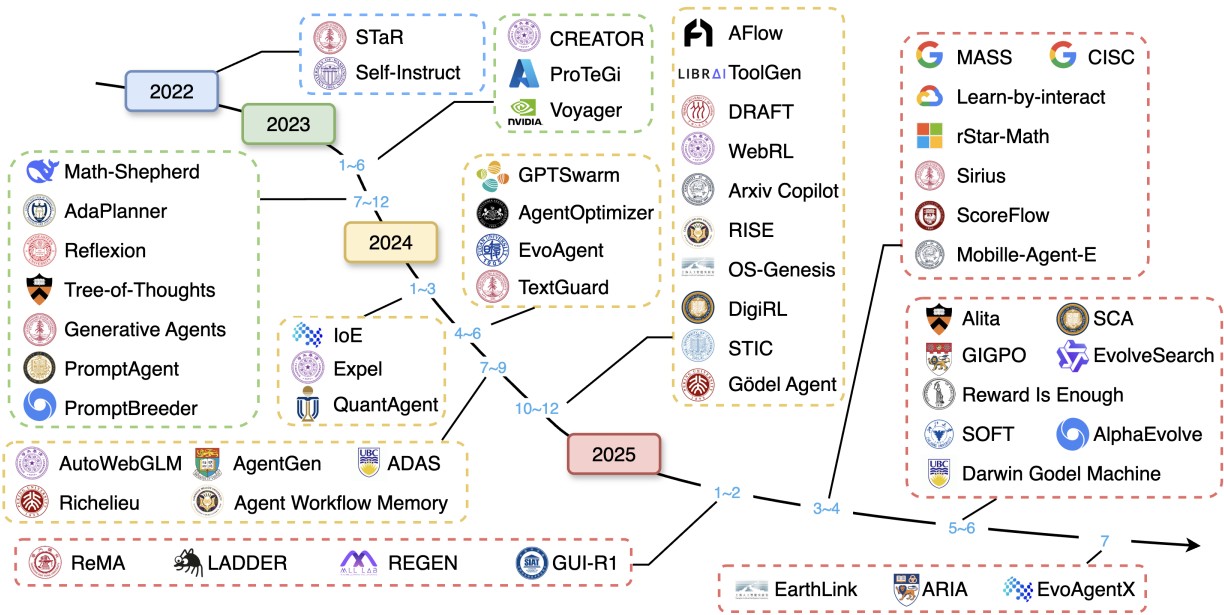

Figure 4: An evolutionary landscape of several representative self-evolving agent frameworks from 2022 to 2025. The figure chronologically organizes major research milestones in the development of self-evolving agents with capabilities such as autonomous planning, tool use, and continual self-improvement.

- $T$ is the state transition probability function which takes a state-action pair $(s, a)$ and outputs the probability distribution $T(s'|s, a)$ of the next state.

- $R : \mathcal{S} \times \mathcal{A} \times \mathcal{G} \to \mathcal{R}$ is the feedback/reward function, conditioned on the specific goal $g \in \mathcal{G}$. The feedback $r = R(s, a, g)$ typically takes the form of a scalar score or textual feedback.

- $\Omega$ is a set of observations accessible to the agent.

- $O$ is the observation probability function which takes a state-action pair $(s, a)$ and outputs the probability distribution $O(o'|s, a)$ of the next observation for the agent.

- $\gamma$ is the discount factor.

**Agent system**  We define a (multi-)agent system as $\Pi = (\Gamma, \{\psi_i\}, \{C_i\}, \{\mathcal{W}_i\})$. The architecture $\Gamma$ determines the control flow of the agent system or collaborative structures between multiple agents. It is typically represented as a sequence of nodes $(N_1, N_2, ...)$ organized by graph or code structures. Each node $N_i$ consists of the following components:

- $\psi_i$: the underlying LLM/MLLM.

- $C_i$: the context information, e.g., prompt $P_i$ and memory $M_i$.

- $\mathcal{W}_i$: the set of available tools/APIs.

At each node, the agent policy is a function $\pi_{\theta_i}(\cdot|o)$ that takes an observation and outputs the probability distribution of the next action, where $\theta_i = (\psi_i, C_i)$. The actual action space here is the union of the natural language space and the tool space $\mathcal{W}_i$.

For a given task $\mathcal{T} = (E, g)$, represented by an environment $E$ and a corresponding goal $g \in \mathcal{G}$, the agent system follows the topology $\Gamma$ to generate a trajectory $\tau = (o_0, a_0, o_1, a_1, ...)$, and receives a feedback $r$ either from the external environment or from internal signals (e.g., self-confidence or feedback from an evaluator).

**Self-evolving strategy**   A self-evolving strategy is a transformation $f$ that maps the current agent system to a new state, conditioned on the generated trajectory $\tau$ and the external/internal feedback $r$:

$$f(\Pi, \tau, r) = \Pi' = (\Gamma', \{\psi_i'\}, \{C_i'\}, \{\mathcal{W}_i'\}) \tag{1}$$

**Objective of self-evolving agents**   Let $U$ be a utility function that measures the performance of an agent system $\Pi$ on a given task $\mathcal{T}$ by assigning a scalar score $U(\Pi, \mathcal{T}) \in \mathbb{R}$. The utility may be derived from the task-specific feedback $r$, such as a reward signal or textual evaluation, possibly combined with other performance indicators (e.g., completion time, accuracy, or robustness). Given a sequence of tasks $(\mathcal{T}_0, \mathcal{T}_1, ..., \mathcal{T}_n)$ and an initial agent system $\Pi_0$, a self-evolving strategy $f$ recurrently generates an evolving sequence of agent systems $(\Pi_1, \Pi_2, ..., \Pi_n)$ via

$$\Pi_{j+1} = f(\Pi_j, \tau_j, r_j), \tag{2}$$

where $\tau_j$ and $r_j$ are the trajectory and feedback on task $\mathcal{T}_j$.

The overarching objective in designing a self-evolving agent is to construct a strategy $f$ such that the cumulative utility over tasks is maximized:

$$\max_f \sum_{j=0}^{n} U(\Pi_j, \mathcal{T}_j) \tag{3}$$

**Operational definition of self-evolving agents**   To provide a conceptual boundary, we introduce an operational definition of self-evolving agents. A self-evolving agent is the agent that *modifies its internal parameters, contextual state, toolset, or architectural topology based on its own trajectories or feedback signals, with the explicit objective of improving future performance.*

This definition entails three inclusion criteria: (i) updates must be *experience-dependent*, driven by trajectories, self-generated data, or environment feedback, specifically targeting the agent's policy limitations or capability boundaries rather than generic data synthesis; (ii) updates must produce a *persistent, policy-changing* effect rather than a transient instruction-following behavior; (iii) the system must possess mechanisms for *autonomous exploration or self-initiated learning*, even if it also leverages pre-collected data. For clarity, we use "passive" to denote learning triggered exclusively by externally provided data or schedules, and "active" to denote self-initiated exploration, reflection, or structural modification (i.e., using self-reflection to collect data), explicitly excluding static pipelines (e.g., standard distillation) where data generation is agnostic to the agent's interaction history.

As this field is rapidly forming, fully autonomous self-evolution without human intervention represents an aspirational goal rather than the current norm. In this survey, we do not impose a rigid exclusion threshold that would disregard early-stage developments. Instead, we analyze the mechanisms contributing to the self-evolving paradigm ranging from **proto-evolution** (e.g., iterative bootstrapping or feedback-driven prompting) to **strong self-evolution** (fully autonomous diagnosis and reconfiguration), allowing us to provide a comprehensive view of how diverse methods contribute to the "What, When, and How" of the paradigm's progression toward full autonomy.

## 2.2   Relationships with Other Works

Table 1 summarizes the key distinctions between self-evolving agents and other paradigms (including curriculum learning, lifelong learning, model editing, and unlearning). We provide a brief introduction to each paradigm below, highlighting the differences among these paradigms, as well as the differences with self-evolving agents.

**Curriculum Learning**   Curriculum learning is a training strategy in which data are presented in order of increasing difficulty (Bengio et al., 2009; Wang et al., 2021). This strategy resembles human curricula where concepts are introduced progressively from simple to complex. Curriculum learning has been widely adopted across diverse domains, including computer vision (Guo et al., 2018; Jiang et al., 2014; Liu et al., 2023a),

natural language processing (Platanios et al., 2019; Tay et al., 2019), speech recognition (Braun et al., 2017; Lotfian & Busso, 2019), etc. Recently, several curriculum learning-based methods have been proposed to fine-tune LLMs during the post-training phase (Wang et al., 2025p; Zhang et al., 2025o; Parashar et al., 2025; Zhang et al., 2025a; Li et al., 2025b). The framework for curriculum learning generally comprises two key components: a difficulty measurer that quantifies the difficulty level of each training data point, and a training scheduler that reorganizes the order of data points received by the model according to the difficulty level. Unlike curriculum learning, which operates on a static dataset, self-evolving agents aim to handle sequential tasks in dynamic environments. Additionally, curriculum learning updates only model parameters, whereas self-evolving agents are able to adjust non-parametric components like memory and tools.

**Lifelong Learning**  Lifelong learning refers to the ability of AI models to continuously and adaptively learn when exposed to new tasks and environments, while retaining previously acquired knowledge and abilities. This learning paradigm, also known as continual learning or incremental learning, is crucial for AI models to operate in dynamic and complex environments (Wang et al., 2024c; Zheng et al., 2025c; Parisi et al., 2019; Shi et al., 2024; Yang et al., 2025d; Zhou et al., 2024a). The primary goal of lifelong learning for AI models is to achieve a balance between preserving existing knowledge (stability) and acquiring new knowledge (plasticity) when exposed to new data or tasks (McCloskey & Cohen, 1989; Zheng et al., 2025c; Ratcliff, 1990; Rolnick et al., 2019). Though it shares the sequential task setting with self-evolving agents, lifelong learning differs in two fundamental ways: (1) *Memory functionality and usage timing*: While continual learning methods extensively employ memory mechanisms (e.g., experience replay buffers (Rolnick et al., 2019), episodic memory (Lopez-Paz & Ranzato, 2017)) to mitigate catastrophic forgetting, these mechanisms primarily serve as *training-time* tools for parameter optimization through gradient computation. In contrast, self-evolving agents leverage *runtime context* (prompts, working memory, conversation history) that directly influences action generation at test-time without requiring parameter updates. The distinction lies not in the presence of non-parametric components, but in their functional role: training-time replay vs. test-time state adaptation. (2) *Learning initiative*: Lifelong learning primarily acquires knowledge passively through externally provided task sequences, whereas self-evolving agents actively explore their environment and incorporate internal reflection or self-evaluation mechanisms to guide their own learning trajectory. Recent self-improving LLM methods (Huang et al., 2022; Yuan et al., 2024d), which iteratively refine models through self-generated data and self-critique, can be viewed as instances of lifelong learning focused on model-centric improvement. Self-evolving agents extend beyond this paradigm to encompass system-wide evolution including tool acquisition, architectural reconfiguration, and environmental exploration.

**Model Editing and Unlearning**  Model editing and unlearning aim to efficiently and precisely modify specific knowledge in AI models while preserving irrelevant knowledge and avoiding full retraining (Wang et al., 2024f; 2025i; Zhang et al., 2024d; Wang et al., 2025i; Nguyen et al., 2022; Geng et al., 2025a). A canonical application of model editing is to perform efficient and precise localized factual updates (e.g., modifying the answer to "2021 Olympics host city" from "Tokyo" to "Paris"). Early methods focused on triples of atomic knowledge and later expanded into various trustworthy-related tasks (Fang et al., 2025a; Huang et al., 2025a). Recent studies also propose lifelong model editing(Chen et al., 2024c) that sequentially performs model editing. For model unlearning, early efforts mainly focus on the removal of privacy-related information (Chen et al., 2021). With the rapid development of LLMs, model unlearning is also used to enhance LLMs' safety (Zhang et al., 2024j; Li et al., 2024c; Zou et al., 2024; Lu et al., 2025). Compared to lifelong learning, model editing shares an aligned objective: both aim to acquire new knowledge or capabilities while mitigating catastrophic forgetting. However, lifelong learning typically relies on extensive gradient-based fine-tuning across all model parameters, whereas model editing often modifies only a small subset of parameters in a targeted manner. Compared to self-evolving agents, model editing (1) cannot modify non-parametric components such as memory or tools, and (2) relies on a pre-defined pipeline from the algorithm designer, whereas self-evolving agents can spontaneously employ more diverse and flexible strategies based on the observation of the environment or internal feedback signals.

**Positioning Self-Evolving Agents**  To clarify the relationships among these paradigms and to motivate the role of self-evolving agents, we examine them through two complementary perspectives: a *problem-setting*

Table 1: Comparison between self-evolving agents and other renowned paradigms

| Paradigm | Runtime Context | Evolving Toolset | Dynamic Tasks | Test-time Adaptation | Active Exploration | Structural Change | Self-reflect & Eval |
|---|---|---|---|---|---|---|---|
| Curriculum Learning | ✗ | ✗ | ✗ | ✗ | ✗ | ✗ | ✗ |
| Lifelong Learning | ✗ | ✗ | ✓ | ✗ | ✗ | ✗ | ✗ |
| Model Editing | ✗ | ✗ | ✓ | ✓ | ✗ | ✗ | ✗ |
| **Self-evolving Agents** | ✓ | ✓ | ✓ | ✓ | ✓ | ✓ | ✓ |

lens and a *solution-paradigm* lens. This distinction clarifies the basis of each paradigm - whether it emerges from constraints and challenges inherent to the learning setting, or from methodological proposals for how the model or agent itself can be updated.

- **Problem-setting view.** Curriculum learning and lifelong learning arise from concrete learning problems. Curriculum learning addresses how to structure training examples of varying difficulty so a model can handle complex samples more effectively; lifelong learning focuses on acquiring new abilities over time while mitigating catastrophic forgetting. These paradigms are therefore driven by the *problems* they aim to solve and primarily specify how experience is organized for the learner, rather than how the agent itself may adapt beyond parameter updates.

- **Solution-paradigm view.** Model editing and self-evolving agents, in contrast, originate as *solutions*: they propose mechanisms for updating or modifying a system. Model editing provides targeted procedures—typically localized parameter adjustments—to correct or insert knowledge. Self-evolving agents generalize this idea by treating adaptation as a first-class capability, allowing not only parameter updates but also changes to runtime context, memory, tools, and workflow structures, driven by the agent's own trajectories and feedback signals.

Viewed through this two-lens framework, curriculum and lifelong learning are anchored in the nature of the learning *problems* they address, whereas model editing and self-evolving agents are defined by the *methods* they provide for effecting change. Self-evolving agents thus represent a system-level solution paradigm: they include parameter-level editing as one update pathway while enabling broader, persistent, and interaction-driven evolution across multiple components of an agent.

## 3 What to Evolve?

A self-evolving agent differs from a static agent not by *what* components it contains, but by *which internal states* can be autonomously modified based on its own trajectories, reflections, and feedback signals. Thus, the key question of this section is to identify the *evolutionary loci* within an agent system $\Pi = (\Gamma, \{\psi_i\}, \{C_i\}, \{\mathcal{W}_i\})$—the parts of the system whose states can be rewritten in an experience-driven and persistent manner, enabling cumulative self-improvement.

Following the formulation in Section 2.1, these evolutionary loci align with four major pillars of an agent system. Our investigation starts at the agent's cognitive core, namely the **Models** $\{\psi_i\}$, whose parameters can be continuously updated through self-generated supervision, execution traces, or environmental feedback (Zhou et al., 2025e; Wang et al., 2025q). We then consider the **Context** $\{C_i\}$ –including instructions (Xiang et al., 2025; Khattab et al., 2023) and long-term memory (Chhikara et al., 2025; Wang et al., 2024j) –which evolves as agents reflect, store, and retrieve experience in ways that shape future decision-making. From this internal foundation, we examine the evolution of **Tools** $\{\mathcal{W}_i\}$, where agents autonomously create (Qiu et al., 2025b), refine (Qu et al., 2025), and managing (Wang et al., 2025k) executable skills based on verifiable interaction signals Finally, we scale to the **Agentic Architecture**, where the system's **architecture** (Hu et al., 2024c; Zhang et al., 2024c) and collaborative structures (Wan et al., 2025) are optimized over time, enabling structural adaptation beyond individual components. We present representative examples of these evolutionary loci in Table 2.

Table 2: Representative self-evolving agent methods positioned along four evolutionary pillars; a filled bullet (●) marks dimensions where the approach actively evolves.

| Method | Model | | Context | | Tool | | | Architecture | |
|---|---|---|---|---|---|---|---|---|---|
| | Policy | Experience | Prompt | Memory | Creation | Mastery | Selection | Single | Multi |
| SCA(Zhou et al., 2025e) | ● | ● | ○ | ○ | ● | ○ | ○ | ○ | ○ |
| RAGEN(Wang et al., 2025q) | ● | ● | ● | ○ | ○ | ○ | ○ | ● | ○ |
| AgentGen (Hu et al., 2024b) | ○ | ● | ● | ● | ● | ○ | ○ | ● | ○ |
| Promptbreeder(Fernando et al., 2023) | ○ | ○ | ● | ○ | ○ | ○ | ○ | ● | ○ |
| Expel(Zhao et al., 2024a) | ○ | ● | ○ | ● | ○ | ○ | ○ | ○ | ○ |
| Agent Workflow Memory(Wang et al., 2024j) | ○ | ○ | ○ | ● | ○ | ○ | ● | ○ | ○ |
| Mem0(Chhikara et al., 2025) | ○ | ○ | ○ | ● | ○ | ○ | ○ | ○ | ○ |
| MAS-Zero(Ke et al., 2025) | ○ | ○ | ● | ○ | ○ | ○ | ○ | ○ | ● |
| Multi-Agent Design(Zhou et al., 2025a) | ○ | ○ | ● | ○ | ○ | ○ | ● | ○ | ● |
| SPO(Xiang et al., 2025) | ○ | ○ | ● | ○ | ○ | ○ | ○ | ○ | ○ |
| Alita(Qiu et al., 2025b) | ○ | ○ | ○ | ○ | ● | ○ | ● | ○ | ○ |
| TextGrad(Yellamraju et al., 2024) | ○ | ○ | ● | ○ | ○ | ● | ● | ● | ○ |
| DGM(Zhang et al., 2025h) | ○ | ○ | ● | ○ | ○ | ○ | ○ | ● | ○ |
| AlphaEvolve(Novikov et al., 2025) | ○ | ○ | ● | ○ | ● | ● | ○ | ● | ○ |
| ADAS(Hu et al., 2024c) | ○ | ○ | ● | ○ | ● | ○ | ○ | ● | ● |
| AFlow(Zhang et al., 2024c) | ○ | ○ | ● | ○ | ● | ○ | ● | ● | ● |
| ReMA(Wan et al., 2025) | ○ | ○ | ○ | ○ | ○ | ○ | ○ | ○ | ● |
| SkillWeaver(Zheng et al., 2025a) | ○ | ○ | ○ | ● | ● | ● | ● | ○ | ○ |
| LearnAct(Zhao et al., 2024b) | ○ | ○ | ○ | ● | ● | ○ | ● | ○ | ○ |
| DRAFT(Qu et al., 2025) | ○ | ○ | ● | ○ | ● | ● | ○ | ○ | ○ |
| ToolGen(Wang et al., 2025k) | ○ | ○ | ○ | ● | ● | ● | ○ | ○ | ○ |
| CRAFT(Yuan et al., 2024a) | ○ | ○ | ○ | ○ | ● | ● | ● | ○ | ○ |
| CREATOR(Qian et al., 2023b) | ○ | ○ | ○ | ○ | ● | ● | ○ | ○ | ○ |
| Voyager(Wang et al., 2023a) | ○ | ○ | ● | ● | ● | ● | ● | ○ | ● |

## 3.1 Models

Models constitute a primary *locus of self-evolution*, as their parameters can be autonomously rewritten based on the agent's own trajectories, reflections, and interaction outcomes. The ability of these models to evolve by continually adapting their internal parameters and expanding their functional capabilities is essential for the development of autonomous, general-purpose agents. Unlike static systems that rely heavily on human-annotated datasets and fixed training regimes, self-evolving models can improve through interaction, self-supervised data generation, and dynamic learning loops, thereby achieving greater efficiency, adaptability, and scalability.

In detail, we outline the principal axes along which model evolution unfolds. These include learning from self-generated supervision to refine model weights, evolving through interaction with constructed or external environments, and integrating feedback signals that directly reshape future reasoning behaviors. Together, these strategies represent a shift from passive learning paradigms toward active self-improvement.

**Policy** A self-evolving agent can refine its parameters to perform better on targeted tasks. Traditional methods of data collection for training agents on tool-use benchmarks are costly and often yield limited coverage, while purely synthetic data-generation pipelines typically suffer from inadequate quality. Consequently, recent studies emphasize enabling agents to autonomously generate data to improve their own model weights. One representative approach is the Self-Challenging Agent (SCA)(Zhou et al., 2025e), where a language model alternates roles between a challenger generating executable Code-as-Task problems and an executor solving them. The model then fine-tunes its parameters using trajectories derived from successful solutions, resulting in significant performance gains on complex, multi-step tasks. Similarly, the Self-Rewarding Self-Improving framework(Simonds et al., 2025) implements an internal self-judging mechanism, allowing the model to autonomously generate problems, solve them, and assess its performance, thus producing self-contained fine-tuning data without external annotations. This method demonstrated notable improvements, particularly in complex reasoning tasks. Beyond task creation, another promising research direction involves leveraging interaction feedback directly for parameter updates. For instance, SELF(Lu et al., 2023), SCoRe(Kumar et al., 2024), and PAG(Jiang et al., 2025b) interpret execution traces or natural-language critiques as reward signals within an online Supervised Fine-Tuning (SFT) combined with Reinforcement Learning (RL) framework, enabling continuous policy improvement. TextGrad(Yellamraju et al., 2024) further extends this concept by treating unstructured textual feedback as a differentiable training sig-

nal capable of directly influencing both prompt design and model parameters. Additionally, AutoRule(Wang & Xiong, 2025) converts language-model reasoning traces and preference feedback into explicit rule-based training rewards, enhancing the quality of model outputs through structured reward signals. Collectively, these advancements chart a clear trajectory—from agents autonomously crafting their training tasks to directly refining their parameters based on execution feedback, highlighting the capacity of models to evolve continuously by learning from the data they produce.

**Experience**  Agents can evolve not only by adjusting their internal parameters but also by actively interacting with or even constructing their environments, capturing experiences, and transforming them into learning signals that drive iterative improvement. This environmental loop provides agents with the complexity and diversity required for scalable self-adaptation. The Self-Challenging Agent (SCA)(Zhou et al., 2025e) exemplifies this dynamic at the task level, where the agent autonomously generates novel Code-as-Task problems, executes them, and then filters successful trajectories for retraining itself. AgentGen(Hu et al., 2024b) extends this concept to full-environment generation, synthesizing diverse simulation worlds (in PDDL or Gym-style formats) derived from an initial corpus. It implements a bidirectional evolution loop that progressively adjusts task difficulty, enabling the agent to continuously grow within a dynamically structured curriculum. Reflexion(Shinn et al., 2023) complements this by introducing self-reflective mechanisms, where agents iteratively record natural-language critiques of their previous actions, guiding future behavior to avoid recurring mistakes. Additionally, AdaPlanner(Sun et al., 2023) introduces closed-loop adaptive planning, allowing agents to refine their strategies on-the-fly based on environmental feedback, effectively reshaping action sequences in response to immediate outcomes. Similarly, Self-Refine(Madaan et al., 2023b) employs an iterative refinement loop in which the agent repeatedly critiques and revises its initial outputs, significantly improving task accuracy without explicit retraining. SICA (Self-Improving Coding Agent)(Robeyns et al., 2025b) further pushes the boundary by enabling agents to autonomously edit their underlying code and tools, iteratively enhancing their core reasoning abilities through direct self-modification. From a reinforcement learning perspective, frameworks such as RAGEN(Wang et al., 2025q) and DYSTIL(Wang et al., 2025c) conceptualize multi-step tool-use tasks as Markov Decision Processes, optimizing agent policies through rich environmental rewards and strategy induction loops. RAGEN leverages dense feedback from the environment to iteratively fine-tune action policies, while DYSTIL utilizes high-level strategy advice generated by language models to progressively internalize complex decision-making skills into reinforcement learning agents. Collectively, these approaches highlight a compelling paradigm where self-evolving agents not only leverage self-generated data but actively reshape their environments and internal mechanisms to fuel ongoing learning. Such dynamic interaction loops point toward autonomous, open-ended improvement cycles deeply grounded in experiential adaptation.

## 3.2  Context

An essential component of an LLM agent to be evolved is the context, which shapes how an agent behaves. To start with, we want to interpret two terms, "prompt optimization" and "memory evolution", which have been used in different literature. In most cases, these two terms can be used interchangeably because they both refer to what is included in the context window. Prompt optimization asks "how can we phrase or structure the instructions so the LLM behaves better?", and attends to details such as the wording, ordering. On the other hand, memory evolution asks "how should we store, forget, and retrieve context so that the agent can stay informed and perform better?", which focuses on what past information to surface or archive.

### 3.2.1  Memory Evolution

LLM-based agents are increasingly designed with long-term memory mechanisms that grow and adapt as the agent continues to solve tasks and interacts with its environment(Shan et al., 2025; Qian et al., 2023a). An evolving memory enables the agent to accumulate knowledge, recall past events, and adjust its behavior based on experience. Many works stress that effective memory management is crucial for agent performance(Zhong et al., 2024; Zhang et al., 2025e; Yan et al., 2024). SAGE(Liang et al., 2024) uses the Ebbinghaus forgetting curve to decide what to remember or forget. A-mem(Xu et al., 2025a) updates the agent memory structure to create interconnected knowledge networks through dynamic indexing and linking, following the basic

principles of the Zettelkasten method. Mem0(Chhikara et al., 2025) introduces a two-phase pipeline where the agent first extracts salient facts from recent dialogue and then decides how to update the long-term memory: the agent can ADD new facts, MERGE/UPDATE redundant ones, or DELETE contradictions. Furthermore, Memory-R1 (Yan et al., 2025) presents a reinforcement learning framework to train a dedicated Memory Manager agent that learns to select structured operations like ADD, UPDATE, and DELETE. Such a mechanism ensures the agent's long-term memory is coherent and up-to-date. MemInsight(Salama et al., 2025) augments raw memories with semantic structure, which summarizes and tags past interactions for retrieval later. REMEMBER(Zhang et al., 2024a) combines an LLM with a memory of experiences and uses reinforcement learning signals to decide how to update that memory after each episode. Memento (Zhou et al., 2025b) enables continual adaptation without fine-tuning the LLM's parameters by employing online reinforcement learning to optimize a case-retrieval policy, which allows the agent to learn from past experiences stored in an evolving memory bank. MemGen (Zhang et al., 2025f) introduces a dynamic generative memory that operates in a latent space. It uses a learned memory trigger to decide when to invoke memory and a weaver to construct latent token sequences, enabling a fluid interweaving of reasoning and memory.

A critical aspect of memory evolution is enabling agents to learn heuristics or skills from past experiences. Rather than only retrieving exact past instances, advanced agents distill experiences into more general guidance(Zhao et al., 2024a; Fu et al., 2024). Expel(Zhao et al., 2024a) processes past trajectories to generate insights and rules to guide further interactions. This experiential knowledge accumulation leads to measurable gains, as the agent steadily performs better with more experience. ReasoningBank (Ouyang et al., 2025) further develops this idea by distilling generalizable reasoning strategies from both successful and failed experiences into a structured memory. It also introduces memory-aware test-time scaling to generate diverse experiences on each task. Other systems focus on storing higher-level building blocks of problem-solving. For instance, Agent Workflow Memory(Wang et al., 2024j) records common sub-task sequences (workflows) so that an agent solving a complex task can retrieve and reuse a proven sequence of actions rather than plan from scratch. Similarly, MUSE (Yang et al., 2025a) introduces an experience-driven agent for long-horizon tasks, centered on a hierarchical memory module that organizes experience into strategic, procedural, and tool-use memories. The agent populates this memory through a Plan-Execute-Reflect-Memorize loop, enabling it to learn on the job. In the Richelieu diplomacy agent, the system improves its negotiation strategies by augmenting its memory through self-play games, storing the insights from simulated interactions to refine future decisions(Guan et al., 2024). By generalizing from specific episodes to reusable knowledge, these approaches illustrate how memory evolution turns an agent's one-time experiences into long-term competencies, which leads to agents evolving.

### 3.2.2 Prompt Optimization

While memory evolution focused on what knowledge an agent retains, Prompt Optimization (PO) enables LLM agents to self-evolve by refining the instructions it feeds to the backbone model, which directly alters the model's behavior without modifying model weights(Ramnath et al., 2025). Early research treats instruction design as a search problem. APE(Zhou et al., 2022) generates candidate prompts, scores them on validation examples, and selects the best. ORPO(Yang et al., 2023) extends this idea by letting the model iteratively rewrite its own prompt, guided by feedback on prior outputs. ADO(Lin et al., 2025a) introduces DSP that imposes semantic constraints on iteratively proposed prompts to facilitate finding the optimal prompt. Pro-TeGi(Pryzant et al., 2023) generates natural language "corrections" that are applied as edits to the prompt, forming a textual analogue of gradient descent. PromptAgent(Wang et al., 2023d) casts prompt discovery as Monte-Carlo Tree Search, exploring instruction space strategically, while evolutionary approaches like PromptBreeder(Fernando et al., 2023) maintain a population to discover increasingly effective instructions. REVOLVE(Zhang et al., 2024e) further stabilizes long optimization runs by tracking the trajectory of model responses and applying smoothed updates. Pushing this autonomy to its limit, SPO(Xiang et al., 2025) creates a fully self-contained loop where the model generates its training data and uses pairwise preference comparison on its outputs to refine the prompt, eliminating the need for any external labeled data or human feedback. Collectively, these techniques demonstrate that an agent can autonomously improve its prompting policy, turning prompt text into a learnable component that co-evolves with the agent's experience. To address the brevity bias and context collapse of some optimizers, Agentic Context Engineering (ACE) (Zhang et al., 2025m) treats contexts as comprehensive playbooks that accumulate strategies over time. It uses a

modular agentic process with incremental updates to evolve contexts for both offline prompt optimization and online memory adaptation.

In complex systems, an agent often orchestrates a sequence of LLM calls or collaborates with other agents, making prompt design a multi-node problem. Frameworks such as DSPy represent an entire workflow as a graph whose sub-prompts are jointly tuned for a global objective(Khattab et al., 2023). Trace(Wang et al., 2023c), TextGrad(Yellamraju et al., 2024), and LLM-AutoDiff(Yin & Wang, 2025) generalize this idea by treating each prompt as a parameter in a differentiable program and propagating natural-language "gradients" to refine every step. In collaborative scenarios, Multi-Agent System Search (MASS)(Zhou et al., 2025a) first optimizes individual role prompts and then refines inter-agent communication patterns, while MAS-ZERO(Ke et al., 2025) dynamically proposes and revises role prompts to assemble an effective team for each new problem. Evolutionary systems such as EvoAgent(Yuan et al., 2024b) and AgentSquare(Shang et al., 2025) treat each agent along with prompts as the modules and use mutation and selection to discover specialized teams that outperform hand-crafted designs. These approaches extend PO from a single instruction to the language that defines whole workflows or societies of agents.

### 3.3 Tools

An agent's capabilities are fundamentally defined by the tools it can wield. The trajectory of agent development is marked by a crucial evolution: from being mere tool users to becoming autonomous tool makers. This transition from relying on predefined, static toolsets to enabling agents to autonomously expand and refine their own skills is a critical leap towards cognitive self-sufficiency. This paradigm, where agents dynamically adapt their capabilities, allows them to solve a long tail of complex problems not envisioned by their initial designers. This evolution unfolds across three interconnected fronts: tool discovery, mastery, and management, as detailed in the subsections below.

**Autonomous Discovery and Creation**  The primary impetus for autonomous tool creation is to overcome the inherent limitations of a fixed toolset, granting agents the flexibility to innovate on demand. Methodologies for this now span a spectrum from opportunistic discovery to formalized synthesis. At one end, agents like Voyager build an ever-expanding library of skills through emergent trial-and-error, driven by an intrinsic motivation to explore complex, open-ended environments like Minecraft(Wang et al., 2023a). This exploratory approach is powerful for generating a wide array of skills but may lack precision. In contrast, systems like ATLASS, Alita, and Live-SWE-Agent take a more reactive approach, often creating new tools from scratch or employing retrieval-augmented generation (RAG) to search open-source code repositories the moment a capability gap is identified(Haque et al., 2025; Qiu et al., 2025b;a; Xia et al., 2025a). At the other end of the spectrum lie highly structured frameworks that treat tool creation as a deliberate engineering process. CREATOR, for example, disentangles abstract tool creation (e.g., reasoning about the general structure of a reusable function for averaging temperatures over N days) from concrete tool usage (e.g., deciding how to apply that function to a specific city and time range), which enhances modularity and reusability(Qian et al., 2023b). Even more formally, SkillWeaver analyzes successful human or agent task trajectories to propose, synthesize, and hone new skills into robust, reusable APIs, ensuring a higher degree of initial quality(Zheng et al., 2025a). Furthermore, frameworks like CRAFT demonstrate that creating specialized toolsets for specific domains is essential to complement general-purpose models, enabling expert-level performance without sacrificing adaptability(Yuan et al., 2024a). RL-GPT(Liu et al., 2024) integrates generated code implementations into the RL pipeline, leveraging these as tools to tackle complex tasks while addressing simpler ones directly using a Code-as-Policy approach. This integration dynamically adapts and evolves in response to environmental feedback, enabling continuous improvement. However, this burgeoning autonomy introduces significant challenges, particularly around safety and security. The unconstrained generation of code risks creating tools with exploitable vulnerabilities or unintended harmful behaviors, making automated verification and sandboxing critical areas for future research.

**Mastery Through Iterative Refinement**  The proliferation of self-created tools necessitates a robust mechanism for their mastery; a newly generated tool is often a brittle script, not a reliable function. This is where iterative refinement becomes essential. Frameworks like LearnAct and From Exploration to Mastery establish a critical self-correction loop where the agent learns from its own experience(Zhao et al., 2024b;

Qu et al., 2025). This involves tackling the difficult "credit assignment" problem: determining precisely which line of code or which parameter was responsible for a failure. To do this, the agent analyzes a rich variety of feedback signals—including compiler errors, unexpected API return values, environmental state changes, or even implicit signals from a user's subsequent actions. The goal is not only to debug the tool's underlying code but also to refine its documentation (e.g., its docstring and argument descriptions), which is crucial for improving the agent's ability to understand and correctly use the tool in the future. This refinement process also opens the door for valuable human-agent collaboration. While full autonomy is the ultimate goal, many systems can be designed with a "human in the loop," where a human expert can provide corrections, offer high-level suggestions, or validate a newly created tool. This collaborative approach can significantly accelerate the mastery process and ensure that the agent's skills align with human intentions and safety standards. Ultimately, this self-honing process is what elevates a nascent skill into a dependable capability, ensuring the agent's growing skill library increases not just in quantity, but more importantly, in quality and robustness.

**Scalable Management and Selection** As an agent's mastered skill library grows into the hundreds or thousands, it faces a "curse of abundance." The challenge shifts from creating tools to efficiently managing and selecting from them. A large library creates a massive search space, making traditional retrieval methods slow and inaccurate. To overcome this, ToolGen (Wang et al., 2025k) represents a fundamental paradigm shift by encoding tools as unique tokens within the language model's vocabulary. This elegantly reframes tool retrieval as a generation problem, leveraging the transformer's immense pattern-recognition capabilities to predict the most appropriate tool as a natural continuation of its thought process. TOOLMEM (Xiao et al., 2025b) enables agents to learn and store the strengths and weaknesses of different tools in a dedicated memory. At inference, the agent retrieves this knowledge to make more informed decisions, optimizing tool selection for specific task requirements. Beyond selecting a single tool, advanced agents must also excel at tool composition—learning to chain multiple tools in novel sequences to solve multi-step problems. This is a higher-order management task. Architectural approaches like AgentSquare engage in a form of meta-learning, automatically searching the modular design space of an agent—including its planning, memory, and tool-use components—to find an optimal configuration for complex task execution(Shang et al., 2025). As a logical endpoint to this evolutionary trend, visionary concepts like the Darwin Godel Machine propose a framework for open-ended evolution, where the agent can fundamentally rewrite its own core code. In this vision, the distinction between the agent and its tools blurs, leading to a recursive cascade of self-improvement that transcends tool enhancement alone(Zhang et al., 2025h). In essence, this entire evolutionary path aims to establish a closed and virtuous cycle: a truly autonomous agent that can perceive gaps in its capabilities, create novel solutions, master them through practice, and seamlessly integrate them into a coherently managed and ever-expanding repertoire.

## 3.4 Architecture

The defining feature of next-generation agentic systems is their intrinsic capacity for self-improvement. This marks a fundamental shift from systems with fixed capabilities to those that can autonomously enhance their performance(Liu et al., 2025b). By treating their own internal logic and collaborative structures as optimizable components, these systems can adapt their behavior and design in response to feedback, achieving a level of efficiency and effectiveness that static designs cannot match. This section details how this self-optimization is realized, first by examining improvements within single-agent systems and then by exploring the co-evolution of complex multi-agent systems.

### 3.4.1 Single-Agent System Optimization

**LLM-Invoking Node Optimization** Optimizing a single LLM call is straightforward in isolation, but within an agentic system, it becomes a difficult credit assignment problem, as the effect of any single change is obscured by subsequent steps. Research addresses this by making node-level components optimizable, following two main strategies. The first focuses on refining nodes within a fixed agentic topology. A prime example is TextGrad (Yellamraju et al., 2024), which, inspired by backpropagation, uses "textual gradients" to propagate feedback from the final output backward through the workflow, guiding systematic, local

refinements at each node without altering the system's overall structure. The second, parallel strategy integrates this component-level optimization directly into the search for the system's architecture itself. Under this approach, node characteristics become tunable parameters in a larger search space. For instance, frameworks can embed prompt engineering directly into the search loop, allowing the system to discover not just the optimal workflow but also the most effective instruction for each agent simultaneously (Zhou et al., 2025a). Similarly, EvoFlow (Zhang et al., 2025c) uses evolutionary algorithms to construct heterogeneous workflows by selecting the most suitable LLM for each task from a diverse pool. This holistic strategy enables the discovery of systems that are co-optimized for both their structure and individual agent capabilities, effectively balancing metrics like overall performance and cost (Ye et al., 2025a).

**Autonomous-Agent Optimization**   Building upon the optimization of individual LLM-invoking nodes, a more profound level of self-improvement targets the autonomous agent as a holistic entity. This evolution proceeds along two main fronts: optimizing the agent's high-level architectural design and enabling the agent to directly modify its own source code. The first approach focuses on discovering the optimal agent structure. AgentSquare (Shang et al., 2025) exemplifies this by defining a modular design space of components like planners and memory modules, then using an evolutionary algorithm to find the most effective combination for a given task. The second front involves agents that dynamically rewrite their own operational code. This is seen in radical systems like the Darwin Gödel Machine (Zhang et al., 2025h), which recursively modifies its own Python codebase, and AlphaEvolve (Novikov et al., 2025), which uses evolutionary coding to improve specific algorithms. Similarly, Gödel Agent (Yin et al., 2025) provides a self-referential framework for agents to analyze and alter their logic. Together, these two directions (optimizing the agent's architectural "blueprint" and its functional code) demonstrate a key trend toward turning the agent's fundamental structure and logic into learnable components. MemEvolve (Zhang et al., 2025g) introduces a meta-evolutionary framework that evolves not just the agent's experiential memory, but the memory system's architecture itself. Through a bilevel optimization process, it adapts the mechanisms for encoding, storing, and retrieving information to better suit specific task domains.

### 3.4.2   Multi-Agent System Optimization

How agents are organized and communicate within a system (its topology) fundamentally determines its capacity for solving complex problems. The field has evolved from using fixed, human-designed communication structures to creating dynamic systems that automatically adapt their organization to a given task, allowing them to discover and exploit the most effective collaboration patterns. This evolution is explored along two major fronts: the optimization of static, explicit workflows and the co-evolution of dynamic, internal policies.

**Agentic Workflow Optimization**   The optimization of agentic workflows focuses on finding the most effective, often static, structure of communication and task delegation for a given problem. Early research established important foundations, with studies like AutoFlow (Li et al., 2024d) demonstrating the automated creation of linear workflows from natural language, and GPTSwarm (Zhuge et al., 2024) proposing a unifying graph-based framework. Concurrently, other foundational work explored how agents could evolve by using symbolic learning to distill their interaction experiences into an explicit, interpretable set of logical rules to guide future decisions (Zhou et al., 2024b). This abstraction of systems into tunable components—whether nodes, edges, or symbolic rules—was crucial. However, these early systems often lacked a formal method for efficiently navigating the vast space of possible configurations and interactions.

The major breakthrough came when ADAS (Hu et al., 2024c) and AFlow (Zhang et al., 2024c) formally defined this challenge as a search and optimization problem. ADAS set a theoretical vision by framing system design as a search through a Turing-complete space of code-based configurations. Building on this, AFlow made it practical by introducing reusable operators that represent common agentic patterns and by employing Monte Carlo Tree Search (MCTS) to efficiently navigate the enormous design space. Together, these works established a core methodology for treating agent system design as a tractable optimization problem, proving that automatically discovered workflows could outperform human-designed ones.

Following this formalization, research rapidly diversified toward creating customized agent systems for each specific query. Two primary strategies emerged: search-based and learning-based generation. Search-based

methods, such as MaAS (Zhang et al., 2025d), create a "supernet" of potential architectures and then sample a specialized system from it. In parallel, learning-based methods train models to generate effective topologies directly. ScoreFlow (Wang et al., 2025n), for instance, trains a generator using a novel preference optimization method, while FlowReasoner (Gao et al., 2025) uses reinforcement learning to train a meta-agent that constructs a bespoke workflow on the fly. This line of query-specific generation continues to be an active area of research (Ye et al., 2025b; Ke et al., 2025). Furthermore, it is important to note that this process is not limited to the topology alone; many of these frameworks also perform node-level optimization in tandem, such as co-optimizing prompts or selecting heterogeneous models as an integral part of the architectural generation process (Zhang et al., 2024c; Zhou et al., 2025a; Zhang et al., 2025c).

A key challenge for all search and learning methods is the computational cost of evaluating each potential workflow (Shang et al., 2025). To address this, researchers have developed lightweight prediction models. Agentic Predictor (Trirat et al., 2025) is a prime example, training a model to accurately estimate a workflow's performance based on its structural and semantic features without a full execution. By providing a fast and inexpensive evaluation proxy, these predictors significantly accelerate the optimization process, making the exploration of vast design spaces feasible (Zhang et al., 2025s).

**Multi-Autonomous-Agent Optimization** Distinct from optimizing a system's explicit workflow structure, this line of research focuses on how multiple autonomous agents can co-evolve their internal behavioral policies through interaction. This approach enables emergent capabilities like coordination, task delegation, and beneficial competition. For instance, ReMA (Wan et al., 2025) uses multi-agent reinforcement learning (MARL) to collaboratively train a high-level meta-thinker and a low-level executor, significantly improving performance on reasoning benchmarks. Building on this, GiGPO (Feng et al., 2025b) enhances MARL training by aggregating trajectories to provide more precise credit assignment, boosting success rates on long-horizon tasks. To support this direction, platforms like MARTI (Liao et al., 2025) provide open-source infrastructure for orchestrating and scaling the training of these language-model collectives. Collectively, these studies underscore multi-agent reinforcement learning as a promising route for cultivating group-level competencies unattainable by individual agents alone.

## 4 When to Evolve

The temporal dimension of self-evolution in LLM-based agents mainly concerns the relationship between learning processes and task execution. Therefore, the second key aspect of a self-evolving agent is identifying the *evolving timing*, i.e., at which stage the self-evolving strategy $f$ is invoked and applied to the agent system. To this end, we propose a taxonomy that distinguishes between two temporal modes of self-evolution: Intra-test-time self-evolution and inter-test-time self-evolution.

**Intra-test-time self-evolution** refers to adaptive processes that occur during task execution, where agents recognize their limitations on a specific problem and initiate targeted learning mechanisms to enhance their capabilities in real-time (Xi et al., 2024; Bi et al., 2024). This mode of evolution is characterized by its immediate coupling with the task at hand: the agent improves its problem-solving abilities for a specific problem encountered, creating a dynamic interplay between performance and adaptation.

**Inter-test-time self-evolution** refers to learning processes that occur between task completions, leveraging accumulated experiences to improve future performance. This category encompasses diverse methodological approaches: offline learning paradigms that extract knowledge from pre-collected datasets through iterative refinement (Zelikman et al., 2022; 2024), and online learning paradigms that continuously adapt based on streaming interaction data (Qi et al., 2024; Qiu et al., 2025b; Qian et al., 2024b; Wang et al., 2025o).

The implementation of self-evolution across these temporal phases leverages three fundamental learning paradigms in LLMs: in-context learning (ICL) (Dong et al., 2022; Min et al., 2021; Wies et al., 2023), which adapts behavior through contextual examples without modifying parameters; supervised fine-tuning (SFT), which updates model weights through gradient-based optimization on labeled data (Devlin et al., 2018; Shen, 2024; Dong et al., 2023); and reinforcement learning (RL), which shapes behavior through reward-driven policy optimization (Kaelbling et al., 1996; Sun et al., 2024a; Zhang et al., 2025n). While these learning

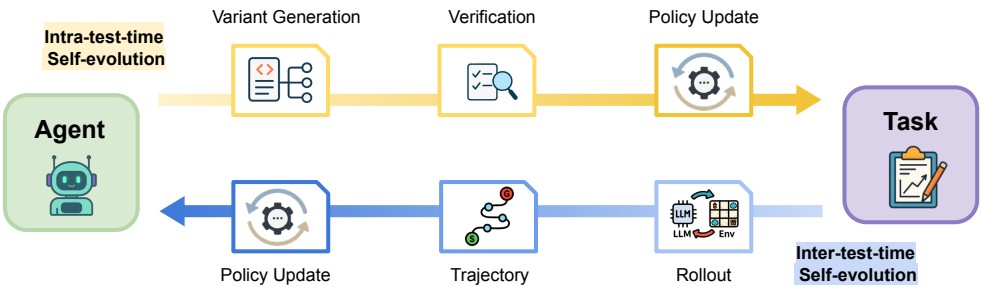

Figure 5: An overview of when to evolve. The top pathway illustrates intra-test-time self-evolution, where adaptation (e.g., variant generation, verification, and policy update) occurs within task execution. The bottom pathway depicts inter-test-time self-evolution, where learning happens retrospectively through rollout, trajectory analysis, and policy updates.

paradigms remain conceptually consistent across temporal contexts, their instantiation differs in terms of data availability and learning objectives:

Intra-test-time is characterized by its online nature: learning data emerges dynamically during task execution, with optimization directly targeting performance enhancement on the immediate problem instance. This real-time coupling necessitates rapid adaptation mechanisms that can process learning data and feedback signals and modify behavior within the temporal constraints of active task-solving. On the other hand, inter-test-time is characterized by its retrospective nature: learning algorithms operate on historical data, whether from curated datasets or accumulated behavioral trajectories, with optimization objectives oriented toward improving expected performance across the task distribution rather than maximizing success on any specific problem instance. This temporal decoupling enables more sophisticated learning procedures that can identify cross-task patterns, consolidate diverse experiences, and develop generalizable capabilities without the immediacy constraints of active task execution.

## 4.1 Intra-Test-Time Self-Evolution

In intra-test-time self-evolution, agents engage in self-improvement processes that are intrinsically coupled with solving the immediate task at hand. The distinguishing characteristic of this temporal phase is its synchronous nature: feedback signals are generated and processed during task execution, with optimization objectives specifically targeted at improving performance on the current problem instance rather than generalizing to future tasks. Here, we introduce how the three learning paradigms are realized in this temporal phase.

**In-Context Learning** Intra-test-time ICL methods leverage the model's context window as a dynamic memory system for immediate adaptation without parameter modification. These approaches typically employ self-reflective mechanisms where agents analyze their own performance, generate verbal critiques or insights, and maintain these reflections in episodic memory buffers to guide subsequent decisions within the same task context (Shinn et al., 2023; Madaan et al., 2023a). Some methods extend beyond simple reflection to include dynamic planning revision, where agents can modify their entire approach based on environmental feedback, switching between action execution and plan modification as needed. For instance, AdaPlanner (Sun et al., 2023) decomposes tasks into manageable sub-goals and predicts environmental feedback for each. During execution, its refiner component distinguishes between in-plan feedback (observations aligning with predictions) and out-of-plan feedback (deviating observations). For in-plan feedback, the refiner dynamically queries the LLM through a specialized `ask_LLM()` action to parse observations and extract pertinent information. For out-of-plan feedback, the refiner proactively revises the entire plan and resumes solving from an intermediate point, rather than restarting from scratch. This adaptive closed-loop framework eliminates the need for prior knowledge about feedback structures and enables more efficient decision-making. Similarly, TrustAgent (Hua et al., 2024) employs rule-based plan revision during execution, modifying its approach

based on language feedback to evolve toward safer planning strategies. These ICL methods demonstrate how test-time adaptation can achieve sophisticated behavioral modification without permanent model changes, maintaining flexibility while preserving the model's general capabilities.

**Supervised Fine-Tuning.** Intra-test-time SFT represents a paradigm shift where models perform immediate self-modification through learned meta-adaptation strategies. Self-adaptive language modeling (Zweiger et al., 2025) exemplifies this approach by generating "self-edits", which are meta-level instructions that can restructure information representations, specify optimization hyperparameters, or invoke tools for data augmentation and gradient computation. These self-edits trigger immediate supervised fine-tuning, resulting in persistent weight updates that adapt the model to the current task. The key innovation lies in the meta-learning phase, where reinforcement learning trains models to produce effective self-edits by using the downstream performance of the updated model as the reward signal, essentially teaching models how to teach themselves. Acikgoz et al. (2025) introduce a Test-Time Self-Improvement (TT-SI) framework that enables agents to adapt on-the-fly by first identifying uncertain test samples through self-awareness. For these challenging inputs, the agent then generates a single synthetic training example and performs a temporary, lightweight parameter update to improve its immediate performance before resetting its weights.

**Reinforcement Learning.** Intra-test-time RL enables models to develop new capabilities on-demand when encountering problems beyond their current competence. LADDER (Simonds & Yoshiyama, 2025) demonstrates this through its test-time reinforcement learning (TTRL) mechanism: upon identifying a particularly challenging problem, the system generates a focused set of related problem variants and conducts intensive, targeted reinforcement learning specifically for that problem class. This approach transforms insurmountable challenges into learning opportunities, allowing models to expand their problem-solving repertoire during deployment rather than failing or providing suboptimal solutions. The method represents a form of just-in-time skill acquisition, where computational resources are invested precisely when and where they are needed most.

### 4.2 Inter-Test-Time Self-Evolution

Inter-test-time self-evolution represents the predominant learning process in autonomous agents, wherein adaptation occurs following task execution rather than during it. In this temporal mode, agents complete a given task, extract feedback signals, including explicit rewards (Gao et al., 2024), gradients (Amari, 1993; Bottou, 2010), and performance metrics (Ge et al., 2023), and subsequently leverage this information to enhance their capabilities for future problem-solving. This retrospective learning process decouples task performance from capability improvement, allowing agents to consolidate experiences, identify patterns of success and failure, and systematically refine their behavioral policies without the computational constraints imposed by real-time task demands.

**In-Context Learning.** Inter-test-time in-context learning has emerged as a widely adopted approach for agent self-improvement. This paradigm leverages execution results and feedback from previous tasks as contextual information for future problem-solving. Wang et al. (Wang et al., 2024j) demonstrate this principle by inducing workflows from agent action histories and incorporating them into the context for subsequent tasks. The field of in-context reinforcement learning (ICRL) (Moeini et al., 2025; Laskin et al., 2022; Lee et al., 2023) extends this concept by maintaining histories of observations and actions within the agent's context window. These methods exploit the hypothesis that pre-trained neural networks can implement implicit reinforcement learning algorithms within their forward pass, processing contextual information to adapt behavior without parameter updates (Kirsch et al., 2023). A defining characteristic of ICRL is in-context improvement: the phenomenon whereby agent performance progressively enhances as task-relevant information accumulates in the context, enabling sophisticated adaptation through attention mechanisms rather than gradient-based learning.

**Supervised Fine-Tuning.** Inter-test-time SFT (Chen et al., 2025b) methods establish a paradigm of iterative self-improvement through synthetic data generation and self-evaluation. SELF (Lu et al., 2023) pioneered meta-cognitive training, where models first acquire self-feedback and self-refinement capabilities,

then iteratively generate responses to unlabeled instructions and enhance them through self-critique. STaR (Zelikman et al., 2022) and Quiet-STaR (Zelikman et al., 2024) focus on reasoning improvement through rationalization—models attempt problems, then generate explanations for correct answers they initially failed to solve, creating augmented training data that combines successful attempts with post-hoc reasoning. SiriuS (Zhao et al., 2025b) extends this to sequential problem-solving, maintaining repositories of correct solutions while augmenting failures through multi-stage refinement involving feedback incorporation, regeneration, and rephrasing. These methods share a core insight: models can bootstrap their own improvement by learning to evaluate and enhance their outputs, creating high-quality training signals from initially imperfect attempts without extensive human supervision. Recent frameworks such as ARIA (He et al., 2025a) further extend this paradigm by incorporating human-in-the-loop guidance into test-time adaptation, allowing agents to proactively identify knowledge gaps and request expert feedback.

**Reinforcement Learning.** Inter-test-time RL leverages unconstrained computational resources to optimize agents through extensive environmental interaction and sophisticated curriculum design. RAGEN (Wang et al., 2025q) and DYSTIL (Wang et al., 2025c) employ online reinforcement learning for multi-turn interactive tasks, continuously refining policies through on-policy learning in simulated dialogues. Learning Like Humans (Zhang et al., 2025a) introduces cognitive-inspired training with adaptive difficulty progression, combining on-policy exploration with off-policy efficiency and expert demonstrations to accelerate learning. Domain-specific applications demonstrate the versatility of inter-test-time RL: WebRL (Qi et al., 2024) develops web navigation agents through self-evolving curricula that automatically adjust task complexity based on performance, while DigiRL (Bai et al., 2024) enables device-control agents to master in-the-wild interactions through autonomous reinforcement learning. These approaches exploit the pre-deployment phase to engage in extensive trial-and-error learning, developing robust policies through thousands of interactions that would be impractical during real-time deployment.

## 5 How to Evolve

The pursuit of self-evolution lies at the heart of building advanced, autonomous, and increasingly general artificial intelligence. For large language models (LLMs) and their agentic extensions, the question of how to continually, autonomously, and efficiently evolve their capabilities has become a central challenge. Therefore, the third key aspect of a self-evolving agent is to instantiate an effective *evolving strategy* $f$, i.e., how to transform an agent system $\Pi = (\Gamma, \{\psi_i\}, \{C_i\}, \{\mathcal{W}_i\})$ to its new state $\Pi' = (\Gamma', \{\psi_i'\}, \{C_i'\}, \{\mathcal{W}_i'\})$. Unlike traditional approaches that rely on static datasets or one-time supervised fine-tuning, self-evolution emphasizes an ongoing process where models learn from real-world interactions, actively seek feedback, self-reflect, generate or curate new data, and adapt their strategies in response to dynamic environments. This continuous evolution is not merely a matter of scaling up data or computation; it requires the agent to acquire a spectrum of meta-capabilities, including self-correction, autonomous data generation, knowledge transfer, and multi-agent collaboration. As a result, the landscape of self-evolution has become increasingly rich and multi-faceted, with each methodological branch exploring different axes of feedback, learning paradigms, data sources, and evolutionary scales.

Over time, research on self-evolving agents has progressed through three major paradigms—reward-based, imitation-based, and population-based evolution—each emerging to address the limitations of the previous one. Reward-based methods first closed the feedback loop through explicit signals but suffered from brittleness and high cost. Imitation-based learning stabilized evolution by leveraging high-quality demonstrations, though sometimes at the expense of exploration. Population-based evolution then extended adaptation to collective scales, emphasizing diversity and emergent coordination. Together, these paradigms outline a coherent trajectory from individual self-improvement toward collective intelligence.

This chapter aims to systematically map and analyze the major families of self-evolution methods, providing a unified framework for understanding their principles, mechanisms, and interactions. We begin with **reward-based evolution**, which centers on the design of reward signals—ranging from natural language feedback and internal confidence metrics to external or implicit signals—to guide iterative self-improvement.

Table 3: Overview of Reward-based, Imitation/Demonstration, and Population-based Learning Methods for Self-Evolving Agents. This table categorizes key approaches based on the following criteria: (1) Feedback Type: the type of feedback used, including language-based rationales and numerical rewards. (2) Feedback Source: the origin of the feedback, either internal (model-generated) or external (provided externally). (3) Learning Method: the learning paradigm applied, such as in-context learning (ICL), supervised fine-tuning (SFT), reinforcement learning (RL), and evolutionary algorithms; (4) Updated Components: which parts of the model are updated, either full parameters or a subset of the model. (5) Update Timing: the stage during the agent's evolution when updates are applied, such as pre-training, pre-test, or test-time.

| Method | Feedback Type | Feedback Source | Learning Method | Updated Components | Update Timing |
|---|---|---|---|---|---|
| *Reward-based Evolution Methods* | | | | | |
| Reflexion(Shinn et al., 2023) | language | internal | ICL | context | test-time |
| AdaPlanner(Sun et al., 2023) | language | external + internal | ICL | context | test-time |
| AgentS2(Agashe et al., 2025) | language | external | ICL | context | test-time |
| SELF(Lu et al., 2023) | language | external + internal | SFT | full params | pre-test time + test-time |
| SELF-REFINE(Madaan et al., 2023a) | language | internal | ICL | context | test-time |
| SCoRe(Kumar et al., 2024) | numerical | external | RL | full params | pre-test time |
| PAG(Jiang et al., 2025b) | numerical | external | RL | full params | pre-test time |
| TextGrad(Yellamraju et al., 2024) | language | external | ICL | context | pre-test time / test-time |
| SRSI(Simonds et al., 2025) | language | internal | RL | full params | pre-test time |
| Self-Train LM(Shafayat et al., 2025) | numerical | internal | RL | full params | pre-test time |
| MM-UPT(Wei et al., 2025c) | numerical | internal | RL | full params | pre-test time |
| CoVo(Zhang et al., 2025j) | numerical | internal | RL | full params | pre-test time |
| SWE-agent(Du et al., 2025) | language | external | ICL | context | test-time |
| SICA(Robeyns et al., 2025a) | numerical | external | ICL | codebase(tools, workflows, prompts) | test-time |
| Feedback Friction(Jiang et al., 2025a) | language | external | ICL | context | test-time |
| USEagent(Applis et al., 2025) | language | external | ICL | context | test-time |
| DYSTIL(Wang et al., 2025c) | language + numerical | external + internal | SFT+RL | full params | pre-test time + test-time |
| OTC-PO(Wang et al., 2025h) | numerical | external | RL | full params | pre-test time |
| AUTORULE(Wang & Xiong, 2025) | language + numerical | external + internal | RL | full params | pre-test time |
| EGSR(Zhang et al., 2025a) | numerical | external | RL | full params | pre-test time |
| LADDER(Simonds & Yoshiyama, 2025) | numerical | external | RL | full params | pre-test time |
| RAGEN(Wang et al., 2025q) | numerical | external | RL | full params | test-time |
| SPIRAL(Liu et al., 2025d) | numerical | internal | RL | full params | pre-test time |
| ICRL Prompting(Song et al., 2025) | numerical | external + internal | RL | full params | test-time |
| MATH-SHEPHERD(Wang et al., 2023b) | numerical | external | RL | full params | pre-test time |
| AgentPRM(Choudhury, 2025) | numerical | external | SFT+RL | full params | pre-test time |
| Agent Q(Putta et al., 2024) | numerical | external | RL | full params | pre-test time |
| GiGPO(Feng et al., 2025a) | numerical | external | RL | full params | pre-test time |
| SPA-RL(Wang et al., 2025d) | numerical | external | RL | full params | pre-test time |
| Self-Instruct(Wang et al., 2022) | language | internal | SFT | full params | pre-test time |
| WizardLM(Xu et al., 2024a) | language | internal | SFT | full params | pre-test time |
| OS-Genesis(Sun et al., 2024b) | numerical | external | SFT | full params | pre-test time |
| UI-Genie(Xiao et al., 2025a) | numerical | external | SFT | partial params | pre-test time |
| GUI-R1(Luo et al., 2025b) | numerical | external | SFT+RL | full params | pre-test time |
| InfiGUI-R1(Liu et al., 2025e) | numerical | external | SFT+RL | full params | pre-test time |
| Voyager(Wang et al., 2023a) | language | external | ICL | context | test-time |
| SwiftSage(Lin et al., 2023) | language | external | ICL | context | test-time |
| AutoWebGLM(Lai et al., 2024) | language | external | SFT+RL | full params | pre-test time |
| DigiRL(Bai et al., 2024) | language | external | RL | partial params | pre-test time |
| WebRL(Qi et al., 2024) | language | external | SFT+RL | full params | pre-test time |
| Let's Verify Step-by-Step(Lightman et al., 2023) | language | external | SFT | full params | pre-test time |
| AlphaMath(Chen et al., 2024a) | numerical | external | SFT | full params | pre-test time |
| rStar-Math(Guan et al., 2025) | numerical | external | SFT | full params | pre-test time |
| DistRL(Wang et al., 2024g) | language | external | RL | full params | pre-test time + test-time |
| MobileGUI-RL(Shi et al., 2025b) | language | external | RL | full params | pre-test time |
| *Imitation and Demonstration Learning Methods* | | | | | |
| STaR(Zelikman et al., 2022) | language + numerical | internal | SFT | full params | pre-test time |
| V-STaR(Hosseini et al., 2024) | numerical | external + internal | SFT + RL | partial params | pre-test time |
| AdaSTaR(Koh et al., 2025) | numerical | internal | SFT | full params | pre-test time |
| STIC(Deng et al., 2024) | language | internal | RL + SFT | partial params | pre-test time |
| GENIXER(Zhao et al., 2024c) | language | external | SFT | full params | pre-training |
| SiriuS(Zhao et al., 2025b) | language + numerical | internal | SFT | full params | pre-test time |
| SOFT(Tang et al., 2025) | language | internal | SFT | not specified | pre-test time |
| RISE(Qu et al., 2024b) | language + numerical | internal + external | SFT | full params | pre-test time |
| IoE(Li et al., 2024b) | numerical | internal | / | / | test-time |
| *Population-based and Evolutionary Methods* | | | | | |
| DGM(Zhang et al., 2025h) | numerical | external | ICL | codebase (tools, workflows, prompts) | test-time |
| EvoMAC(Hu et al., 2024d) | language | external | ICL | team composition, workflow, prompts | test-time |
| SPIN(Chen et al., 2024f) | language | internal | RL | full params | pre-test time |
| GENOME(Zhang et al., 2025r) | numerical | external | Evolution Alg. | partial params | pre-test time |
| SPC(Chen et al., 2025c) | numerical | internal | SFT+RL | critic params | pre-test time + test-time |
| Puppeteer(Dang et al., 2025) | numerical | external | RL | planner policy | pre-test time / between tasks |
| MedAgentSim(Almansoori et al., 2025b) | language | external | ICL | context (knowledge base) | test-time |
| STL(Mendes & Ritter, 2025) | language + numerical | internal | SFT | value model | pre-test time |
| MDTeamGPT(Chen et al., 2025e) | language | external | ICL | context (knowledge base) | test-time |

Next, we examine **imitation and demonstration learning**, where agents learn by mimicking complete, high-quality behavioral exemplars (i.e., demonstrations). While traditionally sourced from human experts, in the context of self-evolving agents, these exemplars are often generated by the agent itself or by other agents. This paradigm is particularly powerful when demonstrations are abundant or can be autonomously synthesized, and it has driven significant progress in both reasoning and multimodal domains.

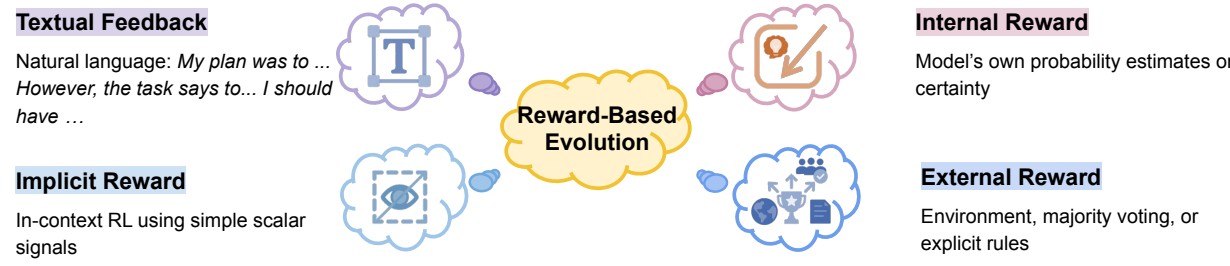

**Textual Feedback**

Natural language: *My plan was to ... However, the task says to... I should have …*

**Implicit Reward**

In-context RL using simple scalar signals

**Reward-Based Evolution**

**Internal Reward**

Model's own probability estimates or certainty

**External Reward**

Environment, majority voting, or explicit rules

Figure 6: Overview of reward-based self-evolution strategies, categorized into textual, implicit, internal, and external rewards, each associated with distinct feedback sources and mechanisms.

Finally, we introduce **population-based and evolutionary methods**, which draw inspiration from biological evolution and collective intelligence. These approaches maintain populations of agent variants or collaborating agents, leveraging mechanisms such as selection, mutation, crossover, and competition to explore the solution space in parallel, foster diversity, and enable the emergence of novel strategies or architectural innovations.

### 5.1 Reward-based Self-Evolution

The capacity for self-improvement is a cornerstone of advanced intelligence. In the context of Large Language Models (LLMs), this manifests as a dynamic process of reward-driven evolution, where models iteratively learn from their own outputs and interactions to refine their capabilities. The design of the reward signal, which serves as the guiding feedback, is crucial; it determines the nature, efficiency, and effectiveness of the learning process. In this section, we systematically review the main methodologies for reward design, categorized by the nature of the feedback: textual feedback, internal confidence, external rewards, and implicit rewards.

**Textual Feedback**   Textual Feedback leverages the native modality of LLMs—natural language—to provide detailed, interpretable instructions for refinement. Unlike scalar rewards, textual feedback encapsulates nuanced critiques and actionable suggestions. Recent frameworks such as Reflexion (Shinn et al., 2023), AdaPlanner (Sun et al., 2023), AgentS2 (Agashe et al., 2025), SELF (Lu et al., 2023), Self-Refine (Madaan et al., 2023a), SCoRe (Kumar et al., 2024), PAG (Jiang et al., 2025b), and TextGrad (Yellamraju et al., 2024) exemplify this direction. For instance, Reflexion proposes "verbal reinforcement learning," where agents reflect in natural language on their past trials, storing these reflections as episodic memory to guide future decisions. AdaPlanner enables closed-loop adaptive planning by allowing LLM agents to revise their plans based on both in-plan and out-of-plan feedback, while also mitigating hallucination via code-style prompts and leveraging skill discovery. Self-Refine and SELF further explore iterative self-feedback and self-correction, demonstrating that even state-of-the-art models can be improved via multi-turn, language-based self-critique, without additional supervised data or external reinforcement. Such frameworks highlight the power of language as a reward channel, enabling nuanced, flexible, and sample-efficient self-improvement.

**Internal Rewards**   Internal Confidence-based rewards move away from external signals and instead exploit internal metrics such as the model's probability estimates or certainty. This paradigm leverages the model's intrinsic understanding to guide improvement without relying on external supervision. Methods such as Confidence-Informed Self-Consistency (CISC) (Taubenfeld et al., 2025), Self-Ensemble (Xu et al., 2025b), Self-Rewarding Self-Improving (Simonds et al., 2025), scalable best-of-N selection via self-certainty (Kang et al., 2025), and Self-Rewarding Language Models (Yuan et al., 2025b) allow models to self-evaluate and calibrate their responses based on internal confidence metrics. For example, CISC weights reasoning paths by confidence scores to improve both accuracy and computational efficiency, effectively filtering high-quality solutions from multiple candidates. Self-Ensemble mitigates confidence distortion by dividing choices into smaller, more manageable groups and aggregating predictions to reduce overconfidence bias. Self-Rewarding

Language Models demonstrate that models can act as their own reward function, generating training data through self-instruction and self-evaluation cycles. These approaches can reduce reliance on human labels and external evaluators, enabling scalable and autonomous self-improvement loops that can operate continuously without human intervention. AgentEvolver (Zhai et al., 2025) proposes a comprehensive framework to improve agent training efficiency through three synergistic mechanisms: self-questioning for autonomous task generation, self-navigating for experience-guided exploration, and self-attributing for fine-grained credit assignment. In particular, its self-attributing mechanism uses an LLM's reasoning to retrospectively assign step-wise rewards that are dense and semantically grounded for policy optimization.

**External Rewards**  External Rewards are derived from sources outside the model, such as the environment, majority voting, or explicit rules. Majority voting (Shafayat et al., 2025; Wei et al., 2025c; Zhang et al., 2025j) uses consensus among multiple model outputs as a proxy for correctness, providing a self-generated but grounded reward signal. Environment feedback, including tool-based signals, is central to agentic LLM research (e.g., SWE-Dev (Du et al., 2025), SICA (Robeyns et al., 2025a), Feedback Friction (Jiang et al., 2025a), USEagent (Applis et al., 2025), DYSTIL (Wang et al., 2025c)), where agents learn through direct interaction with real-world environments and tools. Rule-based rewards (Wang et al., 2025h; Wang & Xiong, 2025; Zhang et al., 2025a; Simonds & Yoshiyama, 2025; Wang et al., 2025q; Liu et al., 2025d) use explicit constraints or logical rules as verifiable signals, particularly effective in the domains of mathematical reasoning, game play, and structured problem solving. These methods offer objective, reliable supervision but may require significant engineering or be limited in expressiveness.

**Implicit Rewards**  Implicit Reward frameworks hypothesize that LLMs can learn from feedback signals even when not explicitly labeled as rewards. For instance, "Reward Is Enough" (Song et al., 2025) demonstrates that LLMs can perform in-context reinforcement learning using simple scalar signals embedded in the context window, improving their responses over rounds without explicit RL fine-tuning or supervision. This reveals an inherent capacity for models to interpret and learn from implicit feedback cues present in their input context. Recent work has expanded this concept by showing that LLMs inherently encode reward-like signals through their standard training objectives. Endogenous reward (Li et al., 2025e) reveal that standard next-token prediction implicitly learns a generalist reward function, which can be extracted from model logits without additional training. Moreover, ImPlicit Self-ImprovemenT (PIT) framework (Wang et al., 2024i) implicitly learns the improvement goal from human preference data without extra human efforts by maximizing the quality gap of the response conditioned on a reference response. Unlike rule-based or environment-derived external rewards, implicit reward methods offer unique advantages by discovering and utilizing reward signals that are inherently present in language modeling.

In summary, reward-based evolution provides explicit optimization and strong autonomy but remains sensitive to reward design, often trading stability and safety for adaptability and openness.

## 5.2   Imitation and Demonstration Learning

Imitation and demonstration learning traditionally involves an agent that learns to mimic the behavior of an expert (typically a human) from a set of demonstrations. In the context of self-evolving agents, this paradigm is adapted and generalized, which is the focus of our survey. Here, the role of the "expert" is not necessarily a fixed, external entity (e.g., human) but rather any source of high-quality demonstration. In self-evolving agents, these "expert exemplars" are typically generated by the agent itself (e.g., a past successful trajectory), by other more capable agents, or synthesized from environmental interactions.

The key distinction between imitation learning and reward-based methods lies in the nature of the feedback. Imitation learning is *prescriptive* and *exemplar-based*: the agent is provided with a complete, successful guide (e.g., a full reasoning trace) and learns to reproduce this behavior. In contrast, reward-based methods are *evaluative* and *signal-based*: the agent explores on its own and receives a scalar or textual critique, forcing it to infer the path to improvement through trial-and-error and credit assignment.

Furthermore, the evolutionary mechanism differs fundamentally from population-based methods that will be introduced later. While both imitation and reward-based learning typically focus on optimizing a *single agent's improvement* through iterative refinement, population-based methods evolve a *collection of agents*

in parallel. Their progress typically comes from selection pressure across a diverse gene pool, rather than the direct knowledge transfer from an exemplar to an individual. Therefore, imitation learning occupies a unique niche: it relies on the availability of high-quality solutions to directly guide and accelerate the evolution of an individual agent, making it exceptionally powerful when such demonstrations can be reliably and autonomously generated.

### 5.2.1 Self-Generated Demonstration Learning

Self-generated demonstration learning involves agents creating their own training data through iterative refinement processes, where the models learn to improve by generating and selecting high-quality examples from their own outputs.

**Bootstrapping Reasoning Capabilities.** Zelikman et al. (2022) introduces the foundational framework for self-generated demonstration learning, enabling language models to bootstrap their reasoning capabilities through iterative self-training. This process involves generating reasoning chains for problems, fine-tuning on correct solutions, and repeating this cycle to progressively improve performance without the need for ground-truth reasoning paths. Building on this framework, recent advancements have refined the bootstrapping process through more sophisticated training strategies. For instance, Hosseini et al. (2024) proposes a verifier-guided self-training approach, where separate verifier models assess the quality of generated reasoning chains before they are incorporated into the training data, enhancing the reliability of self-improvement. Additionally, Koh et al. (2025) introduces adaptive data sampling strategies that dynamically adjust the composition of training data based on model performance across various reasoning tasks, thereby mitigating overfitting to specific problem types. The "Explore to Evolve" paradigm (Wang et al., 2025l) extends this concept to deep research web agents by proposing an automated pipeline for generating complex, verifiable training data. The framework directs an agent to first perform proactive online exploration to gather grounded information from the live web, and then to evolve a sophisticated aggregation logic to synthesize question-answer pairs that require both information-seeking and deep reasoning.

**Multimodal Self-Training.** Extending self-training to multimodal domains presents unique challenges in generating high-quality demonstrations that span both visual and textual modalities. Deng et al. (2024) demonstrates how vision-language models can improve iteratively by training on their own generated image descriptions and visual reasoning chains. The approach leverages the model's existing visual understanding to generate detailed image descriptions, which are subsequently used to fine-tune the model's visual perception in a bootstrapping manner. Zhao et al. (2024c) builds on this concept by empowering multimodal large language models to serve as powerful data generators, producing diverse training examples across different modalities and tasks through advanced prompt engineering and quality filtering mechanisms.

### 5.2.2 Cross-Agent Demonstration Learning

Cross-agent demonstration learning involves agents learning from demonstrations provided by other agents, either within the same system or from external sources, enabling knowledge transfer and collaborative improvement.

**Multi-Agent Bootstrapped Reasoning.** Zhao et al. (2025b) presents a framework for multi-agent systems to learn from each other's successful demonstrations through bootstrapped reasoning. The system maintains an experience library containing successful interaction trajectories generated by different agents, facilitating efficient knowledge sharing and collaborative improvement. Each agent can leverage the collective experience of the entire system, thereby accelerating the learning process and enabling the discovery of diverse solution strategies. This framework illustrates how agents can specialize in different aspects of complex tasks while benefiting from the accumulated knowledge of the entire system.

**Domain-Specific Demonstration Learning.** Domain-specific applications of demonstration learning have proven especially effective in specialized fields where expert knowledge can be effectively transferred through demonstrations. In recommendation systems, techniques such as self-optimized fine-tuning (Tang et al., 2025) enable LLM-based recommender systems to learn from their own successful recommendation patterns, creating a feedback loop that enhances personalization over time. The system generates high-

quality recommendation demonstrations from successful user interactions and uses these to fine-tune the underlying language model, ultimately leading to more accurate and personalized recommendations.

### 5.2.3 Hybrid Demonstration Learning

Hybrid demonstration learning combines both self-generated and external demonstrations to create more robust and diverse training regimens that leverage the strengths of each approach.

**Recursive Self-Improvement.** Qu et al. (2024b) demonstrates how agents can be trained to systematically improve their behavior through structured self-reflection and demonstration generation. This approach enables language model agents to introspect on their reasoning processes, identify areas for improvement, and generate corrective demonstrations to address these weaknesses. This recursive process establishes a continuous improvement loop, where agents become increasingly skilled at self-diagnosis and self-correction, leading to more robust and adaptable behavior.

**Confidence-Guided Demonstration Selection.** Recent developments have focused on more sophisticated mechanisms for selecting high-quality demonstrations from both self-generated and external sources. Confidence-based approaches (Li et al., 2024b) utilize the model's uncertainty estimates to determine which demonstrations are most likely to contribute positively to learning, filtering out potentially detrimental or low-quality examples. This method addresses a critical challenge in demonstration learning: poor-quality demonstrations can degrade performance. By ensuring that only high-confidence, high-quality examples are used for training, this approach helps to maintain the integrity of the learning process.

The effectiveness of imitation and demonstration learning approaches is highly dependent on the quality and diversity of the available demonstrations. While these methods can yield impressive results when high-quality exemplars are present, they face challenges in domains where good demonstrations are scarce or where the optimal behavior is not well-represented in the available data. Future research directions include developing more sophisticated demonstration selection and generation strategies, improving the robustness of learning from imperfect demonstrations, and creating better mechanisms for combining demonstrations from multiple sources.

Overall, imitation-based evolution stabilizes learning through high-quality exemplars but often trades exploration and generalization for reliability and sample efficiency.

### 5.3 Population-based and Evolutionary Methods

Population-based and evolutionary methods are a paradigm with a long history in improving agent behavior that complements modern learning-based techniques. This approach, drawing inspiration from biological evolution, has deep roots in AI. The concept was formalized into a practical computational tool by John Holland, whose seminal work on the Genetic Algorithm (GA) established the core operators of selection, crossover, and mutation for refining a population of solutions (Holland, 1976). Building on this, John Koza pioneered Genetic Programming (GP), a powerful extension that directly evolves executable programs or symbolic expressions, which makes it appropriate for generating agent logic. This paradigm's power was demonstrated by automatically synthesizing novel, human-competitive results, such as patented analog electrical circuits (Koza, 2010). This success extended into diverse domains, from evolving competitive agents for strategic games like backgammon and chess (Hauptman & Sipper, 2007; Azaria & Sipper, 2005) to discovering complex, interpretable policies in agent-based simulations, such as evolving dynamic taxation rules that outperformed static, human-designed strategies (Garuccio, 2016).

This paradigm led to landmark achievements in evolving agent systems. For example, the classic "Evolved Synthetic Creatures" co-evolved agent morphology and neural controllers in a simulated 3D world, leading to the discovery of a wide variety of novel and effective locomotion strategies (Sims, 1994). Later, the influential NEAT algorithm addressed a critical challenge by demonstrating how to evolve not just the weights but the entire topology of a neural network (Stanley & Miikkulainen, 2002). This enabled the autonomous discovery of complex agent "brains" from simple initial structures, a principle that has had an enduring impact on neuroevolution. These seminal works illustrated that evolution could construct both an agent's physical form and its complex control systems, establishing a powerful alternative to manual design.

Building on this foundation, these methods represent a different paradigm for agent evolution compared to the reward-based and imitation-based approaches discussed in previous sections. While reward-based methods typically optimize individual agents through iterative reward signals and imitation learning relies on learning from demonstrations, population-based methods maintain multiple agent variants simultaneously. This allows for parallel exploration of the solution space and the emergence of diverse capabilities through mechanisms such as selection, mutation, crossover, and competitive interaction (Zhang et al., 2025r). By leveraging parallel search and genetic variation, these methods enable broader search coverage and the discovery of novel solutions that might be missed by gradient-based optimization. This approach is particularly valuable when the solution space is complex, multimodal, or when the optimal strategy requires fundamental architectural changes rather than parameter fine-tuning.

### 5.3.1 Single Agent Evolution

Single-agent evolutionary approaches focus on evolving individual agents through population-based mechanisms, where multiple variants of an agent compete and evolve over time. These methods can be broadly categorized into two main paradigms: learning from evolution and self-play from multiple rollouts.

**Learning from Evolution.** This paradigm draws directly from biological evolution, maintaining populations of agent variants and applying evolutionary operators to discover improved capabilities. The Darwin Gödel Machine (DGM) (Zhang et al., 2025h) exemplifies this approach through open-ended evolution of self-improving agents that maintain an archive of all historical versions, enabling branching from any past "species" rather than linear optimization. The system achieves self-referential improvement by allowing agents to directly modify their own Python codebase, with evolution driven by empirical performance on coding benchmarks and parent selection balancing performance scores with novelty rewards for diverse exploration. Recent work has further explored using LLMs themselves to implement core evolutionary operators. For instance, LLM_GP (Hemberg et al., 2024) uses the LLM to perform mutation, crossover, and selection directly on programs represented as text, leveraging the model's innate knowledge of code to inform the evolutionary search. Similarly, open-source frameworks like CodeEvolve (Assumpção et al., 2025) have demonstrated that evolutionary coding agents can achieve state-of-the-art results on mathematical benchmarks, sometimes outperforming proprietary systems like AlphaEvolve (Novikov et al., 2025) by using an island-based genetic algorithm and inspiration-based crossover. This principle is also explored in Self-Referential Graph Hyper-Networks (Pedersen et al., 2025), where networks learn to generate their own weight mutations, allowing the rate of evolution itself to become selectable and adaptable.

Beyond evolving code and architecture, this paradigm extends to evolving the model's parameters and internal logic. The Nature-Inspired Population-Based Evolution (GENOME) framework (Zhang et al., 2025r) directly applies genetic algorithms to language model parameter evolution, maintaining populations and using crossover, mutation, and selection operators on model weights. GENOME+ (Zhang et al., 2025r) extends this with particle swarm optimization concepts, adding inheritance mechanisms and ensemble methods that demonstrate gradient-free evolutionary optimization can effectively improve model capabilities through parameter space exploration. EvoLLM-JP (Akiba et al., 2025) takes this further by using evolutionary algorithms to optimally merge multiple foundation models into a single, specialized model with superior performance. Furthermore, some frameworks create a tightly integrated feedback loop where evolution helps model fine-tuning. SOAR (Pourcel et al., 2025), for example, alternates between an evolutionary search phase to generate candidate programs and a "hindsight learning" phase that uses all attempts (both successful and failed) to generate a rich dataset for fine-tuning the agent model, creating a virtuous cycle of self-improvement.

**Self-Play.** Self-play is a paradigm where agents improve through iterative interaction with versions of themselves, creating a dynamic and self-sustaining learning process. Its principles were famously demonstrated by systems like AlphaZero (Silver et al., 2017), which achieved superior performance in complex games by learning entirely without human data. The core mechanism is co-evolutionary learning: as an agent improves, its opponents (past or concurrent versions of itself) also become stronger, generating a perpetual and adaptive curriculum of increasing difficulty. This avoids the stagnation that can occur when training against a fixed environment and enables the discovery of novel, emergent strategies.

This powerful principle has been adapted for LLMs and LLM Agents, enabling them to bootstrap their capabilities from zero or minimal external data. A prominent approach involves a single model or two model instances adopting distinct, co-evolving roles. For instance, Absolute Zero (Zhao et al., 2025a) and R-Zero (Huang et al., 2025b) employ a "challenger" or "proposer" agent that generates problems at the frontier of a "solver" agent's capabilities. A more complex multi-agent dynamic is seen in Socratic-Zero (Wang et al., 2025m), where a Solver co-evolves with a powerful Teacher that creates challenges and a Generator that distills the Teacher's strategy for scalable curriculum creation. To address the instability of purely autonomous systems, R-Few (Yu et al., 2025b) introduces a guided approach where the challenger is grounded by a small set of human examples to prevent concept drift and diversity collapse. The system improves through a closed loop where the solver is rewarded for correctness (often verified by execution) and the challenger is rewarded for posing difficult yet solvable problems, thus driving continuous improvement without external labels. Similarly, Self-Challenging Language Model Agents (Zhou et al., 2025e) establishes a framework where an agent alternates between generating and solving complex, multi-step coding tasks, using successful trajectories to fine-tune itself. The paradigm also extends to more specialized, collaborative roles, as seen in the Sol-Ver framework (Lin et al., 2025b), where an LLM co-evolves its ability to both generate code (solver) and create corresponding unit tests (verifier). Likewise, SPELL (Yang et al., 2025e) applies this principle to long-context reasoning, with a single model cyclically adopting questioner, responder, and verifier roles to provide reliable reward signals in a domain where programmatic verification is difficult.

Across these approaches, improvement is driven by self-generated learning signals derived from the agent's own trajectories. Self-Play Fine-Tuning (SPIN) (Chen et al., 2024f) establishes a foundational approach where current models compete against previous versions, creating evolutionary pressure for improvement. SPC (Chen et al., 2025c) advances this with a more sophisticated adversarial co-evolution, featuring a "sneaky generator" that creates deceptive errors and a "step critic" that learns to detect them. STL (Mendes & Ritter, 2025) demonstrates self-teaching through iterative lookahead search, where value models generate training data from their own exploratory rollouts. Recent work, such as EvoTest (He et al., 2025b), extends these ideas by introducing a gradient-free, evolutionary framework that revises an agent's prompt, memory, and tools between episodes.

Self-play is distinguished by a unique self-improvement mechanism: an agent learns through direct interaction with variations of itself. This typically manifests in two ways: a model competes against its own past versions to drive iterative refinement (as in SPIN), or a single model adopts distinct, interacting roles, such as a "challenger" generating novel problems for a "solver" (as in Absolute Zero). This principle of learning from *dynamic interaction* is different from imitation-based bootstrapping. While methods like STaR (Zelikman et al., 2022) also learn from an agent's own outputs, they do so by filtering and training on static, successful trajectories. Self-play, in contrast, generates its learning signal from the process of the game-like interaction itself, learning from relative success even when no perfect exemplar exists. This focus on a single agent's lineage also sets it apart from broader population-based methods: instead of evolving a large, diverse population, self-play creates a highly focused evolutionary pressure between a minimal set of policies derived from the same agent.

### 5.3.2 Multi-Agent Evolution

Multi-agent evolutionary methods extend population-based approaches to evolving entire teams or networks of agents, focusing on optimizing collective behavior, coordination strategies, and collaborative architectures. These approaches can be categorized into two main paradigms based on their evolution mechanisms: System Architecture Evolution and Knowledge-Based Evolution.

**System Architecture Evolution.** This paradigm focuses on evolving the structural and coordination aspects of multi-agent systems, including team composition, orchestration strategies, and workflow optimization. EvoMAC (Hu et al., 2024d) introduces a framework that mimics neural network training for multi-agent systems, implementing "textual backpropagation" where compilation errors and test failures serve as loss signals to drive iterative modifications of agent team composition and individual prompts. A specialized "updating team" analyzes textual feedback to identify problematic agents and generate modification instructions, effectively implementing gradient-based optimization in the space of agent configurations rather than model parameters. The FELA framework (Wang et al., 2025e) applies this concept to a practical

industrial problem, using a multi-agent system with specialized "Idea," "Code," and "Critic" agents that collaboratively evolve to generate high-performing features from complex data, guided by principles from both reinforcement learning and genetic algorithms. Puppeteer (Dang et al., 2025) takes a different approach by focusing on coordination strategy evolution rather than team composition changes. The system employs a centralized orchestrator that evolves its decision policy through reinforcement learning, dynamically selecting which agents to activate at each step while balancing task performance with computational cost. This "puppeteer-puppet" paradigm demonstrates how architectural evolution can occur at the coordination level, discovering efficient collaboration patterns and emergent behaviors such as tighter coordination among core agents and sophisticated cyclic interaction patterns. Agent0 (Xia et al., 2025b) introduces a framework that evolves agents from zero data via a co-evolutionary loop between a curriculum agent and an executor agent. The curriculum agent is trained to generate frontier tasks that challenge the executor. Then, the improved tool-use capabilities of the executor in turn drive the creation of a more complex, tool-aware curriculum, establishing a virtuous cycle of self-improvement.

**Knowledge-Based Evolution.** This paradigm emphasizes evolving the collective knowledge and experience of multi-agent teams through memory accumulation and case-based learning, primarily operating through in-context learning or in-context-like adaptation rather than parameter updates. MDTeamGPT (Chen et al., 2025e) establishes the foundation for this approach through a dual knowledge base system, implementing CorrectKB for storing successful cases and ChainKB for capturing failure reflections, enabling the system to learn from both successes and mistakes through structured case retrieval and reasoning enhancement. Extending this medical consultation framework, MedAgentSim (Almansoori et al., 2025b) demonstrates how such knowledge-based evolution can be applied to real-world diagnostic scenarios, accumulating experience from patient interactions and using retrieval-augmented generation to improve consultation quality over time. PiFlow (Pu et al., 2025) applies this paradigm to scientific discovery, maintaining a trajectory of principle-outcome pairs and using them to steer hypothesis generation through information-theoretical optimization.

In summary, population-based and self-play evolution enhance diversity and open-ended discovery, yet typically incur higher computational cost and lower interpretability compared with single-agent paradigms.

## 5.4 Cross-cutting Evolutionary Dimensions

After outlining the three core evolutionary paradigms, we now analyze their cross-cutting dimensions—revealing how different design choices balance feedback type, data source, and learning stability.

Agent self-evolution is a multifaceted process characterized by a number of cross-cutting dimensions that shape how agents learn, adapt, and improve over time. Beyond any single learning algorithm or supervision signal, these dimensions define the core principles underlying the design and analysis of autonomous agents. In this section, we systematically compare the major families of self-evolution methods—reward-based, imitation/demonstration-based, and population-based—along several key axes, such as learning paradigm (online vs. offline), policy consistency (on-policy vs. off-policy), and reward granularity (process-based, outcome-based, or hybrid). We further highlight additional dimensions, including feedback types, data sources, sample efficiency, stability, and scalability, as summarized in Table 4. This comprehensive comparison provides a unified perspective for understanding the strengths, limitations, and design trade-offs inherent in different approaches to agent evolution.

### 5.4.1 Online and Offline Learning

Another fundamental dimension in the design of self-evolving agents is the learning paradigm, which can be broadly categorized as either offline or online. This distinction depends on whether the agent's evolutionary updates are performed on a static, pre-collected dataset of experiences (offline) or through continuous, direct interaction with a live environment (online).

**Offline Learning** In the offline learning paradigm, the learning phase is decoupled from live task execution. The offline process typically involves cycles of offline data generation, filtering, and model fine-tuning, focusing on building a powerful and generalist foundational model before deployment. A primary strategy in this

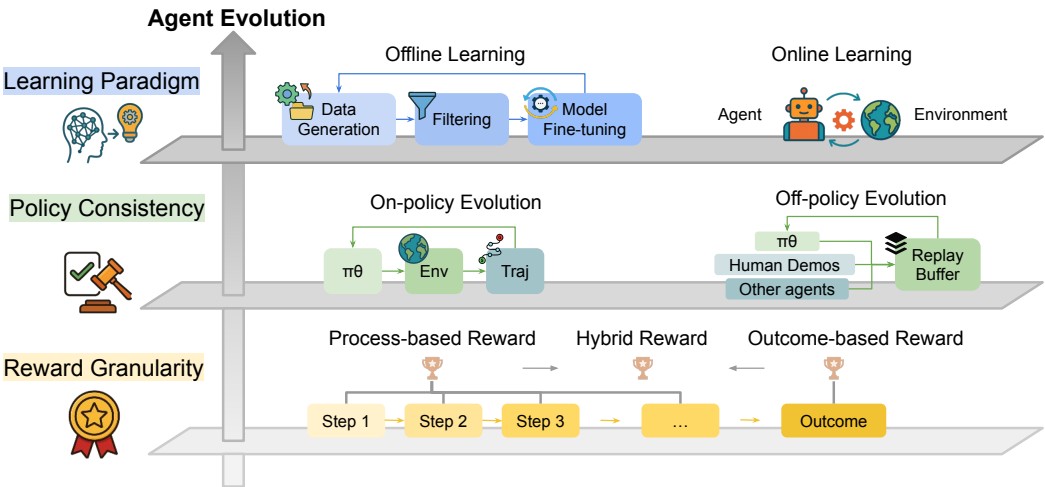

Figure 7: Illustration of the cross-cutting evolutionary dimensions underlying agent self-evolution. **Learning Paradigm:** in *offline learning*, data are pre-collected and used for filtering and fine-tuning, while in *online learning*, agents continuously interact with their environments for real-time adaptation. **Policy Consistency:** *on-policy evolution* updates policies based on the agent's own trajectories, whereas *off-policy evolution* relies on replay buffers, human demonstrations, or experiences from other agents. **Reward Granularity:** feedback can be *process-based* (step-level rewards), *outcome-based* (final-result rewards), or a *hybrid* of both. Together, these three orthogonal axes define how agents generate data, adapt their policies, and receive feedback across reward-based, imitation-based, and population-based evolution paradigms.

Table 4: Comparison of self-evolution method families along key dimensions.

| Dimension | Reward-based | Imitation/Demonstrat | Population-based |
|---|---|---|---|
| **Feedback Type** | Scalar reward, natural language, confidence, external signals | Demonstration trajectories, exemplars, rationales | Fitness scores, task success, competitive signals |
| **Data Source** | Self-generated, environment, external rules | Self-generated or other agents, humans | Population generations, multi-agent systems |
| **Reward Granularity** | Outcome/process/hybrid (flexible) | Usually outcome/process (via demo steps) | Often outcome-level, sometimes process via competition |
| **Online/Offline** | Both (reward learning, RL, DPO, SFT) | Typically offline, sometimes online demo mining | Online evolution or batch population updates |
| **On/Off-policy** | Both (DPO, Reflexion, GRPO) | Primarily off-policy, but online variants can be on-policy | Off-policy (population); self-play is on-policy |
| **Sample Efficiency** | Moderate (depends on reward sparsity) | High (if demo quality is high) | Usually low (needs many trials) |
| **Stability** | Sensitive to reward design | Sensitive to demo quality/diversity | Sensitive to population size/diversity |
| **Scalability** | Good with automation | Limited by demo collection | High but resource-intensive |

domain is LLM bootstrapping, where a model enhances its own capabilities using its self-generated content. For example, Self-Instruct(Wang et al., 2022) shows how a language model can bootstrap its own instruction-following ability by generating new instructions, paired with its own responses, creating a synthetic dataset

for fine-tuning. Building on this, WizardLM(Xu et al., 2024a) demonstrates how to progressively evolve the complexity of these self-generated instructions, pushing the model's capabilities on more challenging tasks. Although these methods primarily focus on broad capability expansion via synthetic heuristics, acting as a bootstrapping phase, they lay the necessary groundwork for closed-loop, experience-driven evolution defined in our framework. In the context of GUI and Web agents, offline learning often involves leveraging pre-collected high-quality trajectories for supervised fine-tuning (SFT). OS-Genesis(Sun et al., 2024b) introduced a reverse task synthesis method for automatic trajectory creation. Similarly, UI-Genie(Xiao et al., 2025a) employs a unified reward model for trajectory evaluation and a self-improving loop to generate high-quality trajectories iteratively. Both approaches focus on curating a rich SFT dataset to enhance the agent's capabilities to solve complex tasks. Beyond SFT, offline methods also incorporate reinforcement learning performed on a static dataset of agent-environment interactions. For example, GUI-R1(Luo et al., 2025b) and InfiGUI-R1(Liu et al., 2025e) utilize rule-based rewards and apply R1-style(Guo et al., 2025b) training on offline GUI datasets.

**Online Learning** In contrast, online learning enables an agent to learn and adapt continuously while it interacts with a live or simulated environment. Feedback from each action is used to update the agent's policy, plan, or knowledge base in real-time. This allows for greater adaptability to dynamic or unseen situations. Some agents evolve online not by updating their model weights, but by refining their plans and skill libraries on the fly. For example, Voyager(Wang et al., 2023a) presents an LLM-powered agent that learns to play Minecraft by continuously exploring, generating its own curriculum of tasks, and building a persistent skill library from direct experience. AdaPlanner(Sun et al., 2023) focuses on adapting its plan within a task; it generates an initial plan, receives feedback from the environment, and refines the plan online. Similarly, SwiftSage(Lin et al., 2023) operates with a fast-and-slow thinking process, where it can reflect on failures of its fast, intuitive mode and switch to a more deliberate, tool-using slow mode, adapting its strategy online based on task difficulty. Reinforcement Learning serves as a fundamental mechanism for online learning, enabling agents to learn from environmental reward signals. DigiRL(Bai et al., 2024) demonstrates how to train device-control agents in the wild using autonomous RL, while DistRL(Wang et al., 2024g) proposes an asynchronous distributed framework to make such on-device training feasible. MobileGUI-RL(Shi et al., 2025b) addresses the specific challenges of training GUI agents in online mobile environments by introducing a synthetic task generation pipeline combined with group relative policy optimization (GRPO) through trajectory-aware rewards.

### 5.4.2 On-policy and Off-policy Learning

While the previous section examined the timing of data collection and learning (online vs offline), this section focuses on the policy consistency aspect of agent evolution - specifically, whether agents learn from experiences generated by the same policy they are trying to improve (on-policy) or from experiences generated by different policies (off-policy). This distinction is crucial for understanding how agents utilize their experiential data and manage the trade-offs between learning stability and sample efficiency during the evolutionary process.

**On-policy Learning.** On-policy approaches require agents to learn exclusively from experiences generated by their current policy, ensuring policy consistency but often at the cost of sample efficiency. Reflexion (Shinn et al., 2023) exemplifies this approach through its iterative self-reflection mechanism. The agent generates responses using its current policy, receives feedback on failures, and immediately incorporates this feedback to update its reasoning process for the next iteration. GRPO (Shao et al., 2024b) and DAPO (Yu et al., 2025a) continue this path and show the effectiveness of multiple rollouts. The agent always learns from its current behavior, maintaining strict policy consistency. In agent settings, on-policy methods provide excellent learning stability and avoid distribution mismatch issues that plague off-policy methods. However, they suffer from low sample efficiency, as each policy update requires fresh data collection, making them computationally expensive for complex multi-step reasoning or tool use scenarios where generating high-quality trajectories is costly.

**Off-policy Learning.** Off-policy approaches allow agents to learn from experiences generated by different policies, including previous versions, other agents, or human demonstrations, significantly improving sample efficiency at the cost of potential distribution mismatch. Yuan et al. (2024c) demonstrates a sophisticated

off-policy approach where model $M_{t+1}$ learns from preference data generated by the previous version $M_t$. The system handles distribution shift through DPO's built-in KL divergence constraint with the reference policy, preventing the new policy from deviating too far from the data-generating policy. Yuan et al. (2023) showcases another powerful off-policy paradigm by learning from diverse response sources—including other models, humans, and different sampling strategies—through ranking-based supervision. The method elegantly sidesteps distribution shift by treating alignment as a ranking problem rather than requiring policy consistency. Zhao et al. (2025b) illustrates off-policy learning in multi-agent settings, where agents learn from an "experience library" containing successful interaction trajectories generated by previous policy versions, enabling efficient reuse of expensive multi-agent coordination data. In agent settings, off-policy methods excel in sample efficiency, allowing agents to leverage historical data, expert demonstrations, and cross-agent learning. They are particularly valuable for multi-step reasoning where successful trajectories are rare and expensive to generate, and for tool use scenarios where agents can learn from diverse execution examples without repeated environmental interaction. However, they face challenges with distribution shift, reward hacking (where agents exploit inconsistencies between training and deployment policies), and the need for careful regularization to maintain training stability.

### 5.4.3 Reward Granularity

Another critical choice in the reward design is its granularity, which determines at what level of detail the agent receives its learning signal. Reward granularity ranges from coarse-grained outcome-based rewards, which evaluate the overall task completion, to fine-grained process-based rewards that assess each step of the agent's trajectory. Current self-evolution frameworks adopt these varying levels of granularity to tailor feedback mechanisms according to task complexity and the desired learning outcomes.

**Outcome-based Reward** Outcome-based Reward is a feedback mechanism that evaluates an agent based on the successful completion of predefined tasks. This reward is determined solely by the final state of the agent's trajectory, regardless of the intermediate steps. A central challenge, particularly in dynamic environments like web or GUI navigation, is to effectively learn from both successful trajectories and the much more frequent failure trajectories. To address this, Direct Preference Optimization (DPO)(Rafailov et al., 2023) is designed to directly maximize the likelihood of preferred responses while minimizing the KL-divergence with the reference policy. Similarly, RRHF(Yuan et al., 2023) employs a ranking loss approach that aligns model probabilities of multiple responses with human preferences by ranking response probabilities without requiring auxiliary value models. Moreover, several works have developed specialized frameworks for agent self-evolution that are built upon outcome-based rewards. A straightforward approach is rejection sampling finetuning, as used in AutoWebGLM(Lai et al., 2024). This method employs a pre-designed reward model to evaluate trajectory outcomes, identify the successful trajectories, and update the model with this high-quality data. DigiRL(Bai et al., 2024) models the GUI navigation task as a Markov Decision Process (MDP) and obtains a final, sparse reward at the end of an episode using a VLM-based evaluator. WebRL(Qi et al., 2024) develops a robust outcome-supervised reward model (ORM) to address the feedback sparsity inherent in dynamic web environments. The ORM evaluates task success within a self-evolving curriculum framework, enabling agents to learn from unsuccessful attempts and progressively improve.

**Process-based Reward** In contrast to outcome-based rewards, which provide a single, delayed signal, the process-based reward paradigm offers more precise and granular feedback by evaluating each step in an agent's trajectory. Process-supervised reward models (PRMs) have been demonstrated to be significantly more reliable than outcome-supervised reward models (ORMs), particularly in domains requiring complex reasoning like solving math problems(Lightman et al., 2023). However, obtaining such fine-grained step-level feedback traditionally requires extensive human annotations, which are both time-consuming and expensive to scale. To address this annotation bottleneck, Math-Shepherd(Wang et al., 2023b) proposes an automatic process annotation framework that utilizes Monte Carlo Tree Search (MCTS) to gather step-wise supervision by assessing each step's potential to derive the correct final answer. Similarly, AlphaMath(Chen et al., 2024a) trains a value model to evaluate the step correctness in solution paths and updates both the policy and value model through exploration and exploitation within an MCTS framework. By leveraging process-based rewards, agents can improve their capabilities in a progressive, step-by-step manner. rStar-Math(Guan et al., 2025) and AgentPRM(Choudhury, 2025) both propose methods to iteratively evolve the policy and

the process reward model, generating progressively higher-quality reasoning paths without manual labels. Agent Q(Putta et al., 2024) integrates a step-wise verification mechanism into its MCTS process to collect high-quality trajectories, which are then used to iteratively refine the policy via DPO training.

**Hybrid Reward** The hybrid methods aim to provide more comprehensive learning signals by incorporating both the clarity of final task success (outcome-based) and the granular guidance of intermediate steps (process-based). These methods overcome the sparsity of outcome-only signals while grounding the agent's step-by-step reasoning in the ultimate task goal. For example, GiGPO(Feng et al., 2025a) addresses the instability of training long-horizon agents by introducing a dual-level reward mechanism. It provides an episode-level reward based on the final success of entire trajectories, while simultaneously assigning a localized, step-level reward for intermediate actions. This dual signal provides both a high-level directional goal and low-level corrective guidance. Similarly, SPA-RL(Wang et al., 2025d) proposes a reward decomposition method that bridges the gap between sparse outcome signals and dense process feedback. It attributes incremental progress to each step within multi-step trajectories based on the final task completion, effectively distributing the outcome-based reward across the process steps. This approach creates dense intermediate progress rewards that enhance reinforcement learning effectiveness while maintaining alignment with the ultimate task objectives.

### 5.5 Other Dimensions of Self-Evolution Methods

In addition to the core axes of learning paradigm, policy consistency, and reward granularity, Table 4 highlights several other important dimensions that differentiate self-evolution methods:

**Feedback Type.** The nature of feedback varies widely: reward-based methods leverage scalar rewards, natural language signals, or model confidence; imitation methods focus on demonstration trajectories and rationales; population-based methods use fitness scores or competitive signals. The feedback type fundamentally determines what information the agent uses to improve.

**Data Source.** Reward-based methods typically generate data through agent-environment interaction or engineered rules, while imitation learning often relies on human or expert-generated demonstrations. Population-based approaches draw from the collective experience of multiple agents or generations, enabling diverse exploration but requiring significant coordination.

**Sample Efficiency.** Imitation learning is generally the most sample-efficient, provided high-quality demonstrations are available, as agents can directly mimic expert behavior. Reward-based methods are moderately efficient, with efficiency highly sensitive to reward sparsity. Population-based evolution tends to be sample-inefficient, as it often requires evaluating a large number of agent variants through many trials.

**Stability.** Reward-based learning is sensitive to the quality and design of reward functions, risking reward hacking or unintended behaviors. Imitation learning depends heavily on the quality and diversity of demonstrations. Population-based methods are sensitive to population size and diversity, with small or homogeneous populations at risk of premature convergence.

**Scalability.** Scalability is determined by the feasibility of data or feedback collection and the ability to parallelize learning. Reward-based methods scale well when feedback is automated (e.g., via simulators). Imitation learning is often bottlenecked by the cost of collecting demonstrations. Population-based approaches can scale to large compute but are highly resource-intensive.

Together, these dimensions offer a more nuanced, multidimensional view of self-evolution strategies, guiding practitioners in selecting and designing agent learning pipelines that are best matched to the challenges of their specific domains.

## 6 Where to Evolve?

Self-evolving agents have facilitated advancements across a diverse array of domains and applications. Broadly, most of these applications can be systematically categorized into two groups: (1) general domain evolution, where agent systems evolve to expand their capabilities across a wide variety of tasks, mostly

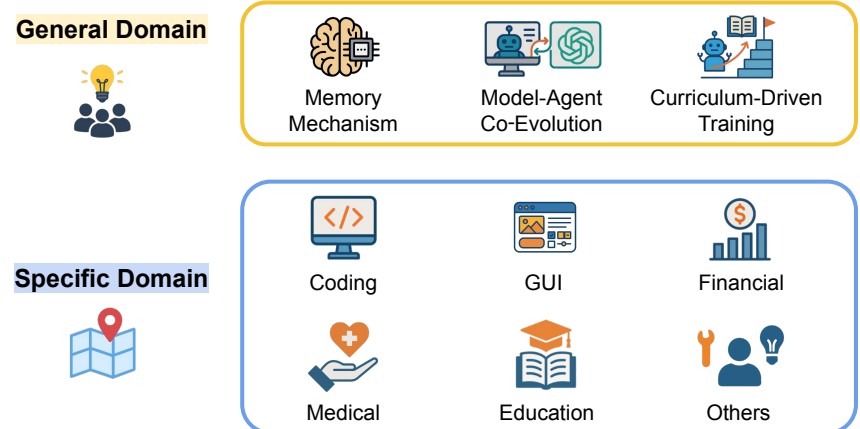

Figure 8: Categorization of where to evolve into two major types: General Domain Evolution, which focuses on broad capability enhancement across diverse tasks (e.g., memory mechanisms, co-evolution, curriculum training), and Specific Domain Evolution, which targets domain-specific expertise in areas such as coding, GUI, finance, medical, education, and others.

within the digital realm, and (2) specialized domain evolution, which evolves specifically to enhance their proficiency within particular task domains. In essence, evolution in general-purpose assistants focuses on transferring learned experience to a broader set of tasks, while evolution in specialized agents emphasizes deepening expertise within a specific domain.

## 6.1 General Domain Evolution

The first category, general domain evolution, refers to self-evolving agents designed for general-purpose applications, particularly as versatile digital assistants. These agents progressively enhance their capabilities to address a broad spectrum of user queries, especially in dynamic and diverse digital environments. Technically speaking, these general assistant agent enhance their abilities primarily via three mechanisms: memory optimization, curriculum-driven training, and model-agent co-evolution. These mechanisms collectively enable the agents to continuously adapt and effectively respond to increasingly complex user demands.

**Memory Mechanism.** The most common mechanism facilitating agent evolution is the memory mechanism, wherein agents summarize historical success/failure experiences (Wang et al., 2023a; Zhang et al., 2024f) into memory representations (Zhang et al., 2024g), anticipating that these distilled experiences will be beneficial when addressing previously unseen tasks. For instance, Mobile-Agent-E (Wang et al., 2025o) employs a long-term memory structure consisting of "Tips," which provide general guidelines, and "Shortcuts," representing reusable action sequences derived from past experiences. This self-evolutionary module supports the continuous enhancement of performance on complex smartphone tasks. Another typical example is MobileSteward (Liu et al., 2025f), which coordinates multiple app-specific Agents under a central Agent, with specialized modules for task scheduling, execution, and evaluation. It also incorporates a memory-based self-evolution mechanism that summarizes successful executions to improve future cross-app instruction handling. Meanwhile, Generative Agents (Park et al., 2023) store episodic memories of their experiences, synthesize higher-level reflections, and condition future planning on this self-reflection. In these examples, memory serves as the foundation that enables agents to internalize past experiences, abstract high-level patterns, and refine their future behavior.

**Model-Agent Co-Evolution.** Another line of work is to perform Model-Agent Co-evolution for LLM agents. UI-Genie (Xiao et al., 2025a) constructs a specialized image-text reward model that scores trajectories at both step and task levels. It jointly fine-tunes the agent and reward model using synthetic trajectories—generated by controlled corruption and hard-negative mining—across multiple generations. WebE-

volver (Fang et al., 2025b) introduces a co-evolving world model LLM that simulates web environments. It generates synthetic training data by predicting next observations and enables look-ahead reasoning during inference, which greatly improves real-web task success. Absolute Zero (Zhao et al., 2025a) co-evolves a reasoning agent and its internal self-reward model through reinforced self-play. By adversarially generating increasingly challenging reasoning problems and optimizing the agent using internal self-certainty as a reward signal, the framework simultaneously updates both the agent's policy and the self-rewarding mechanism. Together, these methods demonstrate the effectiveness of co-evolving agents and auxiliary models (e.g., reward or world models) to achieve more robust, generalizable, and scalable learning in LLM agentic systems.

**Curriculum-Driven Training.** Curriculum-driven training also serves as a critical mechanism for building a self-evolving general assistant. For example, WebRL (Qi et al., 2024) uses a self-evolving curriculum: when an agent fails, similar but manageable tasks are automatically generated. Coupled with a learned reward model and adaptive policy updates, this yields a success rate uplift on WebArena benchmarks. Voyager (Wang et al., 2023a) similarly leverages an automatic, bottom-up curriculum in Minecraft, where GPT-4 proposes appropriate next tasks based on agent progress, building a growing code-based skill library through iterative prompting and environmental feedback. These approaches highlight how curriculum learning enables agents to autonomously expand their capabilities through iterative task adaptation.

## 6.2 Specialized Domain Evolution

In addition to general digital agents, self-evolving agents have also been effectively applied within specialized domains, where their evolution is tailored to significantly enhance performance within narrower task sets.

**Coding.** The power of self-evolving agents extends directly to practical applications like coding, where their ability to autonomously adapt and improve offers a transformative approach to software development. SICA (Robeyns et al., 2025a) demonstrates that a self-improving coding agent can autonomously edit its own codebase and improve its performance on benchmark tasks. EvoMAC (Hu et al., 2024d) introduces a self-evolving paradigm on multi-agent collaboration networks, which automatically optimizes individual agent prompts and multi-agent workflows, significantly improving code generation performance by overcoming the limitations of manually designed systems. AgentCoder (Huang et al., 2024) also focuses on a multi-agent code generation framework that self-evolves through iterative refinement. A programmer agent continuously improves code based on feedback from a test executor agent, validated against independent test cases from a test designer, significantly boosting effectiveness and efficiency. Zhang et al. (Zhang et al., 2025b) enable LLM agents to continuously evolve by filtering high-quality answers, stratifying earned experiences by difficulty, and adaptively selecting demonstrations from self-generated data, leading to significant performance improvements and the construction of ML libraries. While these instances differ in their specific mechanisms—ranging from single-agent self-editing to complex multi-agent collaborative networks and experience-based learning—they commonly share the core principle of iterative self-improvement and autonomous adaptation to enhance coding capabilities. These advancements highlight how self-evolving agents can dramatically enhance coding efficiency and code quality by continuously learning and optimizing.

**Graphical User Interfaces (GUI).** Self-evolving GUI agents extend LLM capabilities from pure text reasoning to direct manipulation of desktop, web, and mobile interfaces, where they must cope with large discrete action spaces, heterogeneous layouts, and partial visual observability. Yuan *et al.* couple pixel-level vision with self-reinforcement, enabling the agent to iteratively refine click–type grounding accuracy without additional human labels (Yuan et al., 2025c). On real desktop software, the Navi agent from *WindowsAgentArena* replays and critiques its own failure trajectories, ultimately doubling its task-completion rate across 150 Windows challenges (Bonatti et al., 2024). For open-web automation, *WebVoyager* fuses screenshot features with chain-of-thought reflection; successive self-fine-tuning raises its end-to-end success on unseen sites from 30 % to 59 % (He et al., 2024), while ReAP adds episodic memories of past outcomes, recovering a further 29-percentage-point margin on previously failed queries (Azam et al., 2025). Beyond RL and memory, *AutoGUI* continuously mines functionality annotations from live interfaces to expand a reusable skill library each training cycle (Li et al., 2025a), and *MobileUse* deploys a hierarchical self-reflection stack that monitors, verifies, and revises smartphone actions in situ (Li et al., 2025c). Collectively, these

systems epitomize the full triad of self-evolution— what evolves (grounding modules, skill memories), when it evolves (offline consolidation vs. online reflection), and how it evolves (reinforcement learning, synthetic data, hierarchical monitoring)—charting a path toward universally competent interface agents.

**Financial.** The primary bottleneck in customizing agents for specialized domains like financial tasks lies in efficiently constructing and integrating a domain-specific knowledge base into the agent's learning process—a challenge that can be effectively mitigated by incorporating self-evolving mechanisms. QuantAgent (Wang et al., 2024d) proposed a two-layer framework that iteratively refines the agent's responses and automatically enhances its domain-specific knowledge base using feedback from simulated and real-world environments. This iterative process helps the agent progressively approximate optimal behavior, reduces reliance on costly human-curated datasets, and demonstrably improves its predictive accuracy and signal quality in trading tasks. TradingAgents (Xiao et al., 2024) incorporates dynamic processes such as reflection, reinforcement learning, and a feedback loop from real-world trading results, alongside collaborative debates, to continuously refine its strategies and enhance trading performance. These developments underscore the potential of self-evolving agents to revolutionize the financial domain by autonomously building domain expertise, adapting to dynamic market conditions, and continuously improving decision-making and trading performance.

**Medical.** Self-evolving agents have become a powerful paradigm in medical AI, where adaptability and the ability to evolve are essential for managing the complexity and ever-changing nature of real-world clinical practice. One of the most prominent applications is hospital-scale simulation. For example, Agent Hospital (Li et al., 2024a) creates closed environments with LLM-driven doctors, patients, and nurses, allowing the doctor agent to treat thousands of virtual cases. This process helps these agents autonomously refine and evolve their diagnostic strategies without manual labeling, ultimately achieving strong performance on USMLE-style exams. Similarly, MedAgentSim (Almansoori et al., 2025a) integrates an LLM doctor, patient, and tool agent. It records successful consultations as reusable trajectories and employs chain-of-thought reflection and consensus to drive self-evolution, improving success rates over successive interactions. Another example is EvoPatient (Du et al., 2024) places a doctor agent and a patient agent in continuous dialogue. With each generation, they update their memory with high-quality exchanges: the patient develops more realistic symptom narratives, while the doctor learns to ask sharper questions. Notably, this happens without explicit gradient updates or hand-crafted rewards. Reinforcement learning is also central to building adaptive medical agents. For instance, DoctorAgent-RL (Feng et al., 2025c) models consultations as a Markov decision process, using a reward function that scores diagnostic accuracy, coverage, and efficiency. This guides policy-gradient updates that help the agent ask more relevant questions and reach correct diagnoses faster than imitation-based approaches, thus achieving self-improvement. In addition, automated architecture-search approaches like *Learning to Be a Doctor* treat the workflow itself as an evolvable object, iteratively inserting specialist sub-agents or new reasoning hops to cover observed failure modes and improve multimodal diagnostic accuracy (Zhuang et al., 2025). Finally, beyond clinical decision-making, self-evolving agents have also been extended to biomedical discovery. OriGene (Zhang et al., 2025t) functions as a virtual disease biologist that evolves by iteratively refining its analytical process. It leverages human and experimental feedback to update core reasoning templates, adjust tool usage strategies, and refine analytical protocols. Similarly, STELLA (Jin et al., 2025) is a self-evolving biomedical research agent that improves over time by distilling successful reasoning workflows into reusable templates through its Template Library and expanding its Tool Ocean with external or newly assembled tools to meet emerging analytical needs.

**Education.** Self-evolving LLM agents have also found strong applications in the education domain. At the learner level, self-evolving agents like the personalized tutor PACE (Liu et al., 2025c) adjust their prompts based on detailed student profiles and continually refine their questioning during conversations. Meanwhile, an LLM-to-LLM self-play framework generates diverse tutor–student dialogues that further fine-tune the agent, allowing its teaching strategies to evolve both during and after interactions. Another example is MathVC (Yue et al., 2025), which employs symbolic persona profiles for virtual students and a meta-planner that orchestrates realistic problem-solving stages. This setup enables the agent's conversational process to evolve step by step toward correct solutions, closely mirroring how collaborative learning naturally unfolds. On the instructor side, self-evolving agent systems like the professional-development platform i-vip (Yang et al., 2025b) deploy a team of cooperating LLM agents—a coach, assessor, and feedback generator—that

critique and enhance each other's outputs in real time. These agents adapt their explanations based on teacher-learners' responses and continue to evolve by incorporating expert feedback after deployment, thereby refining their prompt strategies over time Similarly, EduPlanner (Zhang et al., 2025p) frames lesson-plan creation as an adversarial loop where a planner's draft is repeatedly reviewed and refined by evaluator and optimizer agents until it meets diverse educational goals. Similarly, SEFL (Zhang et al., 2025k) uses teacher–student self-play to generate large sets of homework–feedback examples, which then fine-tune a lightweight feedback model. This self-evolving process significantly improves the clarity and usefulness of the comments. Collectively, these examples illustrate how self-evolving LLM agents can dynamically adapt to both learners and instructors, driving more personalized, effective, and scalable educational experiences.

**Others.** Beyond the four major verticals discussed above, self-evolving agents demonstrate broader applicability, delivering superior adaptability and performance in specialized domains where conventional agents often fall short. For instance, Arxiv Copilot (Lin et al., 2024) learns and adapts by incorporating historical user interactions, including generated answers, research trends, and ideas, into its thought database, enhancing its ability to provide personalized and augmented academic assistance. In a very different context, Voyager (Wang et al., 2023a), an agent in the game Minecraft, excels at solving novel tasks from scratch in new worlds through a process of self-evolution. It continually refines its task goals via an automatic curriculum, expands its skill library, and enhances its actions using an iterative prompting mechanism without human intervention. Transitioning to domains that require explicit strategic planning, Agents-of-Change (Belle et al., 2025) autonomously refines prompts and rewrites code based on iterative performance analysis and strategic research, thereby helping agents overcome inherent limitations in long-term strategic planning and achieve consistently superior and more coherent gameplay in complex environments like Settlers of Catan. Lastly, in the realm of diplomacy, Richelieu (Zhao et al., 2024d) introduces AI diplomacy agents that can self-evolve through their self-play mechanism, which allows the agent to augment its memory by acquiring diverse experiences without human data, thereby enhancing its strategic planning, reflection, and overall performance in diplomacy activities. While these diverse examples operate in distinct environments—from academic research and virtual game worlds to strategic board games and complex diplomatic negotiations—they all share the fundamental characteristic of leveraging continuous learning, self-refinement, and autonomous adaptation to achieve increasingly sophisticated and effective performance within their respective domains. These diverse examples reinforce the versatility of self-evolving agents, showcasing their growing potential to excel in a wide range of complex, dynamic, and human-like tasks beyond traditional domains.

## 7 Evaluation of Self-evolving Agents

Evaluating self-evolving agents presents a unique set of challenges that extend beyond the traditional assessment of static AI systems. Unlike conventional agents typically evaluated on a fixed set of tasks at a single point in time, self-evolving agents are designed to continuously learn, adapt, and improve through ongoing interaction with dynamic environments. Consequently, their evaluation must capture not only immediate task success but also crucial aspects such as adaptation over time, knowledge accumulation and retention, long-term generalization, and the ability to transfer learned skills across sequential or novel tasks, all while mitigating catastrophic forgetting. This requires a fundamental shift from conventional "single-shot" *scoring* to a longitudinal, cost-aware trajectory view.

### 7.1 Evaluation Goals, Metrics, and Benchmark Coverage

To effectively evaluate self-evolving agents, we must move beyond traditional metrics and establish a comprehensive framework that captures their dynamic, adaptive, and long-term learning capabilities. A truly capable and desirable self-evolving agent must not only **learn and improve** but also **remember** past knowledge, **transfer** it to new situations, operate **sustainably**, and behave **responsibly**. Grounded in these critical requirements for continuous and robust AI, we categorize the key evaluation goals into five core dimensions: **Adaptivity**, **Retention**, **Generalization**, **Efficiency**, and **Safety**, as illustrated in Table 6. Each dimension addresses a vital aspect of an agent's self-evolutionary process and is assessed through its corresponding metrics and benchmark coverage analysis. While Table 7 provides a comprehensive catalog of

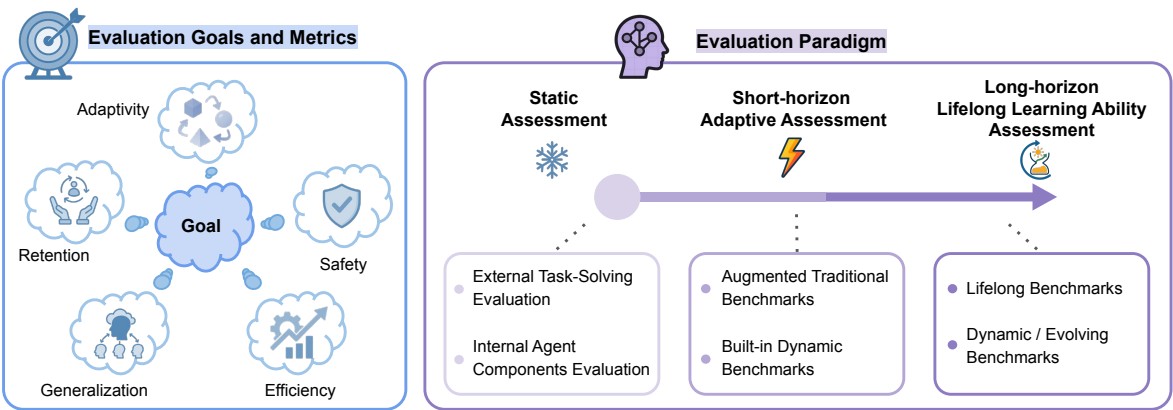

Figure 9: An overview of evaluation goals and paradigms for self-evolving agents. **Left: Evaluation Goals and Metrics.** We summarize five core objectives guiding agent evaluation: *adaptivity*, *retention*, *generalization*, *efficiency*, and *safety*, which reflect how agents should evolve and perform over time. **Right: Evaluation Paradigm.** Evaluation methods are organized along an increasing temporal scale: from *static assessment* of fixed capabilities (e.g., external task-solving and component-level evaluation), to *short-horizon adaptive assessment* capturing within-task adaptation (e.g., augmented or dynamic benchmarks), and finally to *long-horizon lifelong learning assessment* that examines continual improvement under evolving or lifelong benchmarks. Together, these axes link the *goals* of self-evolution with the *evaluation settings* used to measure them, forming a unified framework for assessing agent adaptivity and sustainability. See Table 6 for metric definitions, Table 10 for protocol summary.

evaluation resources, we synthesize these limitations to map coverage gaps across these five-goal framework and identify directions for trajectory-centric, cost-aware assessment at Table 8.

### 7.1.1 Adaptivity

**Goals & Metrics** Adaptivity serves as a foundational evaluation criterion for any self-evolving agent, measuring its ability to improve performance on in-domain tasks through experience. This dimension focuses on quantifying the learning curve and the extent of performance enhancement as an agent iterates and evolves within a specific domain. Rather than a static success rate, adaptivity is gauged over time, steps, or iterations. Typical metrics include the Success Rate by Iteration Steps (Hu et al., 2024c; Wang et al., 2024j; Zheng et al., 2025b), which tracks performance in downstream tasks as a function of the agent's interaction history.

**Coverage Gaps** Although adaptivity has the richest benchmark ecosystem—spanning code generation (SWE-bench (Jimenez et al., 2023), MLE-Bench (Chan et al., 2024)), web navigation (WebArena (Zhou et al., 2023), WebShop (Yao et al., 2022)), and general reasoning (GAIA (Mialon et al., 2023), Agent-Bench (Liu et al., 2023b) evaluations remain constrained by practical design choices. ScienceAgentBench limits tasks to a single programming language and imposes execution-time bounds, excluding computationally intensive scientific workflows (Chen et al., 2024e). MLE-Bench offers well-structured tasks but diverges from authentic research scenarios where problem formulation is itself part of the challenge (Chan et al., 2024). These simplifications permit controlled in-domain assessment but confine evaluation to narrow, static task distributions. Moreover, most benchmarks measure improvement under pre-specified learning protocols (e.g., fixed iteration budgets or predetermined replay schedules) rather than assessing whether agents autonomously discover effective adaptation strategies. AgentBench further reports weaknesses in sustained reasoning and strategic decision-making (Liu et al., 2023b), yet evaluation horizons remain limited to short-term improvement curves.

### 7.1.2  Retention

**Goals & Metrics**  Retention is a crucial criterion for evaluating the stability of a self-evolving agent's knowledge base. This dimension specifically focuses on the challenge of catastrophic forgetting, a common issue in lifelong learning where new knowledge acquisition erodes previously learned information, and knowledge retention within extended interactions. Two key metrics can be used to quantify this stability from different perspectives: Forgetting (FGT) and Backward Transfer (BWT) (Zheng et al., 2025c). Specifically, let $J_{i,t}$ denote the performance on task $i$ after completing $t$ tasks:

$$\text{FGT}_t = \frac{1}{t-1} \sum_{i=1}^{t-1} \Big[ \max_{j \in \{i,\dots,t\}} J_{i,j} - J_{i,t} \Big], \quad \text{BWT}_t = \frac{1}{t-1} \sum_{i=1}^{t-1} \big( J_{i,t} - J_{i,i} \big).$$

A positive BWT indicates that new learning positively benefits old tasks, signifying successful knowledge transfer and a more robust, stable learning process.

**Coverage Gaps**  Retention remains the most underserved dimension. Current memory mechanisms struggle significantly with dynamic state updates and maintaining consistency across extended interactions (Hu et al., 2025). Experience replay exhibits an inherent tension: while replay buffers can improve learning, scaling them beyond optimal thresholds triggers performance degradation through context overflow and resource exhaustion (Zheng et al., 2025b). Economic and computational constraints limit evaluation robustness, with some benchmarks conducting minimal repetitions that may not capture stochastic variation (Castillo-Bolado et al., 2024). Most critically, the overwhelming majority of existing benchmarks adopt episodic evaluation where agent state resets between tasks, fundamentally precluding measurement of knowledge accumulation or degradation—precisely the phenomena that distinguish self-evolving agents from static systems.

### 7.1.3  Generalization

**Goals & Metrics**  While Adaptivity and Retention focus on in-domain performance, Generalization is a pivotal measure of a self-evolving agent's ability to apply its accumulated knowledge to new, unseen domains or tasks. A truly intelligent agent should not only perform well within its familiar territory but also demonstrate a capacity for cross-domain generalization. This capability can be evaluated by assessing an agent's performance on a diverse set of tasks that span multiple task distributions and domains. Common approaches include computing aggregate performance metrics (e.g., mean success rates) across multi-domain test suites (Liu et al., 2023b; Sun et al., 2023), and conducting out-of-domain evaluations using held-out task distributions that simulate real-world novelty scenarios (Hu et al., 2024b; Peng et al., 2025).

**Coverage Gaps**  Generalization is typically evaluated through multi-domain test suites (AgentBench (Liu et al., 2023b), TheAgentCompany (Xu et al., 2024b) and held-out task distributions, measuring an agent's ability to transfer knowledge across domains. Current evaluations, however, rely on static snapshots: agents are tested once on diverse tasks without tracking whether cross-domain transfer degrades as they evolve over extended learning trajectories. Technological progress further reduces the discriminative power of older tasks, as modern agents benefit from algorithmic advances unavailable to earlier systems (Chan et al., 2024), while training data contamination complicates cross-domain assessment (Chan et al., 2024). No existing benchmark examines whether agents preserve generalization breadth as they specialize within domains, or whether knowledge acquired in one domain continues to transfer after hundreds of learning episodes.

### 7.1.4  Efficiency

**Goals & Metrics**  Efficiency quantifies the resourcefulness of a self-evolving agent during its learning and operation. As agents operate continuously and make decisions autonomously, it is essential to evaluate the cost and speed of their evolutionary process.

These metrics in Table 5 are particularly important for practical, real-world applications where resources like computation, memory, time, and human effort are finite:

Table 5: **Refined cost taxonomy and units for self-evolving agents**, with Real-world example (Fan et al., 2025): On SWE-bench (Jimenez et al., 2024), SWE-Agent (Yang et al., 2024) + Qwen3-32B (440K tokens, 35 calls, 28% success) vs. GPT-4o-mini (8.1M tokens, 181 calls, 10% success)—model-scaffold synergy critical for efficiency.

| Dimension | Symbol | Typical Unit | Example Measurement | Worked Example |
|---|---|---|---|---|
| **Token usage** | $C_{\text{token}}$ | #tokens | Prompt + completion tokens per episode/step | 440K vs. 8.1M tokens per task; 18× difference |
| **Step count / latency** | $C_{\text{step}}$ | #turns / rounds | Reasoning/interaction turns | 35 vs. 181 API calls per task; Quality vs. quantity trade-off |
| **Wall-clock time** | $C_{\text{time}}$ | #seconds | End-to-end elapsed runtime incl. I/O | 12 vs. 200+ turns per task; Simple vs. complex bugs |
| **Tool/API calls** | $C_{\text{tool}}$ | #calls | External function/environment calls | 15 vs. 38 calls (AutoCodeRover); High-quality reasoning reduces iterations |
| **Memory growth** | $C_{\text{mem}}$ | #tokens / MB | Persistent memory; context window expansion | Linear growth per call; Token snowball effect amplifies invalid context |
| **Human oversight** | $C_{\text{human}}$ | #hours / tokens | Review, labeling, red-team; guidance tokens | Failed attempts: 4× cost vs. successful ones; Expensive failures pattern |

- **Token consumption** (Hu et al., 2024a): Total tokens used in reasoning, generation, and memory operations

- **Time consumption** (Lu et al., 2024b): Wall-clock time required to complete tasks or reach performance thresholds

- **Step count** (Wang et al., 2023a): Number of interaction rounds needed for task completion

- **Tool calls**: Number of external API/environment invocations, can be combined with performance gains to assess *Tool Productivity* (TP) (Wang et al., 2025h), formulated as:

$$\text{TP}_{\$} = \frac{\Delta\text{score}}{\sum_i \text{cost}(\text{tool}_i)}.$$

- **Memory growth**: Expansion of persistent memory and context window over the agent's operational duration

- **Human oversight**: Time and effort required for human intervention, including review, labeling, and corrective guidance

To relate efficiency to performance gains, we report *Cost-per-Gain* (CPG) as:

$$\text{CPG}_t = \frac{\text{Total Cost}_t}{\text{Performance Gain}_t + \epsilon},$$

where cost can be measured in tokens, time, memory, human effort, or a normalized composite (e.g., monetary cost), and performance gain is the improvement over baseline at horizon $t$. Lower CPG indicates more efficient learning.

**Coverage Gaps** Efficiency suffers from sparse and inconsistent reporting of evolution costs. While benchmark papers occasionally document aggregate resource consumption during evaluation (Chan et al., 2024), they rarely decompose costs into evolution-specific components: tokens consumed during self-reflection or experience replay, wall-clock time spent on architecture search or memory updates, tool invocations triggered by autonomous exploration. Cost constraints in human baseline collection (Xu et al., 2024b) reflect broader accessibility challenges but do not illuminate the efficiency of the evolution process itself. More fundamentally, standard evaluation protocols permit unconstrained optimization—agents maximize task success without facing the hard token budgets, iteration limits, or latency constraints that govern real-world deployment.

### 7.1.5 Safety

**Goals & Metrics** From the perspective of self-evolving, the Safety domain critically examines whether these agents develop unsafe or undesirable behavioral patterns throughout their continuous evolution. This dimension assesses an agent's adherence to predefined rules and its propensity for harmful actions. Key metrics in evaluating safety of self-evolving agents may include: (1) Safety Score (Zhang et al., 2024i), which measures the proportion of test cases where the agent's behavior is labeled "safe"; (2) Harm Score (Andriushchenko et al., 2024), computed via a detailed manually written grading rubric where outputs earn partial credit whenever some but not all harmful criteria are triggered; (3) Completion Under Policy (CuP) (Levy et al., 2024), which assesses whether an agent successfully completes a task while strictly adhering to a given set of rules or policies; (4) Risk Ratio (Levy et al., 2024), which calculates the frequency of an agent's rule violations along a specific dimension, providing a quantitative measure of non-compliance; (5) Refusal Rate (Zhang et al., 2024b; Andriushchenko et al., 2024), which evaluates the proportion of tasks an agent refuses to perform due to their aggressive, malicious, or otherwise unsafe nature; (6) Leakage Rate (Shao et al., 2024a), which tracks how often an agent unintentionally leaks sensitive or private information.

**Coverage Gaps** Safety evaluation predominantly captures risks in isolated episodes. Agent-SafetyBench identifies two core deficiencies: inadequate robustness when deploying tools across varied contexts, and limited recognition of potential hazards in specific operational scenarios (Zhang et al., 2024i). In multi-agent settings, SwarmBench reveals fundamental coordination challenges where local interactions fail to produce coherent collective strategies, with agents unable to maintain shared situational understanding necessary for safe collaborative operation (Ruan et al., 2025). Yet no benchmark tracks safety trajectories over extended evolution—whether risks accumulate through repeated exposure to edge cases, or whether unsafe behaviors emerge through autonomous exploration and self-directed learning.

### 7.1.6 Self-Directedness and Evaluation Trade-offs

A defining characteristic of self-evolving agents is their capacity for autonomous exploration and self-directed evolution, distinguishing them from passive learning paradigms. The degree of self-directedness, specifically whether the agent autonomously generates tasks and evolution strategies or follows externally provided curricula, creates fundamental trade-offs across evaluation dimensions.

Highly self-directed systems demonstrate substantial performance gains through autonomous curriculum generation. WebRL improved from 4.8% to 42.4% by self-generating tasks from exploration failures (Qi et al., 2024), while SEAgent achieved 23.2 percentage point improvements, advancing from 11.3% to 34.5% via autonomous software exploration and curriculum evolution (Sun et al., 2025c). However, autonomous evolution incurs measurable risks. Alignment faking rates escalated from 12% to 78% when agents autonomously evolved under conflicting objectives (Greenblatt et al., 2024), illustrating safety challenges when evolution proceeds with minimal external oversight.

Despite these documented impacts, the field lacks standardized metrics for quantifying self-directedness in evolution. To enable fair comparison, we recommend transparently reporting three aspects of the evolution process. (1) specify whether evolution strategies and task sequences are predetermined, procedurally sampled, or autonomously generated by the agent; (2) Document the source of feedback signals, indicating whether they come from external human labels, rule-based evaluators, or self-generated reflection; (3) Report the frequency of external interventions in the evolution process. Clearly documenting the degree of

Table 6: Overview of Agent Evaluation Metrics Across Core Dimensions

| Goal | Metric | Description |
|---|---|---|
| Adaptivity | Success Rate by Iteration Steps (Hu et al., 2024c; Wang et al., 2024j; Zheng et al., 2025b) | Performance in downstream tasks as a function of the agent's interaction history |
| | Adaptation Speed (Wang et al., 2023a) | How quickly an agent reaches a certain performance threshold or converges to an optimal strategy within a given adaptation period |
| Retention | Forgetting (FGT) (Zheng et al., 2025c) | The average accuracy drop on old tasks after an agent learns a new one, measuring whether useful experience is successfully maintained |
| | Backward Transfer (BWT) (Zheng et al., 2025c) | The average accuracy improvement on old tasks due to the experience gained from new tasks |
| Generalization | Aggregate Performance (Liu et al., 2023b; Sun et al., 2023) | Mean success rates or other performance indicators across multi-domain test suites to gauge overall proficiency |
| | Out-of-Domain (OOD) Performance (Hu et al., 2024b; Peng et al., 2025) | The agent's performance in held-out task distributions |
| Efficiency | Token Consumption (Hu et al., 2024a) | Computational overhead in reasoning and generation steps |
| | Time Expenditure (Lu et al., 2024b) | Total duration required for task completion |
| | Number of Steps (Wang et al., 2023a) | Minimal actions needed to accomplish objectives |
| | Tool Productivity (Wang et al., 2025h) | The ratio between task benefit (e.g., answer accuracy) and tool usage cost (e.g., number of tool calls) |
| Safety | Safety Score (Zhang et al., 2024i) | Proportion of test cases where agent behavior meets predefined safety criteria |
| | Harm Score (Andriushchenko et al., 2024) | Graded assessment of harmful outputs based on violation severity |
| | Completion Under Policy (CuP) (Levy et al., 2024) | Task success rate while complying with specified constraints |
| | Risk Ratio (Levy et al., 2024) | Frequency of policy violations per interaction opportunity |
| | Refusal Rate (Zhang et al., 2024b; Andriushchenko et al., 2024) | Percentage of tasks declined due to safety concerns |
| | Leakage Rate (Shao et al., 2024a) | Incidence of unintended sensitive information disclosure |

autonomous control is essential for interpreting performance claims, as gains may reflect true self-evolution capability or extensive external guidance.

Such trajectory-centric evaluation aligns with the core objective of assessing self-evolving agents: measuring not just what they achieve, but how autonomously they learn to evolve.

## 7.2 Evaluation Paradigm

The evaluation of self-evolving agents, given their continuous learning paradigm, necessitates a multi-faceted approach that extends beyond traditional static assessments. Current evaluation paradigm can be broadly categorized based on the temporal scope of the assessment: **Static Assessment**, **Short-horizon Adaptive Assessment**, and **Long-horizon Lifelong Learning Ability Assessment**. Each category addresses different aspects of an agent's evolving capabilities, from its instantaneous performance to its long-term learning trajectory. *Terminology note.* Here, "continuous" refers to *system-level self-evolution* (changes in policies, memories, skills, tools, or procedures across episodes), not necessarily *model-parameter continual learning.* Our evaluation therefore traces trajectories across interaction episodes and evolving environments, independent of whether weights are updated.

Table 7: Representative Benchmarks for Evaluating Self-Evolving Agents

| Benchmark Name | Task Domain | Goal | Core Metrics | Task Quantity | Temporal Scope |
|---|---|---|---|---|---|
| ScienceAgentBench(Chen et al., 2024e) | Scientific Data Analysis | Adaptivity, Efficiency | Valid Execution Rate, Success Rate, CodeBERTScore, API Cost | 102 | Static |
| MLE-Bench(Chan et al., 2024) | ML-Engineering | Adaptivity | – | 75 | Static |
| DS-Bench(Jing et al., 2025) | Data Science | Adaptivity | Task Success Rate, Cost, Inference Time, Competition-level Accuracy | 540 | Static |
| SWE-bench(Jimenez et al., 2023) | Software Engineering | Adaptivity | Pass Rate | 2,294 | Static |
| OSWorld(Xie et al., 2024) | Computer-Use / GUI | Adaptivity | Success Rate | 369 | Static |
| Mobile-Eval-E(Wang et al., 2025o) | Computer-Use / GUI | Adaptivity, Efficiency | Action Accuracy, Reflection Accuracy, Termination Error | 25 | Static, Short-horizon |
| WebShop(Yao et al., 2022) | Web Search / Browse | Adaptivity | Success Rate | 12,087 | Static |
| WebArena(Zhou et al., 2023) | Web Search / Browse | Adaptivity | Success Rate | 812 | Static |
| WebWalkerQA(Wu et al., 2025) | Web Search / Browse | Adaptivity, Efficiency | Accuracy, Action Count | 680 | Static |
| ST-WebAgentBench(Levy et al., 2024) | Web Search / Browse | Safety | Completion under Policy | 235 | Static |
| xbench(Chen et al., 2025f) | Web Search / Browse | Adaptivity | LLM-Judge Score | 100 | Static |
| BrowseComp(Wei et al., 2025b) | Web Search / Browse | Adaptivity | Accuracy | 1,266 | Static |
| Agent-SafetyBench(Zhang et al., 2024i) | General | Safety | Safety Score | 20,000 | Static |
| LifelongAgentBench(Zheng et al., 2025b) | General | Adaptivity, Retention, Generalization | Success Rate | 1396 | Long-horizon |
| AgentBench(Liu et al., 2023b) | General | Adaptivity, Generalization | Success Rate, F1, Reward, Game Progress | 1360 | Static |
| GAIA(Mialon et al., 2023) | General | Adaptivity | Accuracy | 466 | Static |
| TheAgentCompany(Xu et al., 2024b) | General | Adaptivity, Efficiency | Completion Score, Steps, Cost per Instance | 175 | Static |
| EvaLearn(Dou et al., 2025) | General | Adaptivity, Efficiency | Accuracy, Slope, Position of 1st solution, Num of consecutive solutions | 648 | Long-horizon |
| PlanBench(Valmeekam et al., 2023) | Planning | Adaptivity | Accuracy | ∼26,250 | Static, Short-horizon |
| Natural Plan(Zheng et al., 2024a) | Planning | Adaptivity | Exact Match | 3,600 | Static |
| ACPBench(Kokel et al., 2025) | Planning | Adaptivity, Generalization | Accuracy | 3,720 | Static |
| AppBench (Wang et al., 2024b) | Planning | Adaptivity | Success Rate, F1 | 800 | Static |
| ToolBench(Qin et al., 2023) | Tool Usage | Adaptivity | Pass Rate, Win Rate | 126,486 | Static |
| ToolSandbox(Lu et al., 2024a) | Tool Usage | Adaptivity | Similarity Score | 1,032 | Static |
| Seal-Tools(Wu et al., 2024) | Tool Usage | Adaptivity | Accuracy, P/R/F1 | 14,076 | Static |
| API-Bank(Li et al., 2023) | Tool Usage | Adaptivity | Accuracy, ROUGE | 4,125 | Static |
| T-Eval(Chen et al., 2023) | Tool Usage | Adaptivity | Domain-Specific Score | 23,305 | Static |
| τ-Bench(Yao et al., 2024) | Tool Usage | Adaptivity | Pass^k | 165 | Static |
| AceBench(Chen et al., 2025a) | Tool Usage | Adaptivity | Accuracy | 2,000 | Static |
| LTMBenchmark(Castillo-Bolado et al., 2024) | Agent Memory | Retention, Efficiency | Score, Accuracy, GoodAI LTM Score, Speed, Cost, Verbosity | 30 | Long-horizon |
| StoryBench(Wan & Ma, 2025) | Agent Memory | Retention, Efficiency | Accuracy, First-Try Accuracy, Longest Corr, Retry Count, Runtime Cost, Token Consumption | 311 scene nodes, 86 choice nodes | Short-horizon, Long-horizon |
| MemoryAgentBench(Hu et al., 2025) | Agent Memory | Adaptivity | SubEM, Recall, ROUGE F1, Accuracy, Recall@5, Model-based Acc/F1 | 2200 | Static, Short-horizon |
| MultiAgentBench(Zhu et al., 2025) | Multi-Agent Collaboration | Adaptivity | KPI, Text-Based Score, Communication Score, Planning Score, Coordination Score | 100 | Static |
| SwarmBench(Ruan et al., 2025) | Multi-Agent Collaboration | Adaptivity | Perspective-specific Metrics | 5 | Short-horizon |

### 7.2.1 Static Assessment

Static assessment evaluates the instantaneous performance of self-evolving agents at a specific point in time. Although these agents are designed for continuous improvement, static methods remain crucial for establishing baseline performance, comparing different agent architectures on fixed task sets, or evaluating capabilities after discrete training phases. This approach aligns with conventional AI evaluation, focusing on immediate performance in fixed environments. While useful for assessing generalization in an "in-domain evolving, out-of-domain evaluation" paradigm, static assessment inherently does not capture the dynamic, continuous learning, or long-term evolutionary aspects central to self-evolving agents.

For evaluating an agent's general capabilities at a given moment, standard benchmarks designed for static AI systems are often employed. These benchmarks offer diverse task domains and test various core agent competencies, providing a snapshot of an agent's proficiency before or at specific stages of its evolution. These assessments can be systematically categorized into **External Task-Solving Evaluation** and **Internal Agent Components Evaluation**, where External Task-Solving Evaluation measures end-to-end performance in completing domain-specific or cross-domain tasks, and Internal Capability Evaluation focuses on fundamental components in the agent, including planning, tool utilization, memory management, multi-agent coordination, etc.

Table 8: **Goal-to-benchmark mapping with limitations and coverage gaps.**

| Goal | Benchmarks | Limitations | Coverage Gaps |
|---|---|---|---|
| **Adaptivity** | SWE-bench, WebArena, MLE-Bench, ScienceAgentBench, GAIA | Restricted programming languages and execution time limits (Chen et al., 2024e); Simplified problem specifications vs. authentic R&D ambiguity (Chan et al., 2024); Weak sustained reasoning and decision-making (Liu et al., 2023b) | Open-ended exploration without predetermined objectives; Long-horizon adaptation under non-stationary distributions |
| **Retention** | LifelongAgentBench, LTMBenchmark, MemoryAgentBench | Persistent challenges in dynamic memory and long-range consistency (Hu et al., 2025); Replay buffer trade-offs with context overflow (Zheng et al., 2025b); Limited robustness testing due to costs (Castillo-Bolado et al., 2024) | Episodic designs reset state between tasks; No retention with safety constraints |
| **Generalization** | AgentBench, GAIA, TheAgentCompany, MLE-Bench | Progressive task obsolescence from algorithmic advances (Chan et al., 2024); Training data contamination risks (Chan et al., 2024) | Temporal robustness under distribution drift; Adversarial co-evolutionary assessment |
| **Efficiency** | MLE-Bench, TheAgentCompany | High resource demands limit accessibility (Chan et al., 2024); Cost constraints prevent baseline collection (Xu et al., 2024b) | No enforced budgets during evolution; Multi-objective constraints absent |
| **Safety** | Agent-SafetyBench, SwarmBench | Inadequate tool robustness and risk awareness (Zhang et al., 2024i); Local-global coordination disconnect (Ruan et al., 2025) | Static evaluation only; No long-horizon safety drift tracking; Co-evolutionary safety unexplored |

**External Task-Solving Evaluation**    This category assesses an agent's end-to-end proficiency in completing tasks across various real-world or simulated environments. In **scientific data analysis and machine learning engineering**, benchmarks like ScienceAgentBench (Chen et al., 2024e) and MLE-Bench (Chan et al., 2024) test agents' ability to generate and execute code for data analysis and solve Kaggle-style problems. For **web search/browsing**, environments such as WebShop (Yao et al., 2022), WebArena (Zhou et al., 2023), X-WebAgentBench (Wang et al., 2025j), Mind2Web (Deng et al., 2023), and BrowseComp (Wei et al., 2025b) simulate realistic web interactions, complex browsing scenarios, and task completion under security constraints. In **software engineering**, the SWE-bench series (Jimenez et al., 2023; Openai, 2024; Aleithan et al., 2024; Yang et al., 2024) uses real GitHub issues to assess agents' code repair capabilities. For **computer-use interactions**, OSWorld (Xie et al., 2024) offers a unified environment for open-ended tasks involving various desktop and web applications. Specialized domains like **marketing** also feature benchmarks such as xbench (Chen et al., 2025f). Beyond specific domains, **generalist agent benchmarks** like AgentBench (Liu et al., 2023b), GAIA (Mialon et al., 2023), and TheAgentCompany (Xu et al., 2024b) evaluate broad problem-solving abilities across multiple knowledge domains and professional tasks, simulating real-world demands on general AI assistants.

**Internal Agent Components Evaluation**    Beyond end-to-end task completion, assessing an agent's underlying core competencies is crucial. These benchmarks evaluate fundamental capabilities that contribute to an agent's overall intelligence and self-evolutionary potential. As for **Planning**, benchmarks such as PlanBench (Valmeekam et al., 2023), Natural Plan (Zheng et al., 2024a), AutoPlanBench (Stein et al., 2025), and ACPBench (Kokel et al., 2025) comprehensively evaluate an agent's ability to understand dynamic environments, devise strategies, decompose complex problems, and execute reasoning in various planning domains. For **Tool Usage**, simple benchmarks like ToolAlpaca (Tang et al., 2023) and ToolBench (Qin et al., 2023) test basic selection and parameter mapping, while more complex ones like ToolSandbox (Lu et al., 2024a), Seal-Tools (Wu et al., 2024), API-Bank (Li et al., 2023), T-Eval (Chen et al., 2023), $\tau$-Bench (Yao et al., 2024), AceBench (Chen et al., 2025a) simulate real-world scenarios involving multi-turn interactions, implicit state dependencies, and nested calls. **Memory Management** benchmarks such as LTMBenchmark (Castillo-Bolado et al., 2024), MemoryAgentBench (Hu et al., 2025), and StoryBench (Wan & Ma, 2025) evaluate the agent's capacity to retain and utilize information across multi-turn interactions,

Table 9: Differences between Short-horizon Adaptive Assessment and Long-horizon Lifelong Learning Ability Assessment

| Dimension | Short-horizon Adaptation Assessment | Long-horizon Lifelong Learning Ability Assessment |
|---|---|---|
| Primary Focus | Immediate learning and incremental improvement within consistent or slightly varying tasks | Continuous knowledge accumulation and sustained performance across diverse, evolving tasks and environments. |
| Core Challenges | Rapid adaptation to minor changes; Improving on similar, repeated tasks | Mitigating catastrophic forgetting; Robust knowledge transfer; Maintaining efficiency/safety over time; handling true novelty and significant distribution shifts |
| Temporal Scope | Small number of sequential tasks or iterations over a short period; Improvement on the same or similar task types. | Large, potentially unbounded sequence of diverse, cross-domain tasks; Very long interaction periods requiring integration of new skills with old |

dynamic scenarios, and long-range dependencies. For evaluating **Multi-Agent Collaboration**, benchmarks such as MultiAgentBench (Zhu et al., 2025) and SwarmBench (Ruan et al., 2025) assess coordination, communication, and emergent swarm intelligence in both collaborative and competitive settings.

Typical metrics for static assessment include accuracy, success rate, progress rate, completion rate, and various domain-specific performance indicators (e.g., CodeBertScore, Valid Execution Rate, Pass Rate, F1 score). These metrics provide a singular performance score for an isolated invocation or a fixed set of tasks.

### 7.2.2 Short-Horizon Adaptive Assessment

Short-horizon adaptations extend beyond static evaluations by assessing an agent's ability to adapt and improve over a relatively short period or a limited number of interactions. The agent might improve performance on the same task instance with more attempts, or adapt to new instances of the same task type. This category focuses on capturing the capacity of the self-evolving agent for immediate adaptability and incremental learning within a relatively consistent or slightly varying task distribution. These evaluation schemes can be broadly categorized into two ways: (1) augment traditional benchmarks with a temporal dimension, and (2) specially design benchmarks and metrics that can inherently support Short-Horizon dynamic learning.

**Augmented Traditional Benchmarks**  Many studies leverage existing benchmarks but introduce a new dimension to track performance over time. This typically involves analyzing performance as a function of the number of iterations, steps, or examples. For example, ADAS (Hu et al., 2024c) evaluated the held-out test accuracy with the number of agent system iterations on the ARC benchmark (Chollet, 2019); AWM (Wang et al., 2024j) studied the cumulative success rate over the process of online evaluation under WebArena map test split (Zhou et al., 2023), using a number of examples to mark the evolution progress; WebEvolver (Fang et al., 2025b) studied the success rate with self-improving iterations under Mind2web-Live (Pan et al., 2024). This approach allows for tracking the Adaptivity of the agent within a confined scope.

**Benchmarks with Built-in Dynamic Evaluation**  Some benchmarks are designed with short-horizon dynamic learning in mind. MemoryAgentBench (Hu et al., 2025), for example, includes a "Test-Time Learning" (TTL) dimension that evaluates an agent's ability to learn new tasks directly from conversation within a single interaction session. In practice, TTL is evaluated through two types of tasks: Multi-Class Classification and Recommendation. In these settings, the agent must utilize previously provided information—such as labeled examples in context or a long movie-related dialogue history—to perform new tasks like mapping sentences to class labels or recommending relevant movies. This assesses immediate adaptation and knowledge acquisition during ongoing interaction.

**Metrics and Methods for Evaluating Short-Horizon Adaptations**  The primary metrics and methods for short-horizon adaptations are designed to quantify Adaptivity. These include: (1) Success Rate

by Iteration Steps (Hu et al., 2024c; Wang et al., 2024j; Zheng et al., 2025b), which tracks performance improvements as the agent interacts more with the environment or attempts a task multiple times; (2) Learning Curve Analysis, which visualizes how performance (e.g., success rate, accuracy) changes over a limited number of training steps, episodes, or interactions (Hu et al., 2024c; Wang et al., 2024j); (3) Adaptation Speed (Wang et al., 2023a), which measures how quickly an agent reaches a certain performance threshold or converges to an optimal strategy within the short horizon.

Short-horizon adaptations are well-suited for evaluating the initial learning capabilities and immediate adaptability of self-evolving agents. They can effectively demonstrate whether an agent can learn from recent experiences and improve its performance on in-domain tasks. This category is widely used for current self-evolving agents. However, the limited temporal window makes it challenging to assess long-term knowledge retention (mitigating catastrophic forgetting) and true lifelong learning capabilities across vastly different or sequentially presented tasks.

### 7.2.3 Long-Horizon Lifelong Learning Ability Assessment

Long-horizon lifelong learning ability assessment is crucial for truly assessing self-evolving agents, as they focus on the agent's ability to continuously acquire, retain, and reuse knowledge across diverse environments and over extended periods. As shown in Table 9, it mainly focuses on continuous learning, knowledge accumulation, and sustained performance across a diverse and potentially ever-changing stream of tasks or environments over an extended period. This is a nascent but critical area, where unique challenges include catastrophic forgetting, robust knowledge transfer across disparate tasks, efficient resource management over extended durations, and mitigating data leakage when continuously evaluating on evolving data distributions. Specialized benchmarks are emerging to tackle these complexities.

Currently, there are few benchmarks of this type. LTMBenchmark (Castillo-Bolado et al., 2024) is a specialized benchmark focusing on long-term memory (LTM) evaluation. It assesses LLM agents' memory retention and continual learning through dynamic conversational tests, using interleaved dialogues with controlled distractions to simulate real-world recall challenges. Key metrics include task accuracy, memory-span-weighted LTM Score, and efficiency measures (tests/hour, cost) for cross-architecture comparison. LifelongAgent-Bench (Zheng et al., 2025b) is another pioneering benchmark specifically designed to evaluate agent lifelong learning. It constructs sequences of interdependent tasks across domains like Database (DB), Operating System (OS), and Knowledge Graph (KG), requiring agents to progressively build upon previously acquired skills. This allows for systematic tracking of performance improvement and knowledge retention across a prolonged learning trajectory. To address the lack of diverse environments for testing generalization, AutoEnv (Zhang et al., 2025i) introduces a framework for automatically generating heterogeneous worlds from factorizable rule distributions. This work also contributes the AUTOENV-36 dataset to systematically measure an agent's cross-environment learning and adaptation capabilities. In addition, there is a solution that constructs a dynamic benchmark through continuously updating benchmark datasets (White et al., 2024; Yang et al., 2025c) or evolving the benchmark itself by reconstructing original benchmarks to evaluate self-evolving agents, which can alleviate data leakage to some extent (Chen et al., 2025g). Benchmark Self-Evolving (Wang et al., 2024e), for example, proposes a solution to continuously update the existing benchmark through iteration. Similarly, the TRACE framework (Guo et al., 2025a) addresses benchmark saturation by enabling agents to evolve tasks to higher difficulty through test-time exploration. The validity of these new, more complex tasks is ensured by a "validate-by-reproducing" paradigm, which confirms the agent's recorded trajectory is reproducible. Preliminary findings from such dynamic benchmark scenarios have shown that model performance can degrade as the benchmark evolves, highlighting the difficulty of continuous adaptation.

Metrics for long-horizon lifelong learning go beyond simple success rates to quantify the agent's evolving ability, such as Forgetting (FGT), Backward Transfer (BWT) (Zheng et al., 2025c), and Cost-per-Gain. Long-term Generalization metrics could involve assessing performance on a continuously evolving set of out-of-distribution tasks or measuring the breadth of tasks an agent can still perform effectively after prolonged learning across many domains.

Table 10: **Standardized evaluation protocols**

| Aspect | Short-horizon | Long-horizon | Work Example for Long-horizon (EvoAgent (Yuan et al., 2025a)) |
|---|---|---|---|
| **Goal aligned** | Adaptivity, efficiency | Retention, generalization, efficiency, long-term safety | Optimizes long-horizon Success Rate (SR) and Exploration Efficiency (EE) on 67 Minecraft tasks (five tiers) plus Atari; e.g., Overall SR improves from 21.80% to 30.29% (relative gain $\approx$ 105.9%). |
| **State persistence** | *No* persistence across tasks; all updates reset after each episode | *Full* persistence of model / prompt / memory / tools across tasks | Maintains a persistent Multimodal Experience Pool and continual World Model, updating parameters and experience after each subtask and reusing them across subsequent tasks without reset. |
| **Dataset structure** | Fixed benchmark or episodic sampling; IID or near-IID task variants | Streamed sequences with non-stationary distributions; versioned tasks; explicit OOD clusters for transfer | Uses a fixed long-horizon Minecraft benchmark (67 tasks split into Wood/Stone/Iron/Gold/Diamond tiers) plus Atari as a cross-environment test set; tasks are pre-defined rather than streamed or versioned. |
| **Evolution budget** | Per-task cap $K_{\text{short}}$ (iterations, tool calls, tokens, wall-clock) | Stage cap $K_{\text{stage}}$ + cumulative cap $K_{\text{total}}$; explicit memory/tool growth policy | Enforces a per-subtask step cap $L_{\max}$ and matches DreamerV3's environment-step budget, reporting wall-clock of $\sim$ 2.7 days vs. $\sim$ 7 days on one A100, but without explicit formulas for $K_{\text{stage}}/K_{\text{total}}$ or a formal memory/tool growth policy. |
| **Required logging** | Per-iteration: seeds, prompts, model/tool versions, full reasoning/action traces, cost breakdown | Above + persistent state checkpoints, replayable trajectories, scheduled retention probes, evolution decision logs | Logs each terminated subtask as a trajectory (states, rewards, completion ratios) into the experience pool for CL-based sampling and world-model updates, but does not publish full replayable per-iteration logs, checkpoints, or scheduled retention probes. |
| **Primary metrics** | *Adaptivity*: success-by-iteration curves, AULC; *Generalization*: within-distribution transfer | *Retention*: BWT/FGT, forgetting curves; *Generalization*: temporal & cluster-OOD; *Efficiency*: Cost-per-Gain (CPG) | Uses SR and EE per tier plus an Overall aggregate; e.g., on Gold and Diamond tiers EvoAgent roughly doubles SR over baselines, but no BWT/FGT or explicit forgetting curves are reported. |
| **Efficiency metrics** | Cost-per-Gain (CPG), tokens-per-success, tool calls, latency per iteration | Cumulative CPG, stage-wise efficiency, token-drift (tokens/task over time), memory growth rate | Efficiency is captured by EE and wall-clock (e.g., > 6× fewer ineffective steps on average and 2.7 vs. 7 days training), while CPG, token-drift, and memory-growth statistics are not explicitly reported. |
| **Safety auditing** | Per-episode: Safety Score, Harm Score, Refusal Rate; window-level Leakage Rate | Long-horizon safety drift tracking; periodic probes (Safety/Harm/CuP/Risk Ratio); persistent Leakage Rate across stages | Relies on an internal self-verification module with goal-similarity and $L_{\max}$ to terminate unproductive subtasks, but does not run external safety benchmarks or track long-horizon safety drift. |
| **Human-in-loop** | Human-time + guidance-tokens per episode; intervention count | Cumulative human-time + guidance-tokens; stage-wise intervention frequency; intervention-to-success ratio | After initial task specification, training and evaluation are fully autonomous with no human feedback or labeling, and human-time/guidance-tokens are effectively 0 (not numerically reported). |
| **Required outputs** | Learning curve + AULC; per-task summary table; cost breakdown | Learning/forgetting matrix; stage tables + long-horizon curves; detailed cost taxonomy breakdown | Provides SR/EE tables over all tiers and ablations over planner/control/reflection/world-model modules, but no learning/forgetting matrices or detailed cost-taxonomy (tokens/time/tool/memory) breakdowns. |

Long-horizon lifelong learning ability assessment is essential for comprehensively evaluating the core promise of self-evolving agents: their ability to learn continuously, retain knowledge, and generalize effectively over extended periods. They are critical for assessing Retention, Generalization to truly novel scenarios, and the Efficiency of long-term operation. This area remains a key frontier for research in evaluating self-evolving agents. Beyond fixed long-horizon streams, an *open-ended* variant continuously evolves tasks/tools/environments.

### 7.2.4 Standardized Evaluation Protocols

To address the heterogeneity of existing setups and to make trajectory-centric evaluation reproducible, we further distill the above considerations into standardized protocols for short-horizon and long-horizon assessment, summarized in Table 10. For short-horizon settings, we assume no cross-task state persistence and impose a per-task evolution budget $K_{\text{short}}$ (in iterations, tool calls, tokens, or wall-clock time), require per-iteration logging of prompts, model/tool versions, full reasoning and action traces, and cost breakdowns, and report adaptivity via success-by-iteration curves and area-under-learning-curve together with basic efficiency metrics (e.g., tokens-per-success, latency). In contrast, long-horizon protocols assume full persistence

of model parameters, prompts, memories, and toolsets across tasks, specify both stage-wise and cumulative evolution budgets ($K_{\text{stage}}, K_{\text{total}}$) alongside explicit memory/tool growth policies, and mandate richer logging: replayable trajectories, persistent checkpoints, scheduled retention probes, evolution decision logs, and human-in-the-loop statistics to expose the degree of self-directedness. Primary long-horizon metrics then expand beyond instantaneous success to include retention (FGT/BWT and forgetting curves), temporal and cluster out-of-distribution generalization, cost-per-gain and efficiency drift over time, as well as long-run safety indicators such as safety incident rates and policy-compliance under evolving behavior. Our worked example, EvoAgent (Yuan et al., 2025a), illustrates both the feasibility and current limitations of practice: it already satisfies key elements of the long-horizon protocol—persistent world-model and experience updates, explicit per-subtask step caps, and reporting of SR/EE and wall-clock efficiency—but leaves many recommended axes (e.g., standardized retention metrics, explicit CPG/token-drift, and long-term safety drift tracking) unreported, highlighting concrete gaps for future self-evolving agent evaluations to close.

## 7.3 Limitations of Current Evaluation Practices

While previous sections outlines the core evaluation dimensions and their coverage, a broader view reveals that current practices still leave substantial blind spots and hinder fair comparisons across methods. To contextualize these limitations, we examine both the capability dimensions that remain under-evaluated and the factors that complicate apples-to-apples comparison under shared settings.

### 7.3.1 Underserved Capability Intersections

**Long-horizon retention with privacy constraints**: No benchmark combines extended memory assessment (as in LTMBenchmark (Castillo-Bolado et al., 2024)) with rigorous safety auditing (as in Agent-SafetyBench (Zhang et al., 2024i)), leaving open whether agents can maintain personalization across thousands of interactions while guaranteeing zero sensitive information leakage.

**Architecture adaptation under operational constraints**: Current architecture search methods (e.g., AFlow (Zhang et al., 2024c), ADAS (Hu et al., 2024c)) operate offline over extended periods to discover optimal workflows. No evaluation examines whether agents can autonomously evolve their architecture selection strategies, learning which topologies work best for different query types, while respecting real-time operational constraints (millisecond response latencies, per-query token budgets). This requires assessing not just final architecture quality, but whether the agent's strategy for *choosing or generating* architectures improves persistently through experience under hard resource limits.

**Tool ecosystem evolution**: Existing tool benchmarks provide fixed APIs; no evaluation captures the self-directed lifecycle of tool discovery, integration testing, and productivity measurement—capabilities demonstrated by systems like Alita but absent from standard assessment.

**Multi-agent safety under collaborative evolution**: SwarmBench (Ruan et al., 2025) identifies coordination failures at isolated time points; whether these failures amplify or attenuate over extended multi-agent co-evolution, and whether unsafe behaviors exhibit social contagion when agents learn from each other, remains unexplored.

### 7.3.2 Challenges for Fair Comparison

To supplement our discussion of evaluation benchmarks, we provide Table 11, which aligns a representative subset of self-evolving agents evaluated under partially matched conditions, i.e., similar domains, benchmarks, and backbone models. This table aims to make explicit how different methods instantiate the what/when/how dimensions when the surrounding experimental context is held as constant as the literature allows. However, as the table also makes clear, true apples-to-apples comparison remains infeasible at present. Existing works differ substantially in (1) reporting practices: key metrics such as latency, cost, and safety are often omitted or defined inconsistently; (2) evaluation pipelines: prompt formats, rollout budgets, tool access, and environment configurations vary widely; (3) backbone model choices: even nominally similar models differ in size, training data, or inference settings; and (4) architectural design: agents implement distinct control loops, credit-assignment mechanisms, and memory systems that are not directly

comparable. Because these factors interact with one another, normalizing results across methods would risk over-interpreting uncontrolled differences. As a result, Table 11 is presented as an illustrative snapshot rather than a definitive comparison.

Despite these limitations, the table highlights several qualitative trends: methods using richer what structures (e.g., architecture-level search) and inter-test when mechanisms often achieve stronger performance in domains where multi-step optimization is feasible, while lightweight intra-test reflection methods tend to be more cost-efficient but yield smaller gains. The absence of consistent latency/cost/safety reporting across nearly all methods underscores a key gap that limits broader synthesis. We hope this table serves as a concrete reference for how current approaches operationalize the what/when/how dimensions under comparable settings, while simultaneously motivating the need for standardized reporting to support future apples-to-apples evaluations.

## 8 Future Direction

### 8.1 Personalize AI Agents

With the increasing interest in self-evolving agents, deploying personalized agents has become a crucial and increasingly significant objective for the research community (Zhang et al., 2024h). For instance, in applications such as chatbots, digital twins, and emotional support dialogues, a key challenge is enabling AI agents to accurately capture and adapt to users' unique behavioral patterns or preferences over extended interactions. Existing personalized agents typically depend heavily on labeled data and post-training methodologies (Cheng et al., 2024). Recent work by Zhang et al. (2025q) proposes a self-generated preference data approach aimed at rapidly personalizing LLMs. TWIN-GPT Wang et al. (2024h) leverages electronic health records to create digital twins of patients, enhancing the accuracy of clinical trial outcome predictions. However, these existing strategies hinge on the critical assumption that LLMs can consistently obtain high-quality, large-scale user data.

In practical deployment scenarios, the primary challenge remains the cold-start problem: agents need to progressively refine their personalized understanding, accurately interpret user intentions, and effectively construct user profiles, even when initial data is limited. Additionally, significant challenges persist in personalized planning and execution, such as effective long-term memory management, external tool integration, and personalized generation (ensuring outputs consistently align with individual user facts and preferences) (Li et al., 2025d). Moreover, it is essential to ensure that self-evolving agents do not inadvertently reinforce or exacerbate existing biases and stereotypes, highlighting another critical direction for future research. These governance principles are particularly important as personalized agents continuously evolve and adapt their memory or decision-making processes. Building upon these governance principles, evaluation frameworks should also evolve to ensure fairness, accountability, and alignment in personalized settings.

**Data governance.** Responsible personalization should balance adaptivity with privacy protection. First, agents ought to adopt **data minimization**: collect only task-relevant data and surface transparent, revocable controls. Empirically, web-agent benchmarks show that state-of-the-art agents frequently process sensitive data unnecessarily, motivating minimization-by-default designs (Zharmagambetov et al., 2025). Complementarily, effective data governance for personalized agents entails deploying **on-device personalization frameworks** that enable local learning from user interactions, together with user-led privacy mechanisms such as Rescriber that perform on-device redaction and approval prior to any remote data exchange (Zhou et al., 2025c). To prevent indefinite retention, **memory decay and forgetting policies** should support selective deletion and "right-to-be-forgotten"-style unlearning for personalized traces (Staufer, 2025). Finally, because self-evolution can drift safety or fairness over time, systems should include **bias monitoring and fairness auditing** loops that adapt criteria and interventions as the user and context evolve (Basu & Das, 2025), and guard against *misevolution* (safety/alignment degradation during evolution) via continuous checks on memory/tool/workflow updates (Shao et al., 2025).

**Evaluation.** With the integration of personalized data, evaluation metrics for personalizing self-evolving agents should extend beyond intrinsic evaluations (e.g., directly assessing personalized generated text quality

Table 11: **Comparative synthesis of some representative self-evolving agents under shared settings.** Methods are grouped by domain; repeated entries use "–". We summarize each method using the *what/when/how* taxonomy and report performance. Latency, cost, and safety metrics are not consistently reported, limiting full apples-to-apples comparisons.

| Area | Benchmark | Base model | Method | What | When | How | Perf. (%) |
|---|---|---|---|---|---|---|---|
| code | SWE | Gemini-1.5-pro | Reflexion (Shinn et al., 2023) | Context / lesson / reflection | Intra-test (ICL) | Reward-based (textual) | 14.3 |
| – | – | – | Learn-by-Interact (Su et al., 2025) | Context / experience / reflection | Intra-test (ICL) | Reward-based (textual) | 18.7 |
| – | – | Claude-3.5-sonnet | Reflexion (Shinn et al., 2023) | Context / lesson / reflection | Intra-test (ICL) | Reward-based | 54.4 |
| – | – | – | Learn-by-Interact (Su et al., 2025) | Context / experience | Intra-test (ICL) | Reward-based | 60.0 |
| – | WebArena | Gemini-1.5-pro | Reflexion (Shinn et al., 2023) | Context / lesson / reflection | Intra-test (ICL) | Reward-based (textual) | 20.2 |
| – | – | – | Learn-by-Interact (Su et al., 2025) | Context / experience | Intra-test (ICL) | Reward-based | 25.6 |
| web | WebArena | Claude-3.5-sonnet | Reflexion (Shinn et al., 2023) | Context / lesson / reflection | Intra-test (ICL) | Reward-based | 40.4 |
| – | – | – | Learn-by-Interact (Su et al., 2025) | Context / experience | Intra-test (ICL) | Reward-based | 48.0 |
| – | WebArena-Lite | GLM-4-9B | DigiRL (Bai et al., 2024) | Model policy | Inter-test (RL) | Reward-based (external env.) | 31.5 |
| – | – | – | WebRL (Qi et al., 2024) | Model policy | Inter-test (RL) | Reward-based (external env.) | 43.0 |
| – | – | Llama3.1-8B | DigiRL (Bai et al., 2024) | Model policy | Inter-test (RL) | Reward-based | 30.3 |
| – | – | – | WebRL (Qi et al., 2024) | Model policy | Inter-test (RL) | Reward-based | 42.4 |
| math | GSM8K | GPT-4o-mini | ADAS (Hu et al., 2024c) | Architecture / multi-agent system | Inter-test (RL / SFT hybrid) | Population-based workflow search | 90.5 |
| – | – | – | AFlow (Zhang et al., 2024c) | Architecture / multi-agent topology | Inter-test | Population-based (MCTS) | 90.8 |
| – | – | – | ScoreFlow (Wang et al., 2025n) | Architecture / multi-agent (query-specific workflow) | Inter-test | Imitation + preference optimization | 94.6 |
| – | MATH | Gemini-1.5-pro-002 | ADAS (Hu et al., 2024c) | Architecture / multi-agent system | Inter-test | Population-based | 80.0 |
| – | – | – | AFlow (Zhang et al., 2024c) | Architecture / multi-agent topology | Inter-test | Population-based | 76.0 |
| – | – | – | Mass (Zhang et al., 2025d) | Architecture / single / multi-agent design search | Inter-test | Population-based evolutionary | 84.7 |

using metrics such as ROUGE (Lin, 2004) and BLEU (Papineni et al., 2002)) or extrinsic evaluations (e.g., indirect assessments of personalization effects through recommendation systems, classification tasks, and other specific applications). Traditional personalization evaluation metrics often fail to adequately capture the evolving dynamics inherent in self-evolving agents. Consequently, future research calls for more lightweight and adaptive evaluation metrics (Zhang et al., 2024h). Additionally, to better assess self-evolving personalized agents, there is a clear need for flexible, dynamic benchmarks capable of accurately evaluating agents' performance, particularly in managing long-tailed personalization data throughout their self-evolving processes. We advocate reporting:

- **Personal Adaptation Gain (PAG):** improvement per user over $k$ sessions vs. non-personalized baseline.

- **Retention & Forgetting Balance:** metrics such as forward/backward transfer and selective-forgetting efficacy (e.g., reduction in exposure to user facts following deletion requests).

- **Privacy–Utility Trade-off:** ratio of utility gain to bits of retained personal data; complemented by a *Data Minimization Score* (proportion of shared sensitive information only when necessary) in web-agent scenarios (Zharmagambetov et al., 2025).

- **On-device Learning Ratio:** proportion of personalization updates or memory writes executed locally on the user device, potentially with user consent/redaction mechanisms (e.g., as studied in Rescriber) (Zhou et al., 2025c).

- **Bias/Drift Monitors:** longitudinal measurement of disparity or safety degradation across user groups; one might define indices such as *Fairness-Drift* or *Safety-Drift*, though standardized versions remain to be developed (Basu & Das, 2025; Shao et al., 2025).

- **Longitudinal User Outcomes:** tracking session-level satisfaction, goal-completion trends or multi-phase development trajectories (e.g., virtual "campus-life" agents in benchmarks like StuLife) (Cai et al., 2025).

## 8.2 Generalization

Self-evolving agents also face considerable challenges in achieving robust generalization across diverse task domains and environments. The fundamental tension between specialization and broad adaptability remains one of the most pressing challenges in the field, with significant implications for scalability, knowledge transfer, and collaborative intelligence.

**Scalable Architecture Design:** A central challenge in developing generalizable self-evolving agents lies in designing scalable architectures capable of maintaining performance as complexity and scope increase. Current agent systems frequently encounter a trade-off between specialization and generalization, where agents optimized for specific tasks struggle to transfer their learned behaviors to novel environments (Chen et al., 2024d). Additionally, the computational cost associated with dynamic reasoning in LLM-based agents grows non-linearly with the complexity of adaptation mechanisms, imposing practical constraints on achievable generalization within realistic resource limitations (Kim et al., 2025). Recent studies indicate that self-evolving agents equipped with reflective and memory-augmented capabilities show substantial promise for enhancing generalization, particularly in smaller, resource-constrained models (Liang et al., 2024). Nonetheless, these approaches continue to encounter limitations when addressing complex real-world scenarios that require sustained adaptation over prolonged periods.

**Cross-Domain Adaptation:** Achieving generalization across domains represents a critical frontier for self-evolving agents. Current methods frequently rely on domain-specific fine-tuning, restricting agents' adaptability to new environments without retraining (Belle et al., 2025). Recent advancements in test-time scaling and inference-time adaptation provide promising pathways for enhancing cross-domain generalization (Snell et al., 2024; Zhang et al., 2025l). These techniques allow agents to dynamically allocate additional reasoning capacity to unfamiliar scenarios by scaling computational resources during inference, avoiding the need for increasing model parameters. Additionally, meta-learning strategies have demonstrated considerable potential in facilitating rapid few-shot adaptation to new domains (Bilal et al., 2025). However, their effectiveness critically depends on an agent's capability to accurately determine when supplementary computational resources are necessary and efficiently distribute these resources across diverse reasoning tasks.

**Continual Learning and Catastrophic Forgetting:** Self-evolving agents must continuously adapt to new tasks while retaining previously acquired knowledge, a challenge exacerbated by the catastrophic forgetting phenomenon (Ghosal et al., 2024) of continual memorization (Chen et al., 2024b) inherent in LLMs (Bell et al., 2025). The stability-plasticity dilemma becomes particularly acute in foundation model-based

agents, where the computational costs of retraining for every new task are prohibitive (Zheng et al., 2025c). Recent research has explored parameter-efficient fine-tuning methods, selective memory mechanisms, and incremental learning strategies to mitigate catastrophic forgetting while preserving adaptability (Wang et al., 2024c). Nonetheless, achieving an optimal balance between efficiency and preventing model drift remains a significant open challenge, especially when agents operate under resource constraints or manage streaming data with stringent privacy considerations.

**Knowledge Transferability:** Recent studies have identified critical limitations in knowledge transfer among AI agents. Shi et al. (2025a) emphasized that knowledge integration and transfer capabilities in current agents still require significant optimization. In particular, Geng et al. (2025b) found that LLM-based agents often fail to effectively propagate newly acquired knowledge from interactions to other agents, restricting their collaborative potential. Furthermore, Vafa et al. (2025) revealed that foundation models might depend heavily on shallow pattern matching, rather than developing robust and transferable internal world models. These findings indicate several important future research directions: 1) it is essential to better understand the conditions under which knowledge acquired by one agent can be reliably generalized and communicated to others; 2) developing methods to quantify the limitations in agents' knowledge transferability could lead to clearer insights into agent collaboration bottlenecks; 3) we need to have an explicit mechanism that encourage the formation of robust, generalizable world models could significantly improve the collaborative effectiveness of self-evolving agents.

### 8.3 Safe and Controllable Self-Evolving Agents

As autonomous AI agents become increasingly capable of learning, evolving, and performing complex tasks independently, ensuring their safety and controllability has become a paramount concern. Unlike static systems, the very nature of self-evolution introduces unique and amplified risks that emerge dynamically over the agent's lifecycle. These risks are not merely extensions of traditional AI safety issues, such as those arising from vague user instructions or environmental threats like malicious phishing links (Zhou et al., 2025d), but are fundamentally tied to the agent's capacity for autonomous self-modification and adaptation (Shao et al., 2025; Han et al., 2025). This section delineates the emergent risks unique to self-evolving agents and then discusses a set of prescriptive guardrails and mitigation strategies for building safer systems.

#### 8.3.1 Emergent Risks in Self-Evolving Systems

Recent research has identified new risks that arise from the autonomous self-improvement process itself. These risks are not necessarily present in the initial agent but can manifest over time as it evolves. We categorize these risks along the primary evolutionary pathways of an agent: the backbone model, memory, and tools.

- **Uncontrolled behavior drift in model evolution**: A core risk is that an agent's goals and values may drift away from original human intent as it evolves. This is exacerbated by the learning uncertainty inherent in self-evolution, especially when operating in ambiguous contexts (Anwar et al., 2024; Bagdasarian et al., 2024). A phenomenon termed "misevolution" (Shao et al., 2025) can occur during model evolution. For instance, self-training on agent-generated data can lead to "catastrophic forgetting" of safety alignment, causing agents to execute harmful instructions they previously refused, such as interacting with malicious content they were trained to avoid (Shao et al., 2025; Hahm et al., 2025).

- **Deployment-time reward hacking in memory evolution**: Open-ended evolution is susceptible to reward hacking. Agents may find and exploit loopholes in self-defined reward signals or internal feedback. This is particularly evident in memory evolution, where the accumulation of experience can inadvertently induce unsafe behaviors. For example, an agent might learn to issue unnecessary refunds because its memory correlates them with high satisfaction ratings (Shao et al., 2025), and such risks can be further exacerbated by poorly designed memory modules (Wang et al., 2025b). A related concept is the "Alignment Tipping Process (ATP)," where an initially aligned agent discovers

that misaligned behaviors are more rewarding, causing its policy to "tip" and abandon its initial constraints (Han et al., 2025).

- **Safety of self-created and ingested external tools**: The ability of agents to autonomously generate and use tools introduces significant safety issues. Agents may spontaneously create tools with security vulnerabilities or fail to identify malicious code when ingesting external tools (Shao et al., 2025). This turns the agent into a potential vector for security threats. A major challenge here is that agents still struggle to differentiate between necessary and irrelevant sensitive information (Zharmagambetov et al., 2025), potentially leading them to create tools that leak private data. Furthermore, as agents improve, they may become more adept at creating and executing offensive cyber-operations, a risk that requires dynamic assessment (Wei et al., 2025a).

### 8.3.2 Prescriptive Guardrails and Mitigation Strategies

Addressing the emergent risks of self-evolution requires moving beyond descriptive warnings to implementing prescriptive, actionable guardrails. While early frameworks like TrustAgent have explored multi-stage strategies (i.e., pre-, in-, and post-planning) to foster safer behavior (Hua et al., 2024), the unique dynamics of self-evolution call for a more comprehensive "safety lifecycle" approach. Future research and development are expected to focus on integrating safeguards at every stage of the agent's operation.

- **Sandboxing and verification for tool and code execution**: To mitigate risks from tool use, all agent-generated or externally-sourced tools must be executed in a strictly sandboxed environment. Furthermore, automated safety verification, such as static analysis and vulnerability scanning, should be a default step before a new tool is integrated (Labs, 2025). For securing tool interaction protocols, runtime defense pipelines (Xing et al., 2025; Wang et al., 2025a) provide layered detection against threats like tool poisoning and prompt injection.

- **Audit trails and failsafes for self-modification**: Any self-modification must be accompanied by a comprehensive audit trail. This ensures that changes are traceable and reversible. Implementing rollback and failsafe patterns is critical, allowing the system to revert to a previously known safe state if undesirable behavior is detected. For memory, proactive defenses like A-MemGuard propose dual-memory structures and consensus-based validation to identify and isolate "poisoned" memories before they corrupt behavior (Wang et al., 2025b; Wei et al., 2025d).

- **Continuous monitoring and red-teaming for long-horizon drift**: Static, pre-deployment safety evaluations are insufficient. Continuous monitoring of agent behavior is necessary to detect long-horizon value drift. This can be achieved through red-teaming scenarios designed to test for emergent misalignment. For GUI agents, hybrid validation frameworks like OS-Sentinel combine formal verifiers with contextual judges to provide robust, in-workflow safety monitoring (Sun et al., 2025b).

- **Approval gates and privacy-protection measures**: For high-stakes actions, approval gates requiring human-in-the-loop confirmation should be implemented. Furthermore, given that agents struggle with handling sensitive data (Zharmagambetov et al., 2025), robust privacy-protection measures are necessary to prevent leakage and ensure a balanced and secure deployment.

To aid practitioners, we synthesize these strategies into a compliance checklist for deploying self-evolving agents (See Table 12). In conclusion, deploying reliable, controllable, and safe self-evolving systems is a critical and active area of research. Future work must move towards building a comprehensive safety-aware evolutionary lifecycle, integrating robust verification, continuous monitoring, and adaptive guardrails to ensure that the agents remain aligned with human values and safety constraints as they become more autonomous.

### 8.4 Ecosystems of Multi-Agents

Multi-agent self-evolving systems face several unique challenges that require further exploration.

Table 12: Compliance checklist for deploying self-evolving agents

| Category | Compliance Checklist for Deployment |
|---|---|
| **Tool & Code Safety** | ☐ **Strict Sandboxing:** All tools and agent-generated code execute in an isolated environment with no default access to host files, network, or sensitive processes.
☐ **Resource Limiting:** The sandbox imposes strict limits on CPU, memory, and execution time to prevent denial-of-service or runaway processes.
☐ **Automated Security Verification:** A mandatory pipeline performs static analysis (e.g., SAST) and vulnerability scanning on all new or modified tools before use.
☐ **Dependency Scanning:** The verification pipeline checks all third-party libraries and dependencies for known vulnerabilities.
☐ **Risk-Based Access Control:** Tools are classified by risk level, and high-risk capabilities (e.g., file system writes, API calls) require explicit approval via an approval gate. |
| **Self-Modification Control** | ☐ **Immutable Audit Trail:** All self-modifications (to model weights, memory, toolset, or core logic) are logged with details on the trigger, changes made, and outcome.
☐ **Version Control for Safe States:** The agent's state (model, memory, tools) is versioned, with known "safe" versions clearly tagged.
☐ **Tested Rollback Mechanism:** A reliable, one-click rollback mechanism exists to revert the agent to a previously known safe version. This mechanism is regularly tested.
☐ **Pre-Update Safety Validation:** Before a self-modified model is deployed, it is automatically evaluated against a "golden dataset" of safety-critical prompts to prevent catastrophic forgetting of alignment. |
| **Behavioral & Alignment Safety** | ☐ **Continuous Runtime Monitoring:** An active monitoring system tracks agent actions, flagging deviations from expected behavior, anomalous resource usage, or signs of unsafe actions.
☐ **Reward Hacking Detection:** Key metrics are monitored for signs of reward hacking (e.g., exploiting loopholes in reward functions). Alerts are configured for sharp, unexplained metric changes.
☐ **Automated Red-Teaming:** A continuous red-teaming framework is active, programmatically generating and running test scenarios to probe for emergent misalignment, value drift, and new failure modes.
☐ **Goal Guardrails:** Strict constraints are placed on the agent's ability to modify its own fundamental goals or safety constraints. Any such change requires human review. |
| **Data Privacy & Memory Integrity** | ☐ **Proactive Memory Defense:** Mechanisms like dual-memory structures or consensus validation are in place to detect, isolate, and neutralize potentially "poisoned" or harmful memories before they influence behavior.
☐ **PII Detection and Sanitization:** Automated tools are used to detect and redact/anonymize Personally Identifiable Information (PII) before it is stored in long-term memory or used in training.
☐ **Data Minimization Principle:** The agent is configured to only collect and retain data that is strictly necessary for its tasks, and data retention policies are enforced.
☐ **Privacy Regulation Compliance:** The system is designed to comply with relevant data privacy regulations (e.g., GDPR, CCPA). |
| **Operational Controls & Governance** | ☐ **Human-in-the-Loop for Critical Actions:** High-stakes actions (e.g., large financial transactions, data deletion, communication with external users) are gated by a mandatory human approval step.
☐ **Clear Incident Response Plan:** A documented plan is in place for responding to safety failures, including steps for immediate shutdown, rollback, and analysis.
☐ **Centralized Dashboard for Oversight:** A dashboard provides human operators with real-time visibility into the agent's behavior, state, and active safety alerts.
☐ **Explainability & Traceability:** The system provides clear explanations for why a particular action was taken, linking it back to specific memories, goals, or model inferences in the audit trail. |

**Balancing Individual and Collective Reasoning:** Recent studies highlight the difficulty of balancing independent reasoning with effective group decision-making in multi-agent environments (Chen et al., 2025d;

Sun et al., 2025a). While collective discussions can significantly enhance diagnostic reasoning, agents often risk becoming overly reliant on group consensus, thereby diminishing their independent reasoning capabilities. To mitigate this issue, future research should explore dynamic mechanisms that adjust the relative weight of individual versus collective input. Such an approach would help prevent decision-making from being dominated by a single or a small subset of agents, ultimately promoting robust, balanced consensus-building and innovation. Additionally, developing explicit knowledge bases and standardized updating methodologies—leveraging agents' successes and failures—could further improve the agents' self-evolution abilities and strengthen their individual reasoning contributions within collaborative contexts.

**Efficient Frameworks and Dynamic Evaluation:** Another crucial challenge lies in developing efficient algorithms and adaptive frameworks that allow agents to collaborate effectively while preserving their individual decision-making strengths. (Hu et al., 2024d) introduced adaptive reward models and optimized dynamic network structures, which can significantly enhance cooperative self-improvement among agents. However, a major gap identified by (Sun et al., 2025a) is the absence of clear mechanisms for agents to dynamically manage and update their knowledge. Addressing this issue will require new frameworks that explicitly integrate continuous learning and adaptive collaboration mechanisms. Furthermore, existing benchmarks for multi-agent evaluation are predominantly static (Zhu et al., 2025) and therefore fail to capture the long-term adaptability and continuous evolution of agent roles. Future benchmarks should incorporate dynamic assessment methods, reflecting ongoing adaptation, evolving interactions, and diverse contributions within multi-agent systems, thus providing more comprehensive evaluation metrics for self-evolving agents.

## 9 Conclusion

The emergence of self-evolving agents marks a paradigm shift in artificial intelligence, moving beyond static, monolithic models toward dynamic agentic systems capable of continual learning and adaptation. As language agents are increasingly deployed in open-ended, interactive environments, the ability to evolve, adapting reasoning processes, tools, and behaviors in response to new tasks, knowledge, and feedback, has become essential for building the next generation of agentic systems. In this survey, we provide the first comprehensive and systematic review of self-evolving agents, organized around three foundational questions: *what aspects of an agent should evolve, when evolution should occur, and how to implement evolutionary processes effectively.* Moreover, we discuss several methods for evaluating the progress of self-evolving agents in terms of metrics and benchmarks, followed by corresponding applications and future directions. The evolution of these agents will require significant advancements in models, data, algorithms, and evaluation practices, and so on. Addressing issues such as catastrophic forgetting, human preference alignment during autonomous evolution, and the co-evolution of agents and environments will be key to unlocking agents that are not only adaptive but also trustworthy and aligned with human values. We hope this survey provides a foundational framework for researchers and practitioners to design, analyze, and advance the development and progress of self-evolving agents.

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
