# OpenReview forum: "A Survey of Self-Evolving Agents: What, When, How, and Where to Evolve on the Path to Artificial Super Intelligence"
_TMLR — Accepted by TMLR_

### Review · Reviewer_nmCF · 2025-09-22

**Summary Of Contributions:**

Contributions -
The paper proposes a structured framework for 'self-evolving agents' centered on three orthogonal questions: what to evolve (model, context/memory/prompts, tools, and agentic architecture), when to evolve (intra-test-time vs. inter-test-time via ICL, SFT, RL), and how to evolve (reward-/feedback-driven, imitation/demonstration, and population-based evolution), adding a complementary where dimension for domains and settings. It differentiates self-evolving agents from adjacent paradigms (curriculum learning, lifelong learning, model editing/unlearning) by emphasizing active exploration, non-parametric evolution (memory, tools, workflows), and test-time adaptation, providing formal definitions and an agent system formulation.
It catalogs mechanisms and exemplars across the four 'what' pillars (e.g., self-rewarding/self-challenging policy updates, memory evolution such as Mem0/Expel, prompt optimization, and tool discovery/mastery/management) and across single- and multi-agent architectural optimization. It surveys evaluation goals and paradigms tailored to self-evolution (adaptivity, retention, generalization, efficiency, safety) and argues for moving beyond static benchmarks toward short-horizon adaptive and long-horizon lifelong assessments, listing representative benchmarks and metrics. It synthesizes applications (coding, GUI/computer-use, healthcare, finance, education) and outlines open challenges and future directions (personalization, scalable/generalizable architectures, safety/governance, multi-agent ecosystems, dynamic evaluation co-evolution).

Key strengths -
* Clear, comprehensive taxonomy that unifies scattered literature into a practical design space of 'what/when/how/where' with ample exemplars and cross-references that make the space navigable for both researchers and practitioners.
* Useful formalization and careful distinctions from related paradigms, highlighting the centrality of non-parametric evolution and active exploration at test time.
* Strong coverage of evaluation dimensions and the need for longitudinal, dynamic assessments; the emphasis on co-evolving benchmarks is timely and actionable.
* Breadth of application survey and architectural perspectives (single vs. multi-agent) provides a balanced view from mechanisms to systems.

Key weaknesses -
* Scope and recency breadth come at the expense of depth: several cited lines (e.g., safety verification for code/tool creation, robust reward design, credit assignment in workflow co-optimization) would benefit from deeper critical analysis and clearer best-practice recommendations.
* Limited quantitative synthesis: no meta-analysis or standardized comparative tables across methods under shared evaluation settings, making it hard to infer which evolution strategies work best under which constraints.
* Evaluation guidance, while comprehensive conceptually, could include more concrete protocols, canonical task suites, and reporting templates to accelerate reproducibility and comparability.
* Safety and governance sections flag important risks but stop short of prescriptive guardrails (e.g., sandboxing standards for tool creation, audit trails for self-modifying workflows, red-teaming for long-horizon drift).

**Audience:**

Yes

**Audience Explanation:**

* Research fit: The survey targets an active shift from static LLMs to adaptive, agentic systems, offering a unifying 'what/when/how/where' framework that organizes a rapidly fragmented literature, squarely relevant to TMLR readers working on learning paradigms, agents, and evaluation.

* Practitioner utility: It catalogs concrete mechanisms (model/context/tools/architecture), ties them to temporal adaptation regimes, and contrasts method families with design trade‑offs, giving practitioners actionable scaffolding for system design under real constraints.

* Evaluation emphasis: It synthesizes goals and metrics for adaptivity, retention, generalization, efficiency, and safety, and discusses static, short‑horizon, and lifelong evaluation paradigms, useful to TMLR’s evaluation‑focused community.

* Application breadth: Coding, GUI/web agents, healthcare, finance, and education case studies illustrate transfer from principles to practice, which appeals to applied ML researchers seeking domain-grounded guidance.

* Timeliness and gaps filled: Prior surveys touch agents broadly, this one centers self‑evolution as a first‑class topic, addressing open questions on what to evolve, when, and how, meeting a current need for synthesis

**Broader Impact Concerns:**

* Autonomy and misalignment: Self‑evolving agents can change prompts, memory, tools, and workflows without human oversight, raising risks of goal drift, deceptive strategies, or unsafe tool synthesis; mandate guardrails for self‑modification, human approval gates, and rollback mechanisms.

* Tool/code creation risks: Autonomous tool discovery and code generation can introduce exploitable vulnerabilities, data exfiltration paths, or misuse; require sandboxing by default, static/dynamic analysis, provenance tracking, and least‑privilege execution policies.

* Privacy and data governance: Memory evolution and personalization can accumulate sensitive user data over long horizons, creating re‑identification and leakage risks; specify data minimization, retention limits, consent, on‑device or partitioned storage, and redaction policies; report leakage metrics.

* Safety and evaluation drift: Long‑horizon self‑evolution may degrade safety compliance over time; propose continuous safety auditing, policy‑compliance metrics, and red‑teaming protocols tailored to evolving workflows and tools.

**Claims And Evidence:**

Yes

**Claims Explanation:**

* The descriptive claims are well supported. The taxonomy and definitions are grounded in formalism, with clear what/when/how/where dimensions, and extensive, well-chosen exemplars mapped to each dimension. Figures and tables make the evidence traceable and convincing within a survey’s scope.

* The distinctions from adjacent paradigms (curriculum, lifelong learning, model editing/unlearning) are accurate and justified, with specific criteria (test‑time adaptation, non‑parametric evolution, active exploration) and cited examples that substantiate the differences.

* Evaluation guidance is directionally sound (adaptivity, retention, generalization, efficiency, safety) and supported by benchmark discussions, but it is more programmatic than prescriptive, concrete, standardized protocols and head‑to‑head results are limited.

* Applications are documented with representative systems and use cases, supporting claims of relevance, though assertions about readiness and best practices remain qualitative.

* Net: evidence is accurate and clear for the survey’s organizational and scoping claims, but it is less conclusive for comparative performance or one‑size‑fits‑all evaluation prescriptions due to the absence of unified quantitative syntheses.

**Requested Changes:**

Critical -
* Add standardized evaluation protocols: Provide concrete, reproducible protocols for short‑horizon and long‑horizon assessments, including dataset splits, iteration budgets, reporting templates, and recommended metrics per goal (adaptivity, retention, generalization, efficiency, safety); include at least one end‑to‑end example protocol to anchor the taxonomy.

* Include comparative synthesis: Add summary tables that normalize representative methods under shared settings (where possible), clarifying which “what/when/how” configurations work best under specific constraints (latency, cost, safety) and domains; note gaps where apples‑to‑apples comparisons are not yet feasible.

* Make safety/governance guidance prescriptive: Translate the risk discussion into actionable guardrails (e.g., code/tool sandboxing defaults, audit trails for self‑modifying workflows, approval gates for tool creation, red‑team scenarios, rollback/failsafe patterns), plus a minimal compliance checklist for deployment.

Strengthening -
* Benchmark curation and gaps: Map evaluation goals to specific benchmarks with known limitations, propose modifications to cover dynamic/co‑evolution scenarios, and highlight underserved areas (e.g., long‑horizon memory with safety constraints, query‑specific architecture generation under latency budgets).

* Cost/efficiency accounting: Provide a cost taxonomy (tokens, steps, wall‑clock, tool calls) and simple formulas for cost‑per‑gain and tool productivity; add a small worked example to illustrate trade‑offs.

* Personalization and data governance: Offer guidance for personalized agents (data minimization, consent, on‑device memory, decay/forgetting policies, bias checks), plus evaluation metrics tailored to evolving personal contexts.

---

> ### Author Response · Authors · 2025-11-16
> **Response to Reviewer nmCF**
>
> We thank the reviewer for the constructive feedback. We have updated the manuscript accordingly, with **new content shown in blue** and **revised text in red**.
>
> As the major weaknesses are explicitly summarized in the Requested Changes, we address each of them in turn below.
>
> > Critical 1: Add standardized evaluation protocols
>
> **Response:**
> Thanks for this important suggestion, which helped us clarify and sharpen **Section 7.2 (Evaluation Paradigm)**. We substantially revised to provide concrete evaluation protocols:
> - We have added a subsubsection “Standardized Evaluation Protocols” and **Table 10**, which specify, for both short- and long-horizon settings: state persistence assumptions, dataset structure, evolution budgets and memory/tool growth policies), required logging (e.g., replayable trajectories, retention probes, evolution decision logs, human-in-the-loop statistics), and recommended metrics for each goal (adaptivity, retention, generalization, efficiency, safety).
> - The new column of Table 10 provides an **end-to-end worked example** based on EvoAgent, mapping a recent long-horizon self-evolving agent to all protocol dimensions. This shows how taxonomy can be instantiated in practice and also makes visible which long-horizon aspects (e.g., standardized retention metrics, CPG/token-drift, long-term safety drift) are still missing in current systems.
>
> These changes turn the evaluation taxonomy into concrete protocols that future benchmarks and system designers can directly adopt or extend.
>
> > Critical 2: Include comparative synthesis
>
> We thank the reviewer for the helpful suggestion. We agree that a normalized comparison across representative self-evolving agents would be valuable. However, such a controlled and strict comparison is currently not feasible due to following issues:
>
> (1) inconsistent reporting of latency/cost/safety metrics,
>
> (2) heterogeneous backbone models,
>
> (3) divergent evaluation pipelines (prompts, rollout budgets, tool availability),
>
> (4) substantial architectural variation across methods.
>
> These factors make apples-to-apples normalization unreliable.
>
> *To alleviate your concern and provide a meaningful synthesis, we added a new subsection “Comparative Synthesis under Shared Settings” and Table 11 at Section 7.3.2.* This table compares only methods with partial aligned conditions (i.e., same domain, same benchmark, same backbone models), and presents a descriptive snapshot of how their what/when/how configurations are instantiated. We explicitly avoid over-generalization and highlight missing latency/cost/safety reporting as an open challenge for future standardization.
> We hope this addition addresses the reviewer’s request while remaining faithful to current limitations in the literature.
>
> > Critical 3: Make safety/governance guidance prescriptive
>
> Thank you for the constructive comment. To fully address this point, we have **substantially re-organized Section 8.3** and expanded it into **two new subsections with a clear diagnostic → mitigation structure:**
> - **New Section 8.3.1 Emergent Risks in Self-Evolving Systems**, which offers a comprehensive analysis of the unique and heightened risks associated with self-evolving agents. This discussion covers issues such as uncontrolled behavioral drift, deployment-time reward hacking, and the safety of self-created and ingested external tools. We believe that highlighting these emergent risks provides a timely warning to the community and underscores the urgent need for safety frameworks specifically tailored to self-evolving agents.
> - **New Section 8.3.2 Prescriptive Guardrails and Mitigation Strategies**. Building directly on the risks identified in 8.3.1, this subsection presents actionable, prescriptive guidance. We discuss code/tool sandboxing defaults, audit trails for self-modifying workflows, rollback and failsafe patterns, approval gates for tool creation, and continuous monitoring with red-team scenarios. We also incorporate recent defensive mechanisms such as mcp-scan [cite 1] and MCP-Guard [cite 2].
> - Furthermore, we **synthesize these strategies into a "compliance checklist for deployment" in Table 12**. We also ground these recommendations in recent literature on defensive mechanisms, such as A-MemGuard for memory protection [cite3] and OS-Sentinel for GUI agent validation [cite4]. We believe these substantial additions make the safety and governance section far more prescriptive and valuable.
>
> [cite1] Invariant Labs. Mcp-scan. https://github.com/invariantlabs-ai/mcp-scan
>
> [cite2] Xing et al. MCP-Guard: A Defense Framework for Model Context Protocol Integrity in Large Language Model Applications. https://arxiv.org/abs/2508.10991
>
> [cite3] Wei et al. A-MemGuard: A Proactive Defense Framework for LLM-Based Agent Memory. https://arxiv.org/abs/2510.02373
>
> [cite4] Sun et al. OS-Sentinel: Towards Safety-Enhanced Mobile GUI Agents via Hybrid Validation in Realistic Workflows. https://arxiv.org/abs/2510.24411

---

> ### Author Response · Authors · 2025-11-16
> **Response to Reviewer nmCF (continued)**
>
> > Strengthening 1: Benchmark curation and gaps
>
> **Response:**
> Thank you for the valuable feedback. We have significantly expanded **Section 7.1** and added new material in **Section 7.3** to better connect goals, benchmarks, and gaps.
> - **Goal–benchmark mapping and coverage gaps.** For each goal (Adaptivity, Retention, Generalization, Efficiency, Safety), Section 7.1 now includes a “Goals & Metrics” and a “Coverage Gaps” paragraph. We added Table 7 (representative benchmarks) and Table 8 (goal–to–benchmark mappings with limitations and gaps), which explicitly link benchmarks to goals and discuss limitations such as episodic resets, narrow domains, short horizons, static snapshots, and lack of long-run safety or cost reporting. We also distinguish sandbox settings (e.g., Minecraft) from more realistic GUI/Web/SWE environments (WebArena, OSWorld, etc.).
> - **Underserved capability intersections and co-evolution.** In Section 7.3 (Limitations of Current Evaluation Practices), we added “Underserved Capability Intersections”, which identifies four missing but important scenarios: (i) long-horizon retention under privacy/safety constraints; (ii) architecture adaptation under latency and token budgets; (iii) tool ecosystem evolution with self-directed tool discovery and integration; and (iv) multi-agent safety under collaborative co-evolution. We also added Table 11 to align representative self-evolving agents under shared settings and to show why current reporting still prevents clean apples-to-apples comparison.
>
> Our aim is to move beyond a benchmark list and provide a clear map of what is covered, what is missing, and how evaluation should evolve for truly self-evolving agents in more realistic environments.
>
>
> > Strengthening 2: Cost/efficiency accounting.
>
> **Response:** We have now optimized the cost taxonomy and incorporated it as a key focus of the evaluation. The specific optimizations are as follows:
>
> - In addition to the tokens, steps, wall-clock time, and tool calls you mentioned, we have also included Memory (Context) Growth in the taxonomy for long-horizon scenarios such as self-evolving. Given the current development stage (where fully autonomous self-evolving has not yet been achieved), we have also incorporated the cost of Human Oversight into our considerations. Details can be found in Table 5.
> - Based on practical utility, we have linked cost and efficiency, provided clear definitions and formalization for both **Cost-per-Gain (CPG)** and **Tool Productivity (TP)** and show how they combine the above cost dimensions with performance gains, and added a worked example in Table 5 to facilitate understanding.
> - We introduced a discussion of the evolution that truly arises from self-directness in actual agent operations [subsection 7.1.6], calling attention to cost-aware trajectories.
>
> These changes encourage future work to report cost and efficiency in a more standardized, trajectory-aware way, rather than only reporting success rates.
>
> > Strengthening 3: Personalization and data governance.
>
> **Response:** We expanded the “Section 8.1: Personalize AI Agents”  by adding a new **Data Governance** paragraph that offers concrete guidance for responsible personalization.* This includes detailed discussions on data minimization and user consent [cite 1], on-device personalization and redaction tools such as Rescriber [cite 2], memory decay and right-to-be-forgotten policies [cite 3], and bias/fairness auditing loops to prevent misevolution in self-evolving systems [cite 4, 5].
>
> We also introduced a dedicated Evaluation subsection proposing adaptive metrics: Personal Adaptation Gain, Privacy–Utility Trade-off, On-device Learning Ratio, and Longitudinal User Outcomes on dynamic benchmarks such as StuLife [cite 6], to measure fairness, stability, and personalization quality over time. These changes were made to provide explicit guidance on data governance and evaluation in evolving personalized agents, which clarifies practices for privacy-preserving personalization and introduces a coherent evaluation framework that connects governance principles with measurable outcomes.
>
> [cite 1] Zharmagambetov, A., et al. (2025). AgentDAM: Benchmarking Data Minimization and Sensitive Information Leakage in Web Agents. arXiv:2503.09142.
>
> [cite 2] Zhou, Y., et al. (2025). Rescriber: User-Led Redaction and Consent Mechanisms for On-Device LLM Personalization. arXiv:2505.11763.
>
> [cite 3] Staufer, J., et al. (2025). What Should LLMs Forget? Quantifying Personal Data in Large Language Models for Right-to-Be-Forgotten Requests. arXiv:2502.06475.
>
> [cite 4] Basu, D., & Das, U. (2025). The Fair Game: Auditing and Debiasing AI Algorithms Over Time. arXiv:2508.06443.
>
> [cite 5] Shao, S., et al. (2025). Your Agent May Misevolve: Emergent Risks in Self-Evolving LLM Agents. arXiv:2509.26354.
>
> [cite 6] Cai, J., et al. (2025). StuLife: Building Self-Evolving Agents via Experience-Driven Lifelong Learning. arXiv:2508.19005.

---

> > ### Comment · Reviewer_nmCF · 2025-11-25
> > **Thank you for the responses**
> >
> > Hi authors,
> > Thank you for the thorough and thoughtful revision. The additions on standardized evaluation protocols, comparative synthesis under partially aligned conditions, and prescriptive safety/governance guidance are strong and substantially improve the survey’s practical value. I also appreciate the clearer treatment of personalization and data governance, which helps connect high-level principles to concrete evaluation metrics and deployment practices. Overall, the manuscript is now significantly stronger, and my recommendation remains to accept.

---

> > > ### Author Response · Authors · 2025-11-26
> > > **Official Comment by Authors**
> > >
> > > We sincerely thank the reviewer for the positive feedback. We are glad to hear that the revisions met your expectations and improved the practical value of the survey. We are grateful for your constructive feedback throughout the review process and appreciate your time and your recommendation to accept.

---

### Review · Reviewer_NCWP · 2025-10-14

**Summary Of Contributions:**

This survey focuses on Self-Evolving Agents, which I interpret as being closely related to self-improvement and post-training techniques aimed at building more capable agentic systems using foundational models and LLMs. The paper explores several mechanisms for enabling the evolution process and also discusses long-term goals for developing fully autonomous AI agents.

Stengths
1. The survey is well-written and thoughtfully structured, covering key topics relevant to agent post-training and self-improvement.
2. Section 6 is particularly engaging, offering a compelling overview of potential application areas where self-evolving agents could have a significant impact.

Weaknesses
1. I’m concerned about the framing of the title and abstract. While the paper discusses self-evolving agents, many of the techniques—such as heuristics, reward functions, and training strategies—are still human-designed. The claim that this work "paves the way toward Artificial Superintelligence (ASI)" feels overly ambitious and should be more cautiously worded.
2. The discussion of evolutionary approaches is quite limited. Although AlphaEvolve is briefly mentioned, evolutionary and genetic algorithms have a long history of improving agent behavior and should be covered more thoroughly, especially given the paper’s focus on "evolving" agents. Including this would broaden the scope and enrich the survey beyond learning-based methods.
3. A crucial omission is the topic of self-play, which by definition involves agents improving through interactions with versions of themselves or other agents. This is a core form of self-improvement and should be included in the discussion, along with how it differs from other techniques.
4. The description of imitation learning could be refined. Since imitation learning involves learning from expert demonstrations, this distinction should be clearly articulated to avoid conflating it with other forms of self-improvement.

**Audience:**

Yes

**Audience Explanation:**

The paper will be useful to a diverse audience, including those in reinforcement learning, NLP, and AI for science. I believe it would be a valuable addition to TMLR.

**Claims And Evidence:**

Yes

**Claims Explanation:**

I believe the paper offers a solid survey of methods to improve agents and highlights promising areas of application. The supporting evidence is clear and compelling.

**Requested Changes:**

I encourage the authors to address the weaknesses outlined above. Once these concerns are resolved, I believe the paper will be a strong candidate for acceptance.

---

> ### Author Response · Authors · 2025-11-16
> **Response to Reviewer NCWP**
>
> We thank the reviewer for the constructive feedback. We have updated the manuscript accordingly, with **new content shown in blue** and **revised text in red**.
>
> > Weakness1: I’m concerned about the maturity and readiness of these techniques for deployment...
>
> **Response:**
> We have followed your suggestion to revise the title and abstract of the survey. Specifically, we change the title to "A Survey of Self-Evolving Agents: What, When, How, and Where to Evolve," and remove the speculative claim about ASI in Abstract, Introduction, and Conclusion. We believe these changes will strengthen the manuscript.
>
> > Weakness 2: The discussion of evolutionary approaches is quite limited...
>
> **Response:**
> Thank you for your valuable suggestion. We have added a discussion of the history of using evolutionary and genetic algorithms for improving agent behavior at the **beginning of Section 5.3**. In addition, we have expanded **Section 5.3** (now spanning **more than 2 pages**) to provide a thorough discussion of population-based and evolutionary methods in self-evolving agents. For example, we discuss recent advancements in this direction, such as Darwin Gödel Machine (DGM), GENOME, EvoMAC, and Puppeteer. We also review the latest works, including SOAR, FELA, and CodeEvolve. Please see the new manuscript for details.
>
> We believe this section strengthens the paper by broadening its scope beyond purely learning-based methods and enriching the discussion, which directly addresses your concern.
>
> > Weakness 3: A crucial omission is the topic of self-play, which ...
>
> **Response:** Thank you for the valuable feedback. We agree that a detailed discussion of self-play is crucial. We have significantly revised self-play part at **Section 5.3.1**  to cover the following content:
> - **A comprehensive overview of self-play**, incorporating foundational works like AlphaZero and recent advancements such as Self-Play Fine-Tuning (SPIN), Absolute Zero, and Self-Challenging Agents. In these methods, an agent learns by interacting with a copy of itself (which may adopt a different role, e.g., a proposer vs. a solver) or with previous versions of itself.
> - **A discussion about difference between self-play and other self-improvement methods.** Briefly, self-play represents a distinct form of self-improvement in which an agent derives its learning signal from interactive, game-like dynamics that it participates in, rather than from external supervision or pre-curated exemplars. This interaction-driven feedback enables improvement even in the absence of perfect or authoritative demonstrations, and naturally supports open-ended skill growth as the agent continually encounters stronger iterations of itself.
>
> We believe these changes properly position self-play as a core paradigm for self-evolution. We appreciate your guidance in helping us improve our work.
>
> Note that in **Section 5.3.2** in our original manuscript, we also discuss the case where the entire team or network of agents co-evolves, which focuses on optimizing collective behavior, coordination strategies, and collaborative architectures. This can be viewed as a complementary paradigm to self-play, where agents mainly interact with other agents.
>
> > Weakness 4: The description of imitation learning could be refined...
>
> **Response:**
> Thank you for this insightful comment. We have followed your suggestion to make the following revisions to **Section 5.2 Imitation and Demonstration Learning**:
> 1. **Clarifying the description of imitation learning**: We have revised the introductory paragraph of the section to first acknowledge the classical definition of imitation learning, which involves learning from expert (often human) demonstrations. We then explicitly clarify how this concept is adapted in the context of self-evolving agents. In this setting, “expert demonstrations” are not limited to human-provided data but may also come from the agent’s own successful trajectories, from other agents, or from synthesized data generated within the environment.
> 2. **Emphasizing the distinction from other methods**: We have substantially enriched this distinction to position imitation learning more clearly within the overall framework. The comparison is framed along two key dimensions:
> - **Nature of the feedback:** Imitation learning uses prescriptive demonstrations (complete guides), whereas reward-based methods --provide evaluative signals requiring exploratory inference.
>
> - **Core evolutionary mechanism:** Imitation learning typically optimizes a single agent lineage, in contrast to population-based methods that evolve a set of agents in parallel through population-level diversity.
>
> We believe these revisions clarify that while the source of the demonstration is different, the core mechanism remains one form of imitation, and how this mechanism is distinct from other forms of self-improvement.

---

> ### Comment · Reviewer_NCWP · 2025-11-24
> **Response**
>
> Thank you for addressing my concerns. I believe the revised version makes a valuable contribution as a survey. Given the rapid progress in this field, it effectively covers the key topics relevant to LLM agents. I recommend acceptance.

---

> > ### Author Response · Authors · 2025-11-25
> >
> > We sincerely thank the reviewer for the positive feedback and recommendation for acceptance. We are glad that the revised version has addressed the concerns and that the survey is viewed as a valuable contribution. We appreciate the recognition of the rapid progress in LLM agents and the importance of covering key topics in this area. Thank you for your constructive comments and support.

---

### Review · Reviewer_orgC · 2025-11-02

**Summary Of Contributions:**

This survey introduces self-evolving agents that enable LLMs to adapt continuously. It organizes the field by evolving components, adaptation stages, and learning mechanisms, and reviews evaluation, applications, and key challenges. It provides a roadmap toward more adaptive, intelligent, and autonomous AI systems.

The principal strength of this survey is its structured "what, when, how" framework, which brings much-needed organization to the rapidly growing domain of self-evolving agents and serves as a useful entry point for newcomers.

**Audience:**

Yes

**Audience Explanation:**

The paper will interest some readers.

**Broader Impact Concerns:**

No ethical concerns.

**Claims And Evidence:**

No

**Claims Explanation:**

1. Limited Conceptual Novelty and Contribution
- The core concept of "self-evolving agents" is not sufficiently distinguished from well-established paradigms like Continual/Lifelong Learning, Self-Improving LLMs, and general Adaptive Agents. The distinctions drawn in Section 2.2 are weak and not formally defined.
- The claim that lifelong learning only updates model parameters is inaccurate; the design and management of memory and other non-parametric components are long-standing and central challenges in continual learning.
- The dichotomy between "passive" (lifelong learning) and "active" (self-evolving) knowledge acquisition is arbitrary and lacks a rigorous definition. Many methods cited as "self-evolving" (e.g., using SFT or RL on pre-collected data) fit a "passive" paradigm.
- This lack of a clear, novel conceptual boundary makes the survey feel like a rebranding of general agent adaptation and training techniques, significantly limiting its original contribution. The overlap with surveys like "A Survey on Self-Evolution of Large Language Models" [1] and "A survey on the memory mechanism of large language model-based agents" [2] is substantial.

2. Overly Broad and Unfocused Scope
- The survey attempts to encompass nearly every aspect of LLM-based agents—including model parameters, memory, tools, architecture, and all major learning paradigms (SFT, RL, ICL)—under the "self-evolving" umbrella.
- This results in a lack of focus, making it difficult to discern what truly constitutes self-evolution versus standard agent capabilities or training procedures. By including everything, the survey fails to carve out a unique and meaningful niche.
- The "What to Evolve" section (Section 3) reads like a general component-wise survey of LLM agents, much of which is covered in [2], rather than a focused analysis of how these components autonomously and continuously improve.

3. Lack of Rigor in the Proposed Taxonomy
- While the "What, When, How" framework is a useful organizational tool, it is not critically justified. The dimensions often map directly onto existing, well-understood concepts (e.g., "When to Evolve" is essentially online vs. offline learning; "How to Evolve" includes standard RL and imitation learning).
- The taxonomy risks being a superficial repackaging of existing ideas rather than a novel analytical framework that provides new insights into the field of self-evolution.

4. Insufficient Critical Analysis and Depth
- The survey is primarily descriptive, cataloging methods without providing a critical analysis of their limitations, scalability, computational costs, or real-world applicability.
- Many cited methods are demonstrated in narrow, simulated environments (e.g., Minecraft, coding benchmarks). The survey does not adequately address the significant challenges of transferring these approaches to complex, open-ended, real-world domains.
- There is a lack of discussion on the maturity and readiness of these techniques for deployment, including their robustness to failure and the assumptions they make about the environment.

5. Inadequate Treatment of Safety and Alignment
   While Section 8.3 briefly mentions safety, it fails to grapple with the unique and amplified risks of self-evolving systems. These include:
- Uncontrolled Behavior Drift: How to ensure an agent's goals and values remain aligned with human intentions as it evolves autonomously.
- Reward Hacking in Open-Ended Evolution: The risk of agents finding and exploiting loopholes in their self-defined reward signals or internal feedback mechanisms.
- Safety of Self-Created Tools: The potential for agents to autonomously generate and use tools in unsafe or malicious ways.
- This is a critical oversight given the paper's emphasis on autonomy and its connection to the ambitious goal of Artificial Superintelligence (ASI).

6. Structural and Organizational Deficiencies
- The paper's structure is encyclopedic and repetitive rather than narrative-driven. It often lists methods without building a compelling story about the field's progression or the trade-offs among different evolutionary approaches.
- The inclusion of very standard techniques (e.g., basic SFT or RL) dilutes the core message and creates redundancy with more general agent surveys.
- Some figures (e.g., Figure 3) are overly complex and not explained clearly in the text, reducing their effectiveness.

7. Superficial Evaluation Discussion
- Section 7 on evaluation correctly identifies the challenge but does not delve deeply into how to benchmark true, long-term, open-ended self-evolution. The metrics and benchmarks discussed are essentially adaptations of those used for static or short-horizon adaptive agents.
- There is a lack of discussion on evaluating the autonomy of the evolutionary process itself—for instance, measuring the degree to which improvement is self-directed versus externally guided.


Other Minor Issues:

   (1) The typo in the Introduction: `prompting strategies (?)fernando2023promptbreeder,` should be fixed.

   (2) The term "self-evolving" is used inconsistently throughout the manuscript, sometimes referring to parameter updates, sometimes to prompt optimization, and sometimes to architectural changes, further contributing to the conceptual fuzziness.


[1] A Survey on Self-Evolution of Large Language Models

[2] A survey on the memory mechanism of large language model-based agents

**Requested Changes:**

Please refer to the comments above.

---

> ### Author Response · Authors · 2025-11-16
> **Response to Reviewer orgC**
>
> We thank the reviewer for the constructive feedback. We have updated the manuscript accordingly, with **newly added content shown in blue** and **revised text marked in red** to facilitate review.
> > Weakness 1: Limited conceptual novelty and contribution
>
> **Response:** We thank the reviewer for highlighting the need for a clearer conceptual boundary. To address this concern, we substantially revised Section 2 to sharpen the distinctions between self-evolving agents and existing paradigms.
> 1. **Added a new paragraph Operational definition of self-evolving agents at the end of Section 2.1**, providing explicit inclusion and exclusion criteria. This clarifies what counts as self-evolution, specifying four evolvable components (parameters, contextual state, toolset, architectural topology) and the requirements of experience-dependence, persistence, and self-initiated adaptation. We also explicitly define our use of "active" vs. "passive", grounding these terms in whether learning is externally triggered or self-initiated (i.e., self-critique to collect SFT data).
> 2. **Revised the lifelong learning paragraph to rectify the earlier oversimplification.** We acknowledge that continual learning employs memory modules, but clarify that these are largely used as training-time aids for parameter optimization, whereas self-evolving agents rely on runtime context that directly shapes behavior at test time. We also emphasize that lifelong learning is driven by externally provided task sequences, while self-evolving agents incorporate exploration, reflection, and self-evaluation to guide their own learning trajectory.
> 3. **Added a new positioning paragraph to explain the conceptual layout of the paradigms.** We distinguish curriculum learning and lifelong learning as **problem-driven paradigms** (addressing difficulty scheduling and catastrophic forgetting) and model editing and self-evolving agents as **solution-driven paradigms** (providing mechanisms for system modification). This perspective makes explicit why these paradigms occupy different roles and how self-evolving agents extend beyond existing categories by supporting multi-component, trajectory-conditioned evolution.
>
> Finally, regarding overlap with prior surveys, we carefully examined the cited works. “A Survey on Self-Evolution of LLMs’’ concentrates on weight-level updates, while “A Survey on Memory Mechanisms’’ focuses on memory architectures. Our survey differs by integrating evolution across model, memory, tools, and architecture, introducing a unified what/when/how taxonomy, and providing the first long-horizon evaluation considerations for system-level evolution. Thus, our contribution is not rebranding but synthesizing heterogeneous evolution mechanisms into a coherent framework.
>
> > Weakness 2: Overly broad and unfocused scope
>
> **Response:**
> We thank the reviewer for raising this important concern and have substantially revised the manuscript to improve scope, focus, and conceptual clarity.
> 1. **Revised Section 3 so that the narrative shifts from describing what each component updates to explaining why these components constitute valid loci of autonomous, trajectory-conditioned evolution.** Concretely, we removed background-style exposition and reframed each subsection around the mechanisms that enable persistent self-modification. This revision strengthens focus by ensuring that Section 3 analyzes why and how components evolve, rather than cataloguing agent subsystems.
> 2. **Added a new paragraph—Operational Definition of Self-Evolving Agents—in Section 2.1.** This addition provides explicit inclusion/exclusion criteria, thereby clarifying the scope of our survey and sharpening its analytical focus.
>
> We would like to clarify that **our intention is not to broaden “self-evolving agents” to include all aspects of agent design**. The apparent breadth is deliberate: existing evolutionary mechanisms are scattered across model finetuning, memory rewriting, prompt evolution, tool creation, and architectural adaptation, yet **lack a unified conceptual framing**. Our survey does not treat all capabilities as self-evolution; rather, it reorganizes only those processes that produce persistent, trajectory-conditioned self-modification into a coherent structure. This helps readers distinguish genuine evolutionary mechanisms from static functionalities. Thus, while the survey is comprehensive, its focus remains clear: **to map and clarify the conceptual landscape of how different learning mechanisms contribute to an agent’s ability to autonomously improve.**

---

> ### Author Response · Authors · 2025-11-16
> **Response to Reviewer orgC (continued)**
>
> > Weakness 3: Lack of rigor in the proposed taxonomy
>
> **Response:**
> We thank the reviewer for the thoughtful comments. We clarify our position as follows:
> 1. **On the justification of the “What–When–How” framework.**
> Our framework intentionally reuses mature and widely accepted dimensions rather than introducing new conceptual primitives. Because these axes are already well-established in the community, they do not require additional theoretical justification. In the revised version, we ground this framework in an explicit **operational definition of self-evolving agents** (Section 2.1) and then link it to concrete **evaluation goals, coverage gaps, and standardized protocols** (Section 7). In particular, Section 7.1 uses the taxonomy to structure “Goals & Metrics” and “Coverage Gaps” for each dimension, and Section 7.2 uses it to guide our **standardized evaluation protocols** and worked example. Our contribution lies in reorganizing existing notions into a coherent structure for understanding self-evolution, not in proposing new categories.
> 2. **On the concern that online/offline learning, RL, and imitation learning are standard methods.**
>  We agree that these learning paradigms are conventional tools. In our survey, they are not positioned as the essence of self-evolution, but as the **mechanisms through which evolutionary processes are implemented**. The core of self-evolution is consistent: persistent, trajectory-conditioned updates to an agent’s internal state, context, tools, or architecture. Standard learning methods simply explain how these updates are realized, rather than defining what self-evolution is.
> 3. **On the scope and intent of the survey.**
> Our goal is **not to propose new paradigms or to broaden “self-evolving agents” to cover all agent capabilities**. Instead, we aim to **reorganize existing ideas through the lens of self-evolution**, providing a coherent and accessible structure that helps readers—especially newcomers—understand how diverse learning mechanisms collectively enable an agent to autonomously improve over time. We **added Operational Definition of Self-Evolving Agents in Section 2.1**. clarifying the scope of our survey and sharpening its analytical focus.
>
> > Weakness 4 Insufficient Critical Analysis and Depth
> > - The survey is primarily descriptive, cataloging methods...
> > - Many cited methods are demonstrated in narrow, simulated environments...
>
> **Response:**
> We thank the reviewer for this important feedback on analysis depth and real-world relevance, and have substantially revised Section 7 to address it.
> 1. **From descriptive catalog to critical analysis and cost/scalability.** In Section 7.1, each evaluation goal (Adaptivity, Retention, Generalization, Efficiency, Safety) now has a **paired structure: “Goals & Metrics” and “Coverage Gaps”**. The former formalizes what should be measured (e.g., FGT/BWT for retention, OOD metrics for generalization, safety scores/harm/leakage, and a refined cost taxonomy for efficiency), while the latter critically analyzes limitations of current methods and benchmarks (episodic resets, narrow task distributions, short horizons, data contamination, missing safety and cost reporting). We also **add Table 7 and Table 8** to map benchmarks to goals and summarize their limitations, and introduce a dedicated **cost taxonomy** with formal Cost-per-Gain and Tool Productivity metrics plus a small worked example (Table 5) to make computational cost and scalability explicit rather than implicit.
> 2. **From narrow simulations to complex, real-world–facing settings and a roadmap for transfer.** We now distinguish early sandbox environments (e.g., Minecraft, text games) from more complex, real-world–facing domains such as GUI/Web agents (WebArena, OSWorld, Mobile-Eval-E, X-WebAgentBench, BrowseComp) and SWE agents (the SWE-bench family). The new “Coverage Gaps” paragraphs explain why results in narrow or static settings do not automatically transfer to open-ended domains (e.g., static websites/repos, lack of non-stationarity, no long-horizon safety or budget constraints). In Section 7.3 (Limitations of Current Evaluation Practices), the **“Underserved Capability Intersections”** subsection further identifies concrete missing pieces that are necessary for real-world deployment: long-horizon retention under privacy/safety constraints, architecture adaptation under latency and token budgets, evolution of the tool ecosystem, and multi-agent safety under collaborative co-evolution.
>
> Overall, these revisions turn Section 7 from a mainly descriptive catalog into a critical, cost-aware, and environment-aware analysis, and outline a concrete research agenda for moving self-evolving agents from current simulated benchmarks toward complex, open-ended, real-world domains.

---

> ### Author Response · Authors · 2025-11-16
> **Response to Reviewer orgC (continued)**
>
> > Weakness 4 Insufficient critical analysis and depth
> > - There is a lack of discussion on the maturity and readiness of these techniques for deployment...
>
> **Response:**
> Thank you for raising this important point. We agree that a survey on self-evolving agents must not only describe techniques but also critically examine their maturity and robustness for real-world deployment. In response, we have added explicit discussions on deployment readiness in two places:
> 1. **New Section 8.3.2: Prescriptive Guardrails and Mitigation Strategies.**
>  Building on your suggestion, this subsection now provides actionable guardrails for safe and robust deployment, including code/tool sandboxing defaults, audit-trail logging for all self-modifying workflows, approval gates for tool creation, rollback/failsafe patterns, and continuous red-team monitoring to detect long-horizon drift. These guardrails are framed precisely to address the concerns you raised: they clarify the assumptions required for safe operation (e.g., deterministic tool interfaces, bounded execution, revocable memory), and they highlight known failure modes where current techniques remain immature.
> 2. **Deployment Readiness and Robustness Discussion.**
>  Throughout Section 8.3, we now explicitly evaluate the maturity of existing methods, noting that many rely on strong assumptions such as stable APIs, predictable environment dynamics, or human availability for approval steps. We contrast these assumptions with realistic deployment settings, calling out where techniques remain experimental and where engineering hardening is still required (e.g., tool-chain isolation, memory validation, runtime verification).
> 3. **Minimal Deployment Checklist (Table 12).**
>  To make your concern actionable, we added a “Minimal Compliance Checklist for Deployment” summarizing required safeguards before deploying self-evolving agents. This table synthesizes robustness considerations across reliability, safety, governance, and environment assumptions (e.g., sandboxing enabled, tool provenance validated, rollback enabled, monitoring in place, environment drift accounted for). It provides a concrete measure of readiness and makes clear which components must be in place before systems can be safely deployed.
>
> These additions ensure that our survey not only reviews methods but also critically assesses their deployment readiness, robustness to failure, and environmental assumptions. We believe this significantly strengthens the practical value of the safety and governance section.
>
>
> > Weakness 5: Inadequate Treatment of Safety and Alignment
>
> **Response:**
> Thank you for the insightful feedback. We have followed your suggestion to rewrite Section 8.3 to provide a more comprehensive analysis of the **unique and amplified risks** inherent to self-evolving agents. Our revised subsection **8.3.1 Emergent Risks in Self-Evolving Systems** directly addresses the core concerns regarding **uncontrolled behavior drift, reward hacking in open-ended evolution**, and the **safety of self-created tools**. Specifically, we introduce the concept of "misevolution" from a recent work [cite1], a type of risk where an agent's autonomous self-evolution leads to unintended or even harmful outcomes:
> - **Uncontrolled behavior drift in model evolution:** We discuss how self-training and agentic finetuning can degrade safety alignment, leading to uncontrolled behavior drift.
> - **Deployment-time reward hacking in memory evolution:** We explain how accumulated memories can lead to "deployment-time reward hacking." We also connect this to the "Alignment Tipping Process (ATP)" [cite2], which explains how even aligned agents can become misaligned during the process of memory accumulation and multi-agent communication.
> - **Safety of self-created and ingested external tools:** We detail the risks of agents creating insecure tools or ingesting malicious code. We also discuss the risk of maliciously using self-evolving agents for cyber attacks [cite3].
>
> We believe these added contents, grounded in the latest literature, provide a deeper discussion of the unique risks of self-evolving agents.
>
> In addition to your advice, we have also introduced **Section 8.3.2** to propose actionable guardrails and mitigation strategies, such as sandboxing, audit trails, approval gates, and red-teaming protocols. These practical measures, together with our detailed risk analysis, strengthen the safety and governance framework for deploying self-evolving agents.
>
> [cite1] Shao et al. Your Agent May Misevolve: Emergent Risks in Self-evolving LLM Agents. https://arxiv.org/abs/2509.26354
> [cite2] Han et al. Alignment Tipping Process: How Self-Evolution Pushes LLM Agents Off the Rails. https://arxiv.org/abs/2510.04860
> [cite3] Wei et al. Dynamic Risk Assessments for Offensive Cybersecurity Agents. https://arxiv.org/abs/2505.18384

---

> ### Author Response · Authors · 2025-11-16
> **Response to Reviewer orgC (continued)**
>
> > Weakness 6: Structural and organizational deficiencies
>
> **Response:**
> We thank the reviewer for these helpful comments. We have revised the manuscript to strengthen narrative coherence, sharpen conceptual focus, and clarify visual explanations. Specifically, we:
> 1. **Refine figure explanations.**
>  Clarify the captions and surrounding textual references for Figures 3, 7, and 9, ensuring that each figure is directly integrated into the conceptual flow rather than functioning as a dense standalone diagram.
> 2. **Strengthen the narrative structure of Sections 3 and 5.**
>  Reframe the exposition to emphasize the conceptual progression of self-evolving mechanisms rather than listing methods. Add concise narrative connectors and trade-off summaries to highlight how different evolutionary approaches emerge, interact, and address each other’s limitations.
> 3. **Expand the discussion of evaluation trade-offs.**
>  Add explicit treatment of self-directedness and evaluation trade-offs in Section 7.1 to clarify how different evolutionary paradigms balance autonomy, reliability, and adaptivity.
>
> We would like to clarify that standard techniques as basic SFT are not positioned as the essence of self-evolution, but as the **mechanisms through which evolutionary processes are implemented**. The core of self-evolution is consistent: persistent, trajectory-conditioned updates to an agent’s internal state, context, tools, or architecture. Standard learning methods simply explain how these updates are realized, rather than defining what self-evolution is.
>
>
>
> > Weakness 7: Superficial evaluation discussion
> > - There is a lack of discussion on evaluating the autonomy of the evolutionary process itself...
>
> **Response:**
> We thank the reviewer for highlighting this important aspect. We have made autonomy of the evolution process explicit in our evaluation framework:
> - In **Section 7.1**, we added the subsubsection **“Self-Directedness and Evaluation Trade-offs”**, where we define self-directedness as the extent to which an agent autonomously generates tasks and evolution strategies instead of following externally designed curricula. We discuss trade-offs using concrete examples: highly self-directed systems such as WebRL and SEAgent benefit from autonomous curriculum generation, while Anthropic’s alignment-faking study shows that unconstrained autonomous evolution cazn sharply increase safety risks.
> - We recommend a **practical criteria for evaluating autonomy** that works (i) specify whether evolution strategies and task sequences are fixed, procedurally generated, or agent-generated; (ii) document the sources of feedback (human labels, rule-based evaluators, self-reflection); and (iii) report the frequency and type of external interventions. These aspects are also reflected in our standardized protocols through logging and human-in-loop statistics.
>
> By making self-directedness a visible reporting dimension and tying it to our trajectory-centric protocols, we aim to help readers judge not only how much an agent improves, but also how much of that improvement is genuinely self-evolved versus externally guided.
>
> > Weakness 7: Superficial evaluation discussion
> > - Section 7 on evaluation correctly identifies the challenge but does not ...
>
> **Response:**
> We thank the reviewer for raising this point. We have extended Section 7.2 to better explain how to benchmark genuine long-horizon self-evolution.
> - **Section 7.2 (Evaluation Paradigm)** now clearly distinguishes three regimes: **Static Assessment, Short-Horizon Adaptive Assessment**, and **Long-Horizon Lifelong Learning Ability Assessment**, summarized in **Figure 9** and **Table 9**. For the long-horizon regime, we emphasize metrics specific to self-evolution: FGT/BWT and forgetting curves for retention, temporal and cluster OOD performance for generalization, CPG and efficiency drift over time for efficiency, and safety drift metrics over extended interaction.
> - The new **"Standardized Evaluation Protocols"** subsection and **Table 10** spell out how to instantiate long-horizon evaluation in practice: full state persistence across tasks, stage-wise and cumulative evolution budgets, detailed trajectory logging (including retention probes and evolution decision logs), and reporting templates tied directly to long-horizon goals. The EvoAgent worked example illustrates both how such a protocol can be realized today and which parts of “true” long-term evaluation are still missing.
>
> Our intention is to give the community a concrete blueprint for long-horizon, open-ended self-evolution evaluation, rather than only adapting static or short-horizon metrics.

---

> > ### Comment · Reviewer_orgC · 2025-12-02
> >
> > Thank you for your detailed rebuttal. I appreciate the effort you’ve made to clarify the conceptual boundaries of self-evolving agents, particularly with the addition of the operational definition in Section 2.1.
> >
> > However, I remain concerned about the overly broad and unfocused scope of the survey. While you emphasize that the contribution lies in synthesizing heterogeneous mechanisms into a coherent framework—not merely rebranding or reorganizing existing ideas—I find that the core concept of "self-evolution" still lacks sufficient rigor and discriminative power.
> >
> > In your response to Weakness 1, you state: “our contribution is not rebranding but synthesizing heterogeneous evolution mechanisms into a coherent framework.” Yet in response to Weakness 3, you note: “Our contribution lies in reorganizing existing notions into a coherent structure… not in proposing new categories.” These statements, while consistent in intent, underscore my concern: this work primarily offers a re-framing of established techniques (e.g., SFT, ICL, RL) under the umbrella of self-evolution, without a sufficiently robust or novel conceptual foundation to justify the synthesis as a distinct paradigm.
> >
> > More specifically, your operational definition hinges on three key criteria: experience-dependence, persistence, and self-initiated adaptation—where “self-initiated” is exemplified by processes such as self-critique to generate training data for SFT. However, this risks conflating well-established data synthesis practices (e.g., using LLMs to generate code or math problems for fine-tuning) with genuine self-evolution. If any model that uses self-generated data for improvement qualifies as “self-evolving,” then the term becomes so broad as to lose its meaning. Many existing pipelines in distillation, bootstrapping, or synthetic data generation would fall under this definition, even when they involve no autonomous architectural, contextual, or behavioral reconfiguration.
> >
> > Furthermore, the notion of “trajectory-conditioned” self-modification remains ambiguous. In the context of LLM agents, nearly all learning and inference processes are conditioned on some form of interaction history or trajectory—whether it's a dialogue context, a sequence of actions in an environment, or a chain-of-thought reasoning path. Without a precise delineation of what constitutes a meaningful trajectory-dependent update—beyond simple data recycling—it is difficult to distinguish self-evolution from standard adaptive training procedures.

---

> ### Author Response · Authors · 2025-12-05
> **Response to Reviewer orgC**
>
> We sincerely thank the reviewer for the continued engagement and for the clarifications offered in this round of comments. **We appreciate the reviewer’s acknowledgment that our previous response resolved most concerns, and we welcome the opportunity to further clarify the survey’s scope and the precision of our definition.** The revisions are shown in cyan.
>
> (1) On the scope and positioning of our survey.
>
> We agree that surveys vary in purpose: some aim to provide newcomers with a broad map of an emerging area, while others focus narrowly on mature, well-established subfields. **Because self-evolving agents represent a nascent research direction, without yet having a consolidated vocabulary, canonical benchmarks, or clearly delineated boundaries, there is limited prior work that would allow us to craft a highly specialized, domain-specific survey. Therefore, our intention is to strike a balance**:
> - Broad enough to provide newcomers with a coherent, high-level overview of all relevant mechanisms that have been invoked in discussions of autonomous or self-improving AI systems.
> - Specific enough to structure these heterogeneous mechanisms into a consistent framework that highlights their functional roles in enabling self-evolution.
> We have revised the introduction and scope statement to emphasize this positioning more clearly. Rather than claiming that self-evolution is a fully established paradigm, we now explicitly frame the survey as a guiding synthesis for a rapidly forming research area, where conceptual boundaries are still being actively negotiated within the community.
>
> (2) On the definition and discriminative power of ''self-evolving agents''
>
> We appreciate the feedback on discriminative power and have formalized the operational definition in Section 2.1 with three specific characteristics to explicitly exclude generic data synthesis pipelines that are agnostic to agent history. Regarding the scope, as clarified in our revision, we do not impose a rigid exclusion threshold that would disregard early-stage developments. Instead, we analyze the field ranging from "proto-evolution" (e.g., iterative bootstrapping) to "strong self-evolution" (fully autonomous diagnosis and reconfiguration). This allows us to provide a comprehensive view of how diverse methods contribute to the paradigm's progression toward full autonomy. Finally, we removed the ambiguous "trajectory-conditioned" terminology to ensure precision.
>
> We hope these clarifications better articulate both the intended contribution of the survey and the rationale behind the operational definition. We again thank the reviewer for the constructive feedback, which has significantly strengthened both the conceptual framing and the presentation of the manuscript

---

> > ### Comment · Reviewer_orgC · 2025-12-06
> >
> > Thank you for your responses. I still believe that the self-evolving agent is essentially a rebranding of existing agent techniques. Since this is the first survey on the topic, I would recommend accepting it.

---

> > > ### Author Response · Authors · 2025-12-06
> > >
> > > We sincerely thank the reviewer for the positive feedback and the recommendation for acceptance. We also appreciate the recognition of the significance of our work as the first survey in this area. Thank you for your constructive comments and support.

---

### Decision · Action_Editor_Edkh · 2025-12-19

**Recommendation:** Accept as is

**Audience:**

Yes

**Audience Explanation:**

This is a timely survey for an emerging topic of LLM based agentic research. The reviewers all agree it's the first survey focused on self-evolving agents that could be useful for a board TMLR audience.

**Claims And Evidence:**

Yes

**Claims Explanation:**

This is a survey paper on self-evolving LLM agents. It's structured under a "what, when, how" framework and provides a comprehensive coverage on related topics. Reviewers expressed concerns in their initial comments around the following aspects:
- clear conceptual boundary: self-evolving vs other related and well-established paradigms
- critical analysis depth
- Inadequate discussion on safety risks
- Inadequate discussion of evolutionary approaches
- Lack of in-depth evaluation discussion

The authors provide a significant revision that addressed the majority of weaknesses. Particularly, it provides an operational definition of self-evolving agents; prescriptive safety/governance guidance; cost-per-gain and tool productivity metrics for evaluation. All reviewers are leaning to acceptance after discussion, although one reviewer is still concerned about the lack of conceptual difference of the newly proposed "self-evolving" vs existing agent techniques.